# Modelling the firn thickness evolution during the last deglaciation: constraints on sensitivity to temperature and impurities

Camille Bréant [1,2], Patricia Martinerie [2], Anaïs Orsi [1], Laurent Arnaud [2] and Amaëlle Landais [1]

[1]Laboratoire des Sciences du Climat et de l'Environnement, UMR8212, CEA-CNRS-UPS/IPSL, Gif-sur-Yvette, France

[2]Univ. Grenoble Alpes, CNRS, IRD, IGE, UMR5001, Grenoble, F-38000, France

The transformation of snow into ice is a complex phenomenon which is difficult to model. Depending on surface temperature and accumulation rate, it may take several decades to millennia for air to be entrapped in ice. The air is thus always younger than the surrounding ice. The resulting gas-ice age difference is essential to document the phasing between $CO_2$ and temperature changes especially during deglaciations. The air trapping depth can be inferred in the past using a firn densification model, or using $\delta^{15}N$ of air measured in ice cores.

All firn densification models applied to deglaciations show a large disagreement with $\delta^{15}N$ measurements in several sites of East Antarctica, predicting larger firn thickness during the Last Glacial Maximum, whereas $\delta^{15}N$ suggests a reduced firn thickness compared to the Holocene. Here we present modifications of the LGGE firn densification model, which significantly reduce the model-data mismatch for the gas trapping depth evolution over the last deglaciation at coldest sites of East Antarctica (Vostok, Dome C), while preserving the good agreement between measured and modelled modern firn density profiles. In particular, we introduce a dependency of the creep factor on temperature and impurities in the firn densification rate calculation. The temperature influence intends to reflect the dominance of different mechanisms for firn compaction at different temperatures. We show that both the new temperature parameterization and the influence of impurities contribute to the increased agreement between modelled and measured $\delta^{15}N$ evolution during the last deglaciation at sites with low temperature and low accumulation rate, such as Dome C or Vostok. We find that a very low sensitivity of the densification rate to temperature has to be used in coldest conditions. The inclusion of impurities effects improves the agreement between modelled and measured $\delta^{15}N$ at cold East Antarctic sites during the last deglaciation, but deteriorates the agreement between modelled and measured $\delta^{15}N$ evolution in Greenland and Antarctic sites with high accumulation unless threshold effects are taken into account. We thus do not provide a definite solution to the firnification at very cold Antarctic sites but propose potential pathways for future studies.

## 1. Introduction

Ice cores are important tools to decipher the influence of different forcings on climate evolution.
They are particularly useful to reconstruct the past variations of polar temperature and greenhouse
gases. The longest record covers 8 last glacial – interglacial cycles (EPICA community members,
2004; Jouzel et al., 2007; Loulergue et al., 2008; Lüthi et al., 2008) and very high resolution climate
records can be retrieved from ice cores drilled in high accumulation regions (Marcott et al., 2014;
Rhodes et al., 2015; WAIS Divide Project Members, 2013, 2015).

Polar ice is a porous medium, and contains bubbles filled with ancient atmospheric air, allowing the
reconstruction of the atmospheric composition in the past. The air is trapped at about 50-120 m
under the ice sheet surface. Above that depth, the interstitial air in firn pores remains in contact
with the atmosphere. Consequently, the air is always younger than the surrounding ice and this age
difference, Δage, can reach several millennia at the low temperature and accumulation rate sites of
East Antarctica.

A precise determination of Δage is essential to quantify the link between temperature changes
recorded in the water isotopic measurements on the ice phase and greenhouse gas concentrations
recorded in the gas phase. Still, quantifying the temporal relationship between changes in
greenhouse gas concentrations in air bubbles and changes in polar temperature recorded in the
isotopic composition of the ice is not straightforward. One way to address this question goes
through the development of firn densification models that depict the progressive densification of
snow to ice, and the associated decrease of porosity. Below a certain threshold density, the pores
seal off and the air is trapped. The firn densification models thus calculate the Lock-in Depth
(hereafter LID) according to surface climatic conditions. A higher temperature accelerates the firn
metamorphism and leads to a shallower LID. On the other hand, a higher snow accumulation at the
surface will have the effect of increasing the firn sinking speed and hence deepening the LID.
On glacial – interglacial timescales, increasing temperature is associated with increasing snow
accumulation. Indeed, the thermodynamic effect dominates when dealing with long term averages
(several thousands of years), even if accumulation and temperature are not always correlated on
millennial and centennial timescale in polar regions, especially in coastal areas (e.g. Fudge et al.,
2016; Altnau et al., 2014). As a consequence, we observe for all available ice cores covering the last
deglaciation joint increases in both accumulation and temperature. In the firn densification model,
both effects partially compensate each other, with the temperature effect being dominant in the
current densification models for the LID simulation over glacial – interglacial transitions in deep
drilling sites of the East Antarctic plateau, hence leading to the modelled LID decrease.
A first class of densification models is based on an empirical approach to link accumulation rate and
temperature at different polar sites to densification rates (allowing the match between the
modelled and the measured density profiles) (e.g. Herron and Langway, 1980). The Herron and
Langway (1980) model assumes that the porosity (air space in the firn) variations directly relate to
the weight of the overlying snow, hence the accumulation rate. A temperature dependence
following an Arrhenius law is also implemented to account for a more rapid compaction at higher
temperature. Finally, the exact model sensitivity to temperature and accumulation rate is adjusted
empirically in order to simulate observed density profiles. Measured density profiles exhibit
different densification rates above and below 550 kg/m$^3$ so that different empirical laws are used
for densities above and below this threshold. Indeed, 550 kg/m$^3$ corresponds to the observed
maximum packing density of snow (e. g. Anderson and Benson, 1963), hence to a change in the
driving mechanism of firnification.

Despite its simple empirical description, and although more sophisticated empirical models have
been developed (Arthern et al., 2010; Helsen et al., 2008; e.g. Li and Zwally, 2004; Ligtenberg et al.,
2015), the Herron and Langway (1980) firn model often provides good quality results and is still used
in a number of ice core studies (e.g. Buizert et al., 2015; Overly et al., 2015, Lundin et al., 2017).
However, its validity is questionable when used outside of its range of calibration, such as glacial
periods at cold sites of the East Antarctic plateau for which no present-day analogue exists. As a
consequence firn models including a more physical description of densification have been
developed (e.g. Arnaud et al., 2000; Salamatin et al., 2009). The model developed over the past 30
years at LGGE (Arnaud et al., 2000; Barnola et al., 1991; Goujon et al., 2003; Pimienta, 1987) aims
at using a physical approach which remains sufficiently simple to be used on very long time scales
(covering the ice core record length). More complex models, explicitly representing the material
micro-structure have been developed but require a lot more computing time (Hagenmuller et al.,
2015; Miller et al., 2003). Still, the simplified physical mechanisms in our model include parameters
adjusted through comparison of modelled and measured present-day firn density profiles which
may induce biased results outside the range of calibration.

In parallel to firn densification modelling, past firn LID can also be determined using the $\delta^{15}N$
measurements in the air trapped in ice cores. Indeed, in the absence of transient thermal gradients,
the $\delta^{15}N$ trapped at the bottom of the firn is mainly related to the diffusive column height (DCH).
This is due to gravitational settling in the firn following the steady state barometric equation (Craig
et al., 1988; Schwander, 1989; Sowers et al., 1989):

$$\delta^{15}N_{grav} = \left[\exp\left(\frac{\Delta m g z}{R T_{mean}}\right) - 1\right] 1000 \approx \frac{g z}{R T_{mean}} \Delta m \times 1000 \ (‰) \tag{1}$$

Where $\Delta m$ is the mass difference (kg/mol) between $^{15}N$ and $^{14}N$, $g$ is the gravitational acceleration
(9.8 m/s$^2$), $R$ is the gas constant (8.314 J/mol/K), $T_{mean}$ is the mean firn temperature (K), and $z$ is the
diffusive column height noted (DCH). In the absence of convection at the top of the firn, the firn LID
is equal to the DCH.

In Greenland ice cores, where strong and abrupt surface temperature changes occurred during the
last glacial period and deglaciation, $\delta^{15}N$ is also affected by strong thermal fractionation. An abrupt
warming (on the order of 10°C in less than 50 years) indeed induces a transient temperature
gradient in the firn of a few degrees (Severinghaus et al., 1998; Guillevic et al., 2013; Kindler et al.,
2014). $\delta^{15}N$ is thus modified as $\delta^{15}N_{therm} = \Omega * \Delta T$, where $\Omega$ is the thermal fractionation coefficient
(Grachev and Severinghaus, 2003) and this thermal signal is superimposed on the gravitational one
(the $\delta^{15}N_{therm}$ observed is in most cases lower than 0.15‰).

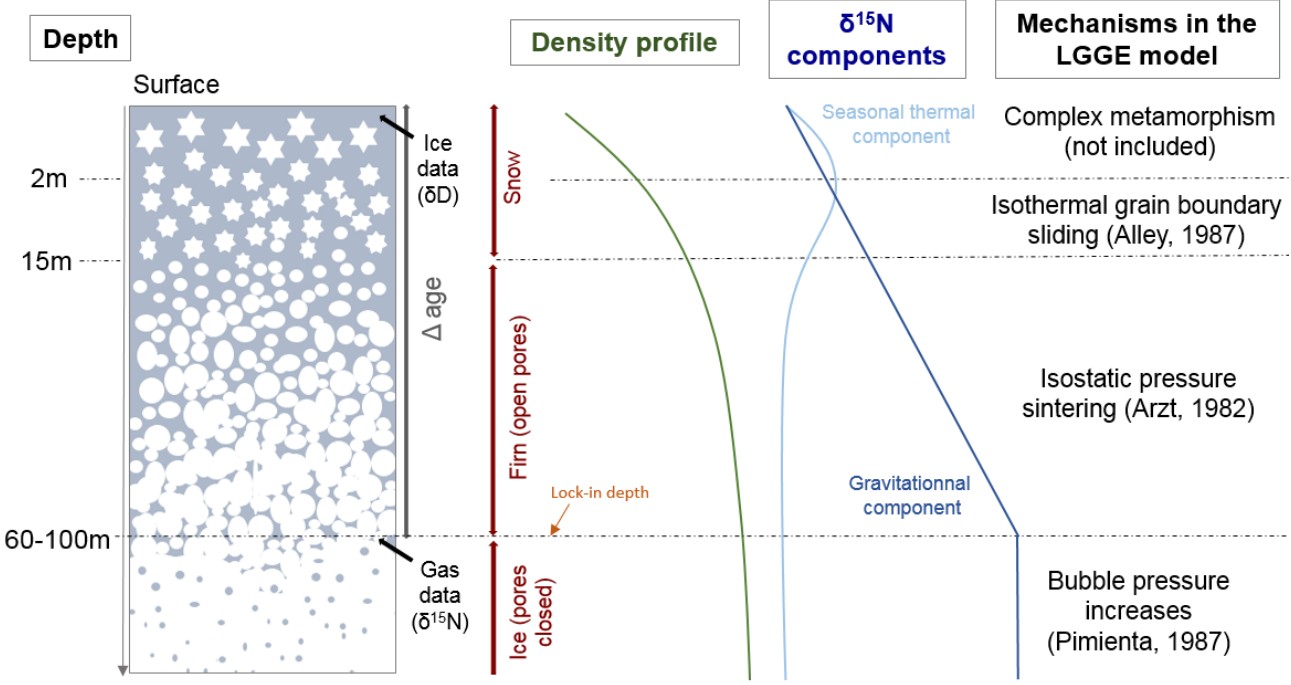

*Figure 1: Overview of snow densification and influence on the $\delta^{15}N$ profile in the absence of any significant convective zone as observed in most present-day $\delta^{15}N$ profiles (Landais et al., 2006; Witrant et al., 2012).*

While models can reproduce the observed $\delta^{15}N$ at Greenland sites over the last climatic cycle, a strong mismatch is observed for cold Antarctic sites, especially on the East-Antarctic plateau (Dreyfus et al., 2010). In particular, both the empirical and physical models predict a decrease of the LID during glacial to interglacial transitions (Goujon et al., 2003; Sowers et al., 1992) while the $\delta^{15}N$ evolution indicates an increase of the LID (Capron et al., 2013; Sowers et al., 1992). The decrease in the LID in the models is caused by the increase in temperature during the deglaciation, which has a stronger impact than the increase in the accumulation rate. The differences in modelled and measured $\delta^{15}N$ for glacial periods in cold sites of the East-Antarctic plateau have important consequences for the $\Delta$age estimate and hence the ice core chronology: using the firn densification models, the modelled $\Delta$age for glacial period at Vostok and Dome C is too large by several centuries (Loulergue et al., 2007; Parrenin et al., 2012).

Several hypotheses have already been invoked to explain the $\delta^{15}N$ model-data mismatch in Antarctica as detailed in Landais et al. (2006), Dreyfus et al. (2010) and Capron et al. (2013). First, the firnification models have been developed and tuned for reproducing present-day density profiles and it is questionable to apply them to glacial climate conditions in Antarctica for which no present-day analogues are available. Second, increasing impurity concentration has been suggested to fasten firn densification during glacial period (Freitag et al., 2013; Hörhold et al., 2012). Third, a

~20 m deep convective zone has been evidenced in the megadunes region in Antarctica (Severinghaus et al., 2006) hence suggesting that deep convective zones can develop in glacial periods in Antarctica and explain the mismatch between firn densification model and $\delta^{15}$N data (Caillon et al., 2003). This hypothesis can explain the mismatch between modelled and measured $\delta^{15}$N at EDML during glacial period by invoking a 10 m convective zone (Landais et al., 2006). However, it has been ruled out for explaining the strong mismatch between model and $\delta^{15}$N data at EDC for the last glacial period (Parrenin et al., 2012). Fourth, firn densification is very sensitive to changes in temperature and accumulation rate so that uncertainties in the surface climate parameters can lead to biased value of the modelled LID and hence $\delta^{15}$N. Fifth, a significant thermal fractionation signal can affect the total $\delta^{15}$N signal. However, this hypothesis has been ruled out by Dreyfus et al. (2010) based on $\delta^{15}$N and $\delta^{40}$Ar data on the last deglaciation at EDC.

In this study, we test whether simple modifications of the LGGE model can reduce the model-data mismatch for the LID evolution over the last deglaciation in sites on the East Antarctic plateau. In particular, it has been suggested by Capron et al. (2013) that the firn densification rate is underestimated at very low temperature. We also examine the possible influence of impurity concentration in the LGGE model following the approach by (Freitag et al., 2013; Hörhold et al., 2012). The manuscript is organized as follows. In the next (second) section we present the physical model with a focus on recent modifications. In a third section, we confront the model output to present-day observed firn density profiles and $\delta^{15}$N data over the last deglaciation at different polar sites from Greenland and Antarctica. Section 4 summarizes our conclusions.

## 2. Densification model description and improvements

An in-depth description of the LGGE firn densification model is provided in Goujon et al. (2003). Here we first briefly summarize its content, and then detail the modifications introduced in this study. The main inputs to the model are temperature and snow accumulation rate (Supplementary Text S1). During climatic transitions occurring at similar or shorter time scales than firnification, the propagation of the atmospheric temperature signal into the firn has to be taken into account (Schwander et al., 1997). The thermo-mechanical model comprises four modules. A simple ice sheet flow module calculates the vertical speed in a 1D firn and ice column. This vertical speed is used in the thermal module to calculate heat advection. The thermal module solves the heat transfer equation, which combines heat advection and heat diffusion across the whole ice-sheet thickness.

Using the resulting temperature profile in the firn, the mechanical module evaluates the
densification rates resulting from three successive mechanisms detailed below. Finally, a gas-age
module keeps track of snow layers sinking in a Lagrangian mode and uses a gas trapping criterion in
order to evaluate the gas trapping depth and the ice age – gas age difference (Δage).
The model does not take into account the complex mechanisms associated with snow
metamorphisms under the influence of strong temperature gradients, wind and sublimation/re-
condensation (Colbeck, 1983; Kojima, 1967; Mellor, 1964). This kind of metamorphism affects the
1-3 meters at the top of the firn and has a minor role on the modelled LID.
Below this depth, the densification of snow into ice has been divided into three stages (e.g. Maeno
and Ebinuma, 1983 and references therein; Figure 1). The first stage, named "snow densification"
as in Goujon et al. (2003), corresponds to a rearrangement and packing of snow grains until
approaching the maximum compaction at a density of about 550 kg/m$^3$ (or 0.6 on a unitless scale
relative to the density of pure ice) defined as the critical density. The second stage represents the
"firn densification" by sintering associated with visco-plastic deformation. Finally, when the bubbles
are closed (at a relative density of about 0.9), the ice densification is driven by the difference in
pressure between air trapped in bubbles and the solid ice matrix subject to the weight of the
overlying firn structure. In reality, the adjacent densification mechanisms likely coexist at
intermediate densities. Below we further describe the mechanical structure of the model with a
focus on recent modifications and proposed parameterizations. We refer to Arnaud et al. (2000)
and Goujon et al. (2003) for more details.

The model uses macroscopic (simplified) mechanical laws, which link the densification speed
(dD$_{rel}$/dt, in terms of relative density ($D_{rel} = \frac{\rho}{\rho_{ice}}$)) to its main driving force: the overburden
pressure of overlying snow. It is important to note that in our model, the accumulation rate
influences firn densification only through the overburden pressure:

$$P(h) = g \int_o^h \rho dz \qquad\qquad (2)$$

where g is the gravity constant and ρ is the density in kg/m$^3$. This differs from the Herron and
Langway (1980) model where the effect of accumulation rate is adjusted and expressed with a
different power law for snow and firn densification rates. In porous materials, the overburden
pressure P is transmitted through contact areas between grains rather than the entire surface of
the material. This is expressed by replacing P with an effective pressure P$_{eff}$ in mechanical stress-
strain laws. The relationship between P and $P_{eff}$ depends on the material geometry (e.g. Equation
A4 in Goujon et al., 2003). A higher temperature (T) facilitates the deformation of materials, and
this effect is commonly represented by an Arrhenius law: $e^{\left(\frac{-Q}{RT}\right)}$ where R is the gas constant and Q
an activation energy. The value of the activation energy depends on the underlying physical
mechanism of ice and snow deformation but Arrhenius expressions cannot represent deformation
effects linked to ice melting. The relationships between densification speed and overburden
pressure take the following general form:

$$\frac{dD_{rel}}{dt} = A_0 \times e^{\left(-\frac{Q}{RT}\right)} \times (P_{eff})^n \tag{3}$$

where $A_0$ = 7.89x $10^{-15}$ Pa$^{-3}$.s$^{-1}$ (Goujon et al., 2003, Eq. A5) and n is the stress exponent. In the rest
of the manuscript, we will refer to $A = A_0 \times e^{\left(-\frac{Q}{RT}\right)}$ as the creep parameter.

## 2.1 Densification of snow

During the first stage, the dominant snow densification mechanism is assumed to be isothermal
boundary sliding and the model of Alley (1987) is used (Figure 1). The geometrical approximation
used to build the model is to represent snow as equal size spheres with a number of contacts
between neighbours increasing with density. In the LGGE model, the Alley mechanism is
implemented as Equation A1 in Goujon et al. (2003):

$$\frac{dD_{rel}}{dt} = \gamma \left(\frac{P}{D_{rel}^2}\right)\left(1 - \frac{5}{3} \times D_{rel}\right) \tag{4}$$

It directly relates to Equation (5) in Alley (1987):

$$\frac{dD_{rel}}{dt} = \frac{2}{15} \times \frac{\lambda}{v} \times \frac{R}{r^2} \times \left(1 - \frac{5}{3} * D_{rel}\right) \times \frac{P}{D_{rel}^2} \tag{5}$$

where $\lambda$ is the bond thickness, $v$ the bond viscosity, $R$ the grain radius and $r$ the bond radius. $P$ is
expressed as a function of accumulation and gravity (Equation 2).
The important simplification in the LGGE model is the replacement of geometry dependent
parameters, not available for past conditions, with a variable $\gamma$, adjusted in order to obtain a
continuous densification rate at the boundary between the first and the second stage of
densification.
A first modification in this module consists of extending the Alley (1987) scheme to the upper two
meters of the firn rather than using a constant density value. Indeed, since the model is not able to
represent the metamorphism of the first two meters, we impose a constant pressure of 0.1 Bar (see
Equation 6), which is an approximation of the pressure at 2-3 m depth. It results in a nearly constant
densification rate in the top 2-3 m rather than a constant density in the top 2 meters.
The second modification concerns the transition between the snow and firn densification stages at
the relative density of 0.6. In Equation (4), the term $\left(1 - \frac{5}{3} \times D_{rel}\right)$ implies that the densification
speed drops to zero at $D_{rel} = \frac{3}{5}$ (i.e. 0.6 the maximal compaction density). The second stage of
densification (firn densification) is driven by an important overburden pressure on the contact area
hence associated with a high densification speed. The transition between the sharp decrease of the
densification speed for $D_{rel}$ values close to 0.6 in the snow densification stage and the high
densification speed at the beginning of the firn densification (i.e. in the same range of value for $D_{rel}$)
causes some model instabilities especially at sites with high temperature and accumulation rate. In
order to improve the model stability, we go back to the definition of the term $\left(1 - \frac{5}{3} \times D_{rel}\right)$ in the
initial formulation of Alley (1987). This term relies on a correlation between the coordination
number (N) and relative density: $D_{rel} = 10\ N$. We slightly modified this relationship and impose $D_{rel} =$
*10 N - 0.5* which better matches the data on Figure 1 of Alley (1987). This results in replacing the
term $\left(1 - \frac{5}{3} \times D_{rel}\right)$ in Equation (4) with $\left(1 + \frac{0.5}{6} - \frac{5}{3} \times D_{rel}\right)$. This modification shifts the density at
which the densification rate becomes zero from 0.6 to 0.65 and suppresses the model instability.

We also examine the effect of temperature on the first-stage densification mechanism and on the
critical density. Alley (1987) calculated a viscosity ($v$) related activation energy of 41 kJ/mol,
consistent with recommended values for grain-boundary diffusion (42 kJ/mol) or measured from
grain growth rate (Alley, 1987 and references therein). In Goujon et al. (2003), no explicit
temperature effect is used but the parameter $\gamma$ varies by several orders of magnitude from site to
site. The parameter $\gamma$ is calculated to maintain a continuous densification rate between the first and
second stages at a chosen critical density. We translate the variations from site to site of $\gamma$ = (2 $\lambda$ R)
/ (15 $v\ r^2$), where $\lambda$ is the bond thickness, *R* the grain radius, $v$ the bond viscosity and *r* the bond
radius (as in Equation 5), into $\gamma$ = $\gamma'$ exp(-Q/RT) , and calculate the activation energy Q using a
classical logarithmic plot as a function of 1000/T (see e.g. Herron and Langway, 1980). We obtain a
value of 48 kJ/mol. Using the revised temperature dependency for the firn densification mechanism
(see next section), a slightly higher value of Q =49.5 kJ/mol is calculated (Supplementary Figure S1).
This is fairly similar to the values in Alley (1987) but much higher than the value in the upper firn of
the Herron and Langway (1980) model: 10.16 kJ/mol. Incorporating this explicit temperature
dependency term, we obtain our new final expression for the upper firn densification rate:

$$\frac{dD_{rel}}{dt} = \gamma' \left( \frac{\max(P, 0.1\ bar)}{D_{rel}^2} \right) \left( 1 + \frac{0.5}{6} - \frac{5}{3} \times D_{rel} \right) \times e^{\left( -\frac{Q}{RT} \right)} \qquad (6)$$

where $\gamma' \times e^{\left( -\frac{Q}{RT} \right)}$ is equivalent to $\gamma$ in Equation (4). However $\gamma$ varies by two orders of magnitude
as a function of temperature whereas $\gamma'$ remains in the range from $0.5 \times 10^9$ to $2 \times 10^9$ bar$^{-1}$.
Finally, the temperature dependency of the critical density, which defines the boundary between
the first and second stage densification mechanisms, is also re-evaluated. According to Benson
(1960) and Arnaud (1997; 2000), this critical density increases with temperature. However the slope
change in density profiles associated with the critical density may be difficult to locate and the
Benson (1960) and Arnaud (1997) parameterizations are based on only few observation sites. We
evaluate the critical density values which allow the best match of density data by our model results
at 22 sites and do not find any correlation between critical density and temperature or accumulation
rate (Supplementary Figure S2). We thus remove this dependency with temperature included in the
old version of the LGGE model and use a mean relative critical density of 0.56 at the boundary
between the first and second stage of densification in the new version of the model. The effect of
surface density was also tested and does not have a strong impact on the model results
(Supplementary Figure S3).

2.2 Densification of firn

At this stage, the observation of density profiles with depth suggests that the densification rate is
controlled by a classical power law creep as used for ice deformation (Arzt et al., 1983; Maeno and
Ebinuma, 1983; Wilkinson and Ashby, 1975). Arzt (1982) proposed a pressure sintering mechanism
for firn densification following a power law creep and taking into account the progressive increase
of the coordination number. He solved the geometrical problem of compressing a random dense
packing of monosized spheres with associated deformation of each sphere into irregular polyhedra.
Equation (23) of Arzt (1982) is directly used in the firn densification model.

2.2.1 Revised temperature sensitivity of the firn densification rate

A strong assumption in the firn densification module is the constant activation energy corresponding to self-diffusion of ice (60 kJ/mol). This choice corresponds to a unique mechanism supposed to drive densification. Densification is thus assumed to be driven by dislocation creep (Ebinuma and Maeno, 1987) in which the associated mechanism is lattice diffusion or self-diffusion. At the grain scale, we can describe the lattice diffusion processes associated with dislocation as diffusion within the grain volume of a water molecule from a dislocation site in the ice lattice to the grain neck in order to decrease the energy associated with grain boundaries (Blackford, 2007). Typically, an activation energy of 60 to 75 kJ/mol is associated with this mechanism (Arthern et al., 2010; Barnes et al., 1971; Pimienta and Duval, 1987; Ramseier, 1967 and references therein).

However, multiple studies have already shown that several (6 or more) mechanisms can act together for firn or ceramic sintering (Ashby, 1974; Blackford, 2007; Maeno and Ebinuma, 1983; Wilkinson and Ashby, 1975): lattice diffusion from dislocations, grain surfaces or grain boundaries; vapor transport; surface and boundary diffusions. In order to properly take these different mechanisms into account, different activation energies (one activation energy per mechanism) should ideally be introduced in the firn densification model. Actually, it has been observed that, at warm temperature, an activation energy significantly higher than 60 kJ/mol could be favoured (up to 177 kJ/mol between -1 and -5°C [Jacka and Li, 1994]) in order to best fit density profiles with firn densification models (Arthern et al., 2010; Barnes et al., 1971; Jacka and Li, 1994, Morgan, 1991). This suggests that a mechanism different from lattice diffusion is dominant for grain compaction at high temperature (i.e. higher than -10°C). At low temperature (-50°C), by analogy with ceramic sintering, lattice diffusion from the surface of the grains and/or boundary diffusion from grain boundaries should be favoured (Ashby, 1974). The activation energy for surface diffusion is estimated to be in the range 14-38 kJ/mol (Jung et al., 2004; Nie et al., 2009).

Following these arguments and despite the lack of experimental constraints to test this assumption, we propose a new heuristic parameterization of the activation energy in the LGGE firn densification model which increases the firn densification rate at low temperatures. We have thus enabled introduction of three adjusted activation energies as proposed in Table 1 and Figure 2. We have replaced the creep parameter in Equation (3) by:

$$A = A_0 \times \left( a_1 \times e^{\frac{-Q_1}{RT}} + a_2 \times e^{\frac{-Q_2}{RT}} + a_3 \times e^{\frac{-Q_3}{RT}} \right) \tag{7}$$


We have chosen a minimal number of mechanisms (3) for simplicity in the following but the
conclusions of our work would not be affected by a choice of more mechanisms.

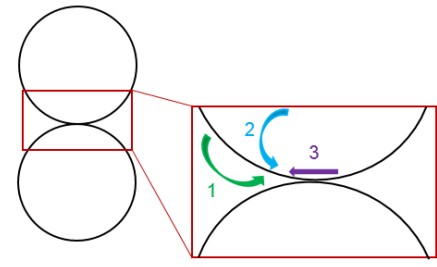

- (1) mechanism 1: close to melting temperature - mass transfer by diffusion (potential mechanism for high temperature)

- (2) mechanism 2: low temperature - lattice diffusion (classical mechanism)

- (3) mechanism 3: very low temperature - boundary diffusion from grain boundary (potential mechanism for low

temperature)

*Figure 2: Different sintering mechanisms of snow for different temperatures proposed by analogy with the*
*hot ceramic sintering (inspired by Figure 1 in Ashby, 1974). Note that more sintering mechanisms can be found*
*in the literature: in its initial figure, Ashby (1974) mentioned 6 different mechanisms but only 2 permit*
*densification (lattice diffusion and boundary diffusion from grain boundary). The attributions of 3 different*
*mechanisms for the firn densification model based on the powder aggregate study from Ashby (1974) is only*
*a working hypothesis here.*

When building the new parameterization of the activation energy (Equation 7), the determination
of $Q_1$, $Q_2$ and $Q_3$ on the one side and $a_1$, $a_2$ and $a_3$ on the other side are not independent from each
other. We first determine three temperature ranges corresponding to the dominant mechanisms,
then we attribute values to the activation energies $Q_1$, $Q_2$ and $Q_3$. The coefficients $a_1$, $a_2$ and $a_3$ are
finally adjusted to produce the expected evolution of the creep parameter with temperature, to
best reproduce $\delta^{15}N$ evolution over deglaciations (Section 3.2) and respect the firn density profiles
available (Section 3.1).
Hundreds of sensitivity tests have been performed imposing 3 activation energies at 3 different
typical temperatures, $T_i$. The initial values for $Q_i$ are chosen as explained above (high value for $Q_1$
[Jacka and Li, 1994], classical value between 60 and 70 kJ/mol for $Q_2$ and low value for $Q_3$ to increase
the densification rate at low temperature). The initial values for $a_i$ are derived through $a_i*\exp(-$
$Q_i/RT_i)=a_0*\exp(-60000/RT_i)$ and variations around the initial values of $Q_i$ and $a_i$ are randomly
generated. Only the values leading to realistic densification speed are kept and we found the
optimal tuning through reduction of the mismatch between model and data especially for the
deglacial amplitude of $\delta^{15}N$ in Dome C and Vostok. The constraint of keeping a correct agreement
of model results with present day density profiles and for the last deglaciation at warm sites strongly
reduces the possible choices of $a_i$ and $Q_i$ (Section 3). The best value obtained for $Q_3$ is lower than
published values for surface or boundary diffusion but is necessary to reproduce the deglaciation at
cold East Antarctic Sites. Sensitivity test C will illustrate the effect of using a higher value.

The resulting expression for the creep parameter A (Equation 7), does not strongly differ from using
simply $A = A_0 \times e^{\left(-\frac{60000}{RT}\right)}$, as used in the original model. To illustrate this point, we calculated an
equivalent activation energy, $Q_{eq}$, such that $A = A_0 \times e^{\left(-\frac{Q_{eq(T)}}{RT}\right)}$, and found $Q_{eq}$ varying between
54 and 61 kJ/mol (Supplementary Figure S4). Thus only moderate changes to the densification
equation are needed to improve the behaviour of the model at cold temperature. In addition, only
moderate changes in $Q_{eq}$ are allowed to preserve the consistency between model results and
present-day density profiles.

| Activation Energy (J/mol) | Coefficient |
|---|---|
| $Q_1$= 110000 | $a_1$= 1.05*10$^9$ |
| $Q_2$= 75000 | $a_2$= 1400 |
| $Q_3$= 1500 | $a_3$= 6.0*10$^{-15}$ |


*Table 1: Preferred set of values for the three activation energies and associated pre-exponential constants*

2.2.2 Sensitivity of the firn densification rate to impurities

Firn densification can be influenced by impurity content in snow. Alley (1987) already suggested
that grain growth is influenced by impurities dissolved in ice, and that impurities in the grain
boundaries affect the relative movement of snow grains. More recently, Hörhold et al. (2012)
observed a correlation between the small scale variability of density and calcium concentration in
Greenland and Antarctic firn cores. Based on this observation, Freitag et al. (2013) proposed that
the densification rate depends on the impurity content. They implemented an impurity
parameterization in two widely used densification models (Herron and Langway, 1980; Barnola et
al., 1991), and were able to reproduce the density variability in two firn cores from Greenland and
Antarctica.

We have implemented this parameterization in our model with the simple assumption that the
impurity effect is the same for all mechanisms. It allows us to keep the number of tunable
parameters to a minimum, even though this assumption is probably not correct for the vapor
diffusion process. Note however that this will not affect the applications discussed below since
vapor diffusion is only important for warm sites. Concretely, we start again from the evolution of
the creep parameter with respect to temperature given in Equation (7) and add a dependency to
calcium concentration such as:

$if \ [Ca^{2+}] > [Ca^{2+}]_{crit} : \ Q' \ = \ f_1 \ \times \left[ 1 - \beta \ln \left( \frac{[Ca^{2+}]}{[Ca^{2+}]_{crit}} \right) \right] \times Q$        (8)
$if \ [Ca^{2+}] < [Ca^{2+}]_{crit} : \ Q' \ = \ f_1 \ \times Q$        (9)

With, $[Ca^{2+}]_{crit}$ = 0.5 ng/g (the detection limit of continuous flow analysis). Q' represents the new
activation energy calculated as a function of the calcium concentration for each site. Our main
simulations are performed with the $f_1$ and β calculated by Freitag et al. (2013) for application within
the Herron and Langway model: $f_1$ = 1.025, β = 0.01. Using the values for application within the
Pimienta-Barnola model ($f_1$ = 1.015, β = 0.0105) leads to similar results (Section 3.2). For a first
evaluation of the impurity effect in our model, both the temperature and impurity effects are
combined through the application of Equations (8) and (9) to each of the three different activation
energies $Q_1$, $Q_2$ and $Q_3$. We use raw data of the calcium concentration for all the sites when available
even if question may arise on calcium concentration being the best diagnostic for dust content.
The values of $a_i$ and $Q_i$ were not readjusted after the implementation of impurity effects to avoid
adding tuning parameters. Still, because the large range of calcium concentrations encountered in
past climate conditions has a strong impact on model results, this may be a solution to reduce the
model-data mismatch. This is explored in Section 3 through a sensitivity test D. In the same section,
we will also propose a modification of the Freitag parameterization using thresholds to reduce the
model-data mismatch.

2.3 Densification of ice

As in Goujon et al. (2003), the final densification stage begins at the close-off density derived from
air content measurements in mature ice. Further porosity reduction results in an air pressure
increase in the bubbles (Martinerie et al., 1992, Appendix 1). This density is calculated using the
temperature dependent close-off pore volume given by Martinerie et al. (1994). Further
densification of this bubbly ice is driven by the pressure difference between ice matrix and the air
in bubbles (Maeno and Ebinuma, 1983; Pimienta, 1987). The densification rate strongly decreases
with depth as these two opposite pressures tend to balance each other (Goujon et al., 2003). This
stage is not essential for this study since $\delta^{15}N$ entrapped in air bubbles does not evolve anymore.

2.4 Lock-in depth

In the previous version of the model, the LID was computed as a fixed closed to total porosity ratio.
The ratio value used has been adjusted for each drilling site, for example it is 21% for Vostok and
13% at Summit in Goujon et al. (2003), but it was time independent and thus insensitive to climate.
We revised the LID definition in order to relate its present day geographic variations to climatic
parameters.

Ideally, $\delta^{15}N$ profiles in the open porosity of the firn follow the barometric slope in the diffusive
zone, and show no variations in the lock-in zone. However $\delta^{15}N$ data can deviate from this
behaviour, especially at the very low accumulation rate sites such as Dome C, Vostok or Dome Fuji,
where no $\delta^{15}N$ plateau is observed in the lock-in zone (Bender et al., 1994; Kawamura et al., 2006;
Landais et al., 2006). Moreover, as we aim at comparing our model results with $\delta^{15}N$ data in deep
ice cores, the most consistent LID definition should refer to $\delta^{15}N$ data in mature ice but very few
measurements are available for recent ice. Systematic $\delta^{15}N$ measurements in the closed porosity of
the deep firn or recently formed mature ice would be very helpful to better constrain the LID in the
future. We take advantage of recent advances in gas transport modelling (Witrant et al., 2012) that
allowed correct simulation of the $\delta^{15}N$ behaviour in deep firn. Observations of modern firn air
profiles show that the thickness of the lock-in zone (the zone in the deep firn with constant $\delta^{15}N$)
increases when the snow accumulation rate increases (Witrant et al., 2012). We estimate $\delta^{15}N$ in
ice, i.e. after complete bubble closure, at 12 firn air pumping sites with the Witrant et al. (2012)
model. For each site, the lock-in density ($\rho_{LI}$) is then defined as the density at which the modelled
$\delta^{15}N$ value in the open porosity of the firn equals the modelled $\delta^{15}N$ in ice. The resulting lock-in
density is strongly related to the accumulation rate (Supplementary Figure S5). As a result, we
parameterized the lock-in density ($\rho_{LI}$) as a function of the accumulation rate, following:

$\rho_{LI} = 1.43 \times 10^{-2} \times \ln (1/Ac) + 0.783$    (10)

This parameterization leads to $\rho_{LI}$ variations in the range 780-840  kg/m$^3$ (Supplementary Figure S5)
and a much better agreement between the modelled LID and $\delta^{15}N$ measured in firn samples at

available sites than when using a fixed closed / total porosity ratio. However, when used for simulating the LID during glacial periods with extremely low accumulation rate, it can predict a lock-in density that is higher than the close-off density, which is unrealistic. We thus also added a threshold in our new definition of the lock-in density: when $\rho_{LI}$ exceeds the close-off density ($\rho_{CO}$, Section 2.3), we impose $\rho_{LI}$ to be equal to $\rho_{CO}$.

## 3. Results

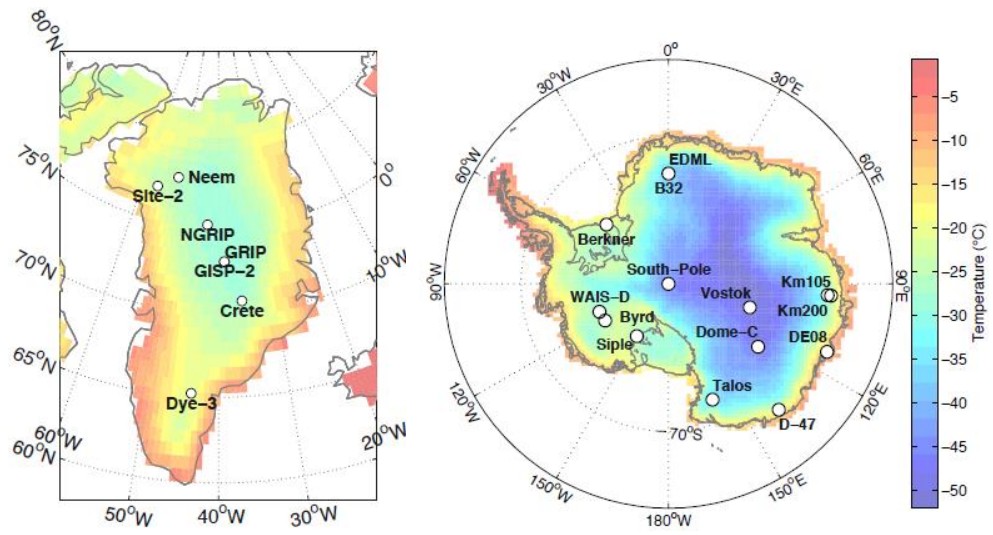

*Figure 3: Maps of Greenland and Antarctica showing field sites and mean annual temperature from ERA interim (Dee et al., 2011)*

### 3.1 Firn density profiles

We assessed the behaviour of the model by comparing measured and modelled firn density profiles from 22 sites from Greenland and Antarctica (Figure 3). Figure 4 shows this comparison at Byrd, NEEM, Dome C and Vostok, and other sites are displayed in the supplement (Supplementary Figure S6). A polynomial fit was adjusted to the density data in order to facilitate the comparison with model results. The data dispersion around the fit can be due natural density variations and/or measurement uncertainties.

A comparison of snow density measurement methodologies concluded that uncertainties are about 10 % (Proksch et al., 2016). Moreover, although firn density profiles are often used, the measurement technique is not always well documented. Efforts were made in this study to mention the methodology when available (Supplementary Table S1). At high densities (below bubble closure

depth), the hydrostatic weighing technique is expected to be about 10 times more precise than simple volume and mass measurements (Gow, 1968) but rarely used, although it is important to correctly evaluate the fairly small density difference with pure ice density. We should note that the agreement between our model results and data is good at high densities for the three sites where hydrostatic weighing technique was used: Site 2 and D-47 (Supplementary Figure S6) as well as Byrd (Figure 4).

High-resolution measurements on small samples often aim at documenting the natural variability of density. Our model only simulates bulk density, and to illustrate a meaningful comparison, the highest resolution data (at DE08, B29, B32 and Dome C) were averaged over 0.25 m windows before being plotted. At some sites, a similar averaging was already performed before data publication (e.g. 1 m averaging at Byrd and Site 2, 0.5 m averaging at Mizuho). At a large number of sites, especially deep ice core drilling sites, measurements were performed on large volume samples. Still, it should be noted that at NEEM, although large volume samples were used, the data dispersion is higher than for Byrd (Figure 4) and part of the discrepancy between the model and data may be due to the uncertainty in the data.

For our study we have gathered density data covering the whole firn depth range, for which we had confidence in the data quality and the major site characteristics (temperature, accumulation). Although the effects of uncertainties on the data and natural density variability cannot be completely separated, we evaluate the data dispersion around the polynomial fit:

$$\sigma_{fit-data} = \sqrt{\left[\sum_{i=1}^{Nmax} \frac{\left(\rho_{fit}^i - \rho_{measured}^i\right)^2}{N_{max}}\right]} \tag{11}$$

where $N_{max}$ is the number of steps of data points, $\rho_{fit}$ represents the regression of the density profile and $\rho_{measured}$ the measured density averaged on a 0.25 m window. $\sigma_{fit-data}$ generally lies below 10.0 kg/m$^3$ (Figure 5).

In order to visualize the model data comparison with the different versions of the model on the 22 selected sites, we calculate the following deviation in parallel to the $\sigma_{fit-data}$ above (Equation 11):

$$\sigma_{model-fit} = \sqrt{\left[\sum_{i=1}^{Nmax} \frac{\left(\rho_{model}^i - \rho_{fit}^i\right)^2}{N_{max}}\right]}$$ (12)

Note that we compare here the model to the fit of the data and not directly to data because of the
strong site to site differences in the data (e.g. data resolution, sample size). Figure 5 and
Supplementary Table S1 display the $\sigma_{model-fit}$ for the 22 different sites before and after modifications
detailed in Section 2.

3.1.1. Data – model comparisons using the old model

Comparing our model results to density data is not trivial due to the diversity in measurement
techniques and samplings discussed above, as well as the natural variability in density that we do
not capture with a simplified model aiming at simulating very long time scales. A rough indication is
given by comparing $\sigma_{model-fit}$ and $\sigma_{fit-data}$. They are of the same order of magnitude although $\sigma_{fit-data}$ is
always lower than $\sigma_{model-fit}$ (Figure 5), confirming that the old model is likely not able to fully
represent the diversity of the density profiles at the 22 measurement sites.
The model-data agreement is variable among the different sites even for those with similar surface
climatic conditions. The temperatures and accumulation rates at Dome C and Vostok being similar,
model results at these sites are similar, but the density data have a clearly different shape. At
Vostok, a high densification rate is observed well above the critical density of about 550 kg/m$^3$. One
possible reason is the very different flow regimes of the two sites, one being at a Dome summit, and
the other on a flow line and subject to a horizontal tension (Lipenkov et al., 1989). This is not taken
into account in our simplified 1D model. Some density data at other sites also show no densification
rate change near the critical density, resulting in model-data mismatches (see Siple Dome, km 105,
km 200, Mizuho on Supplementary Figure S6).
The main disagreement between the old model and data is observed at the transition between the
first and the second densification stage with too high modeled densities and an associated slope
change in the density profile that is too strongly imprinted. This effect is due to a densification rate
that is too high in the first stage.

3.1.2. Data – model comparisons using the new model with only one activation energy

The modifications of the first densification stage described in Section 2.1 mainly reduce the slope
change at the transition between the Alley (1987) and Arzt (1982) mechanisms (not shown). It also
suppresses an instability of the previous model version which could fail to find a continuous
densification rate at the boundary between Alley (1987) and Arzt (1982) mechanisms.
However the new model still shows a tendency to overestimate the snow densification rate and
then underestimate the densification rate in the firn, as shown for NEEM and Vostok on Figure 4.
Still, looking at all different firn profiles, the general agreement between modeled and measured
firn density profiles is preserved. The agreement between measured and modeled firn density is
increased for some sites at (1) low accumulation rate and temperature in Antarctica (Dome A,
Vostok and Dome C but not South Pole) and at (2) relatively high temperature and accumulation
rate (Dye 3, Siple Dome, NEEM). In parallel, a larger disagreement between model and data is
observed for some other sites particularly in coastal Antarctica (DE08, Km 200, WAIS Divide). When
introducing these modifications for simulating $\delta^{15}$N evolutions over the last deglaciation, no
significant changes are observed with respect to simulations run with the old LGGE model. This is
not unexpected since most of the modifications concern the first stage of densification (top 10-15
m of the firn). The other modification concerns the LID definition, it only has a small impact on the
model results for the glacial-interglacial transitions and slightly increases the model – data mismatch
over deglaciations (Supplementary Figure S7).

3.1.3. Data-model comparisons using the new model with three activation energy and
implementation of impurity effect

The introduction of three different activation energies for different temperature ranges leads to
changes of the modeled density profiles at high densities (above about 800 kg/m$^3$). A clear
improvement is obtained for example at South Pole (Supplementary Figure S6), although the overall
impact of using three activation energies remains small.
The incorporation of the impurity effect following the Freitag et al. (2013) parameterization in our
model slightly deteriorates the model-data agreement because no specific re-adjustment of model
parameters was performed. However the model prediction of the density profiles remains correct
although the impurity effect parameterization was developed for a different purpose, i.e.,
simulating density layering (Freitag et al., 2013). This encouraged us to test this simple
parameterization in glacial climate conditions.

Overall, $\sigma_{model\text{-}fit}$ is only improved by 3% when using the modified model (3 activation energies and
implementation of impurity effect) instead of the former Goujon et al. (2003) mechanical scheme.
We thus conclude that the two versions of the model perform equally well.

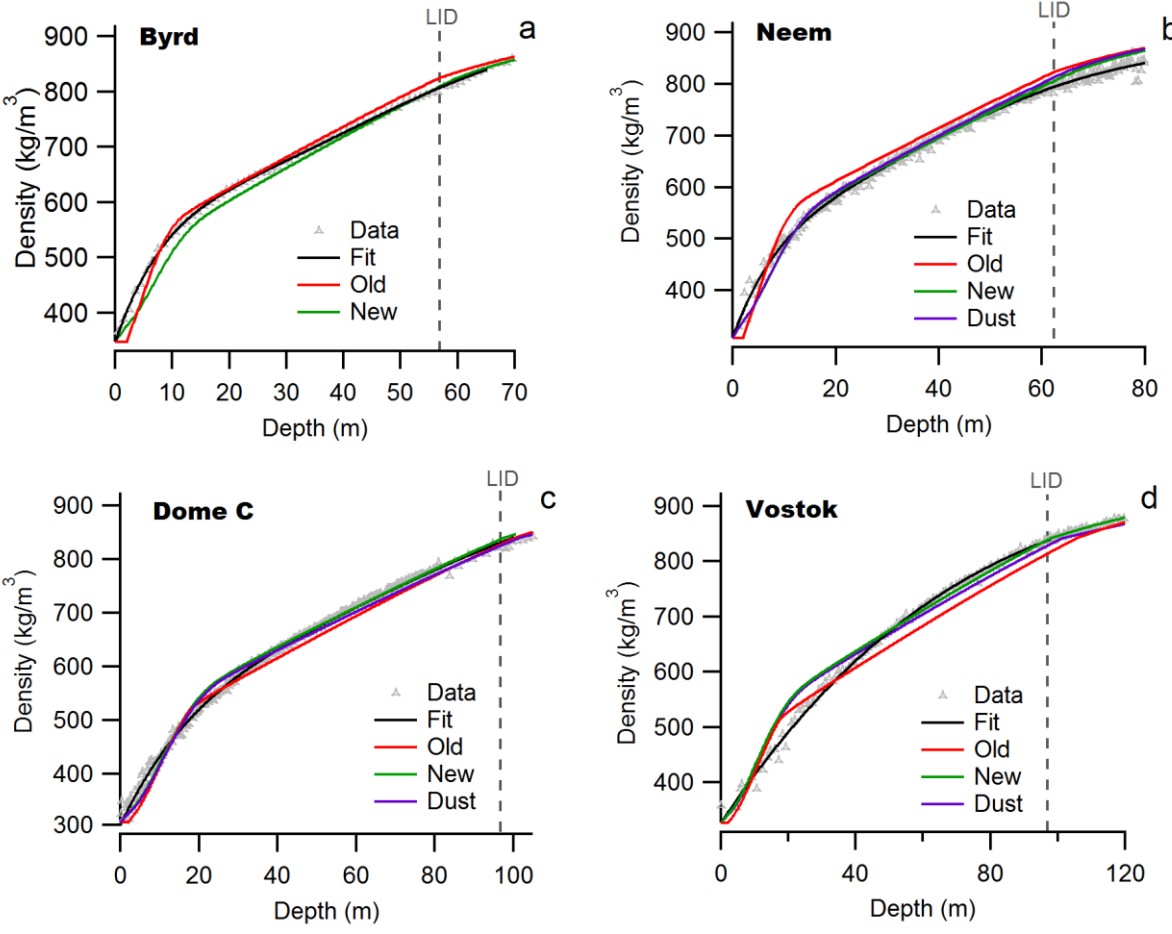

*Figure 4: Density profiles of Byrd (a), NEEM (b), Dome C (c) and Vostok (d). The grey triangles correspond to the data.*
*The black line corresponds to the polynomial fit, the red one to the old simulation, the green one to the new simulation*
*and the purple one to the new simulation with impurity effect.*

Finally, it should be noted that our main purpose is to improve the agreement between the
modelled LID and the evolution of $\delta^{15}N$ over deglaciations in Antarctica. Thus, in addition to the
above comparison of density profiles, we compared the depths at which the LID density, as defined
by Equation (10), is reached in the polynomial fit to the data and in the new model results. In the
old version of the model, the LID differences between the model and data range between -17.9 m
(at South Pole) and +8.6 m (at km 200) with a small mean value of -1.9 m and a standard deviation
of 6 m. In the new version, the LID differences between the model and data are comparable, ranging
between -14.1 m (at South Pole) and +12.8 m (at Talos Dome) with a small mean value of -0.7 m and
a standard deviation of 6 m. Similar results are obtained for Δage (see Supplementary Table S2): the
agreement with the data is similar for all model versions, and most cold sites are improved with the
new model. However the $\sigma_{model\text{-}fit}$ values remain high compared to the variability of the data ($\sigma_{fit\text{-}data,}$
black bars in Figure 5). We thus conclude from this section that the LGGE new firn densification
model preserves the good agreement between (1) modelled and measured firn density profiles and
(2) modelled and measured LID. We explore in the next section the performances of the new model
for coldest and driest conditions by looking at the modelled LID and hence $\delta^{15}$N evolution over
glacial – interglacial transitions.

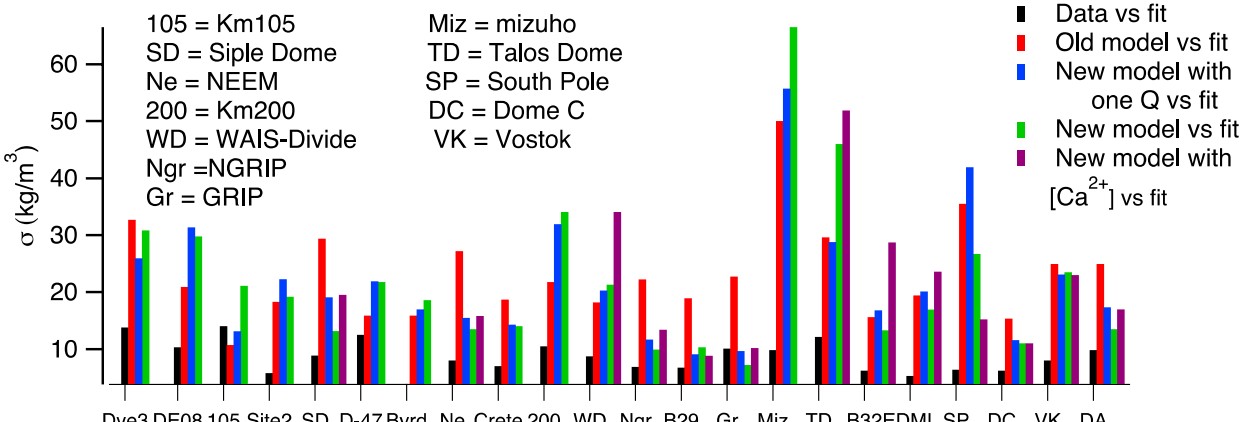


*Figure 5: Representation of the $\sigma_{fit\text{-}data}$ in black and the $\sigma_{model\text{-}fit}$ (in red for the old model, in blue for the model*
*with the new parameterization except the three activation energies, in green for the new model with three*
*activation energy and in purple for the new model with the impurity effect) at 22 Greenland and Antarctic*
*sites. The site characteristics are provided in Supplementary Table S1.*

3.2      $\delta^{15}$N glacial-interglacial profiles

In order to test the validity of the densification model in a transient mode, we model the time
evolution of $\delta^{15}$N over the last deglaciation, and compare it to measurements at 4 Antarctic and
Greenland deep ice-core sites: Dome C (cold and low accumulation site in Antarctica with a strong
mismatch observed between data and the old model), EDML (intermediate temperature and
accumulation rate in Antarctica with a significant mismatch between data and the old model), WAIS-
Divide (high temperature and accumulation rate site in Antarctica with a good model-data
agreement) and NGRIP (Greenland site with a good agreement between model and data) (Figure 3).
The computation of $\delta^{15}$N depends on the convective zone thickness, the LID and on the firn
temperature profile. The gravitational $\delta^{15}$N signal is indeed calculated from the LID and mean firn
temperature according to the barometric equation (Equation 1). The thermal $\delta^{15}N$ depends on the
temperature gradient between the surface and the LID. A small thermal signal exists in Antarctica
because of geothermal heat flux (with an average change of about 0.02 ‰ during deglaciation) but
no millennial variations are expected because the temperature variations are slow (<2°C/1000
years) compared to abrupt climate changes observed in Greenland (e.g. NGRIP).
The model calculates for each ice core depth the firn diffusive column height and thermal
fractionation at the bottom of the firn. To take into account the smoothing due to gas diffusion in
the open pores and progressive bubble close-off (Schwander et al., 1993), we smooth the $\delta^{15}N$
output with a log-normal distribution, of width $\Delta$age/5 and sigma=1 (Köhler et al., 2011; Orsi et al.,
2014). This formulation of the smoothing takes into account the variations of the gas-age
distribution with time. Note that it has been suggested that the width in Köhler et al. (2011) is too
wide (http://www.clim-past.net/7/473/2011/cp-7-473-2011-discussion.html). Still, using a smaller
width does not modify the modelled amplitude of the $\delta^{15}N$ signal over the deglaciation so that our
conclusions are not affected by such uncertainty.

3.2.1 Input scenarios

For the simulation of the $\delta^{15}N$ evolution over the last deglaciation, the firn densification model is
forced by a scenario of surface temperature and accumulation rate deduced from ice core data
(Supplementary Table S3). In Greenland (NGRIP, GISP2), the temperature is reconstructed using the
$\delta^{18}O_{ice}$ profiles together with indication from borehole temperature measurements (Dahl-Jensen,
1998) and $\delta^{15}N$ data for NGRIP (Kindler et al., 2014) for the quantitative amplitude of abrupt
temperature changes. Greenland accumulation rate is deduced from layer counting over the last
deglaciation (e.g. Rasmussen et al., 2006). The uncertainty in the temperature reconstructions can
be estimated to ± 3°C over the last deglaciation in Greenland (Buizert et al., 2014). As for the
Greenland accumulation rate, an uncertainty of 20% can be associated with the LGM value (Cuffey
and Clow, 1997; Guillevic et al., 2013; Kapsner et al., 1995). In Antarctica, both temperature and
accumulation rate are deduced from water isotopic records except for WAIS-Divide, where layer
counting back to the last glacial period is possible (Buizert et al., 2015). Temperature uncertainty for
the amplitude of the last deglaciation is estimated to -10% to +30% in Antarctica (Jouzel, 2003). The
reason for such asymmetry is mainly linked to outputs of atmospheric general circulation models
equipped with water isotopes. These models suggest that the present day spatial slope between
$\delta^{18}O$ and temperature most probably underestimates the amplitude of the temperature change

between glacial and interglacial period. We have used this estimate of asymmetric uncertainty on the amplitude of temperature change during deglaciation in our study. Recent studies have also suggested that the relationships between water isotopes and temperature and between water isotopes and accumulation rate can be applied with confidence in Antarctica for glacial temperature reconstruction (Cauquoin et al., 2015) while one should be cautious for interglacial temperature reconstruction with warmer conditions than today (Sime et al., 2009). Finally, a recent estimate of the deglacial temperature increase based on $\delta^{15}$N measurements at WAIS (Cuffey et al., 2016) led to a 11.3°C temperature increase over the last deglaciation (1°C warming to be attributed to change in elevation). This is larger than the temperature increase reconstructed in East Antarctica from water isotopes by 2-4°C and again not in favour of a "warm" LGM.

In the construction of the AICC2012 chronology (Bazin et al., 2013; Veres et al., 2013), the first order estimate of accumulation rate from water isotopes for EDML, Talos Dome, Vostok and Dome C has been modified by incorporating dating constraints or stratigraphic tie points between ice cores (Bazin et al., 2013; Veres et al., 2013). The modification of the accumulation rate profiles over the last deglaciation for these 4 sites is less than 20% and the uncertainty of accumulation rate generated by the DATICE model used to build AICC 2012 from background errors (thinning history, accumulation rate, LID) and chronological constraints is 30% for the LGM (Bazin et al., 2013; Frieler et al., 2015; Veres et al., 2013). Still, it should be noted that the uncertainty of 20% on LGM accumulation rate on central sites as given in the AICC2012 construction is probably overestimated. Indeed, deglaciation occurs around 500 m depth at Dome C, hence with small uncertainty on the thinning function and on the accumulation rate. These values are consistent with previous estimates of accumulation rate uncertainties over the last deglaciation (± 10% for Dome C (Parrenin et al., 2007) and ± 30% in EDML (Loulergue et al., 2007)).

We showed in Section 2.1 that surface density does not have a strong impact on the LID determination (Supplementary Figure S3). We do not have any indication of surface density in the past, so we impose a constant surface density of 0.35 for all sites at all times for transient runs. In order to convert the LID (deduced from density) to the diffusive column height measured by $\delta^{15}$N, we need an estimate of the convective zone in the past. We use a 2 m convective zone for all sites, except Vostok, where we use 13 m, in accordance with firn measurements (Bender et al., 2006). We assume that the convective zone did not evolve during the last deglaciation, consistently with dating constraints at Dome C and at Vostok during Termination 2 (Parrenin et al., 2012; Bazin et al., 2013; Veres et al., 2013; Landais et al., 2013).

692

### 3.2.2 Transient run with the old model

694

In this section, we focus on the $\delta^{15}$N evolution over the deglaciation at different Greenland and Antarctic sites as obtained from the data and as modelled with the old version of the LGGE model. This comparison serves as a prerequisite for the comparison with outputs of the revised model over the same period for the same polar sites. The comparison between the old LGGE model and $\delta^{15}$N data over the last deglaciation shows the same patterns already discussed in Capron et al. (2013). At Greenland sites, there is an excellent agreement between model and data showing both the decrease in the mean $\delta^{15}$N level between the LGM and the Holocene and the ~0.1 ‰ peaks in $\delta^{15}$N associated with the abrupt temperature changes (end of the Younger Dryas, Bølling-Allerød, Dansgaard-Oeschger 2, 3 and 4, Figure 6 and Supplementary Figure S8). On the other hand, the modelled and measured $\delta^{15}$N over the last deglaciation show significant dissimilarities in Antarctic $\delta^{15}$N profiles displayed on Figure 6 and Supplementary Figure S8, except at the relatively high accumulation rate and temperature site of WAIS-Divide where the model simulates properly the $\delta^{15}$N evolution in response to the change in accumulation and mean firn temperature estimated from water isotopic records and borehole temperature constraints (Buizert et al., 2015). Note that in Buizert et al. (2015), the modelled $\delta^{15}$N was obtained from the Herron and Langway model. For the other Antarctic sites (Figure 6), we observe that model and data disagree on the $\delta^{15}$N difference between the LGM and Holocene levels. At EDML, Dome C and Vostok, the model predicts a larger LID during the LGM, while $\delta^{15}$N suggests a smaller LID compared to the Holocene (with the assumption of no change in convective zone during the deglaciation). In addition, the measured $\delta^{15}$N profiles at Berkner Island, Dome C, EDML and Talos Dome display an additional short term variability, i.e. $\delta^{15}$N variations of 0.05‰ in a few centuries during stable climatic periods. These variations can be explained by the ice quality (coexistence of bubbles and clathrates) at Dome C and EDML. Indeed, for pure clathrate ice from these two sites, such short term variability is not observed (e.g. Termination 2 at Dome C, Landais et al., 2013). At Berkner Island and Talos Dome, these variations cannot be explained by the quality of the measurements, by thermal effects nor by dust influence. They are also not present in the accumulation rate and temperature forcing scenarios deduced from water isotopes (Capron et al., 2013). In the absence of alternative explanations, we can thus question the existence and variations of a convective zone and/or the accuracy of the reconstruction of past accumulation rate and temperature scenarios from water isotopes in Antarctica except at WAIS-Divide where layer counting is possible over the last deglaciation. We

thus explore further the influence of accumulation rate and temperature uncertainties on the $\delta^{15}$N
modelling.

The uncertainties in the changes of temperature and accumulation rates over the deglaciation
significantly influences the simulated $\delta^{15}$N, as already shown in previous studies and this sensitivity
of $\delta^{15}$N has even been used to adjust temperature and/or accumulation rate scenarios (Buizert et
al., 2013; Guillevic et al., 2013; Kindler et al., 2014; Landais et al., 2006). We tested the influence of
the accumulation rate and temperature scenarios on the simulated $\delta^{15}$N profiles for the last
deglaciation, but even with large uncertainties in the input scenarios, it is not possible to reproduce
the measured Antarctic $\delta^{15}$N increase at Dome C and EDML with the old version of the LGGE model.

This result is illustrated on Figure 7 where we display a comparison between the amplitude of the
measured $\delta^{15}$N change and the amplitude of the modelled $\delta^{15}$N change with the Goujon version
over the last deglaciation. For this comparison, we calculated the Last Glacial Maximum (LGM) $\delta^{15}$N
average over the period 18-23 ka and the Early Holocene (EH) $\delta^{15}$N average over the period 6-10 ka
(or smaller, depending on available data, cf blue boxes on Figure 6). We estimated the uncertainty
in the measured $\delta^{15}$N change by calculating first the standard deviation of the $\delta^{15}$N data over each
of the two periods, LGM and EH as $\sigma_{15N\_data\_EH}$ and $\sigma_{15N\_data\_LGM}$ and then the resulting uncertainty
in the $\delta^{15}$N change as: $\sigma_{15N\_EH-LGM} = \sqrt{\sigma^2_{15N\_data\_EH} + \sigma^2_{15N\_data\_LGM}}$

As for the modelled $\delta^{15}$N change, associated error bars are deduced from the uncertainty in the
temperature and accumulation input scenarios (shown on Supplementary Figure S9 for the
improved model). The total error bar hence shows the difference between most extreme
accumulation rate or temperature input scenarios. In these sensitivity tests, we assumed that it is
not possible to have an underestimation of the temperature change while at the same time have an
overestimation of the accumulation rate (or the opposite) because changes in accumulation rate
and temperature are linked, at least qualitatively when comparing LGM and Holocene mean values.

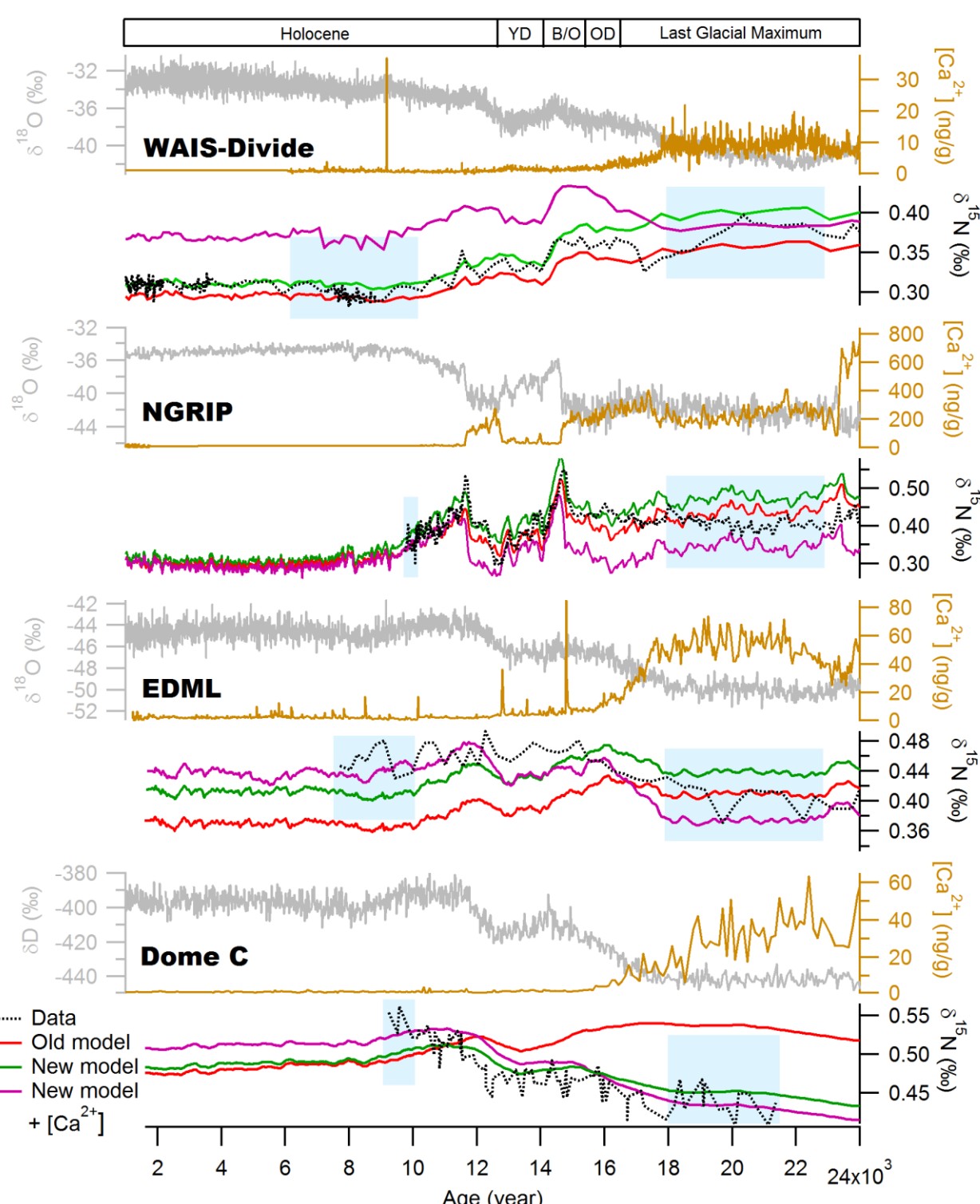


*Figure 6**: Comparison of the measured δ¹⁸O or δD (grey), the calcium concentration (gold), the measured δ¹⁵N (black)*
*and the modelled δ¹⁵N (old (red), new version (green) and new version with impurity (purple)) of the LGGE model for*
*WAIS-Divide, NGRIP, EDML and Dome C. Blue boxes for each sites indicate the periods over which the δ¹⁵N average for*
*the LGM and EH have been estimated for the calculation of the amplitude of the δ¹⁵N change over the deglaciation.*

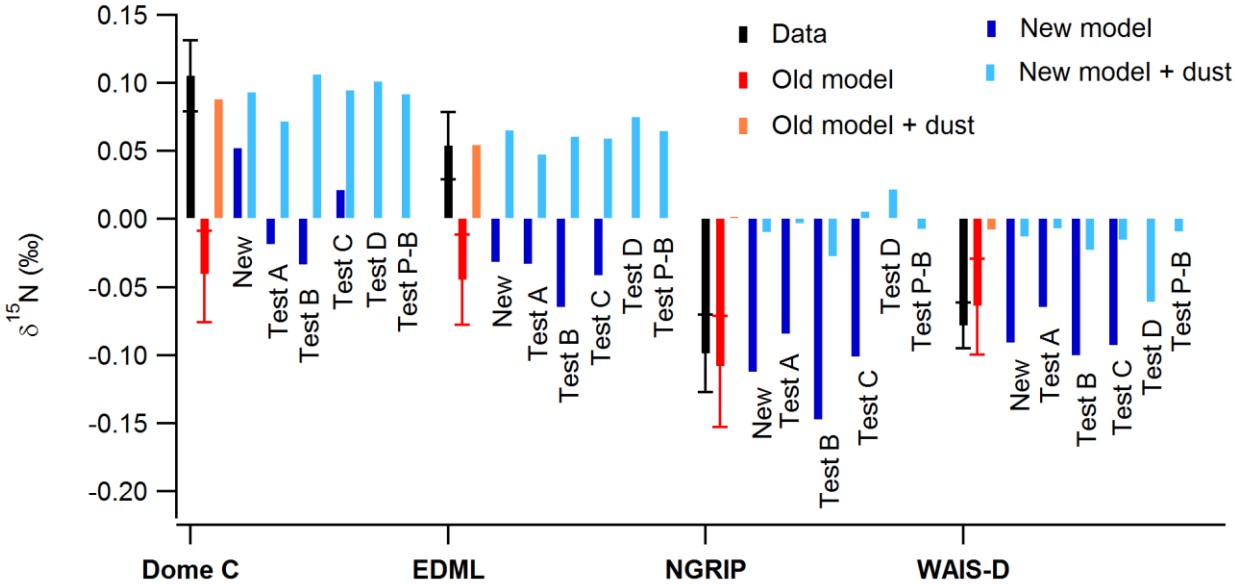


*Figure 7: Difference between EH and LGM δ<sup>15</sup>N at 4 different polar sites (raw data are given in Supplementary Table S4).*
*The measured δ<sup>15</sup>N difference is shown by a black bar (data). The modelled δ<sup>15</sup>N difference is shown with colours: old*
*version in red (orange with the impurity influence), new version in blue with different parameterizations. "New"*
*corresponds to the parameterization of Table 1. Parameterizations for sensitivity tests A, B, C and D are given in Table*
*3. When "+ dust" is mentioned, it corresponds to the addition of the impurity influence as parameterized by Freitag et*
*al., (2013) (Equations 8 and 9). Test Pimienta-Barnola (P-B) corresponds to a test with implementation of the impurity*
*effect in the "New" parameterization following the Freitag parameterization adapted to the Pimienta-Barnola model*
*instead of the Herron and Langway model used for the other sensitivity tests. We display the modelled error bars only*
*on the old model outputs (red) but the same uncertainty can be applied to all model outputs (New, Tests A, B, C, D and*
*P-B) at each site.*

### 3.2.3 Results with updated temperature parameterization

By construction, the new LGGE firn model with the temperature dependency of the firn densification
module depicted on Section 2.2.1 is expected to improve the agreement between model and data
for cold sites of East Antarctica over the last deglaciation by increasing densification rates at low
temperature. This new parameterization modifies the densification rate through the creep
parameter given in Equation (7). Figure 8 shows the evolution of the creep parameter with
temperature for different choices of the three activation energies $Q_1$, $Q_2$ and $Q_3$. Compared to the
old model, the densification rate is higher at low temperature, below -55°C (i.e. for LGM at Dome C
and Vostok, Table 1). At higher temperature (between -55°C and -28°C corresponding to present-
day temperature in most polar sites), the creep parameter is slightly lower than in the old model.
The difference between the 2 curves is however not large so that densification rate is not strongly
modified over this range. This is in agreement with comparable firn density profiles obtained for the
different polar sites using the old or the improved LGGE model (Section 3.1, Figure 4).
In the improved model, the simulated profiles of $\delta^{15}$N are comparable to $\delta^{15}$N simulated with the
old model at the sites that were already showing a good agreement between the old model outputs
and data, for example NGRIP, GISP-2, Talos Dome and WAIS-Divide (Figure 6 and Supplementary
Figure S8). This is expected since the corresponding densification rate is only slightly reduced in the
temperature range of -55°C/-28°C which corresponds to the temperature range encompassed over
the last deglaciation at these sites. This results in a deeper LID and hence higher $\delta^{15}$N level, which is
in general compatible with the data (except at Talos Dome). Some differences are also observed for
the timing of the $\delta^{15}$N peaks for Bølling-Allerød and end of Younger Dryas at NGRIP when using the
different model versions reflecting variations in the simulated $\Delta$age (cf Supplementary Table S5);
the general agreement with the measured profile is preserved with even a slight improvement of
the modelled $\Delta$age with $\delta^{15}$N constraints with the modified model. At the coldest sites (Dome C,
Vostok), the agreement between data and modelled profiles is largely improved with a modelled
LGM $\delta^{15}$N smaller than the modelled EH $\delta^{15}$N, but a perfect match cannot be found. At the
intermediate EDML site, it is not possible to reproduce the sign of the slope during the deglaciation.


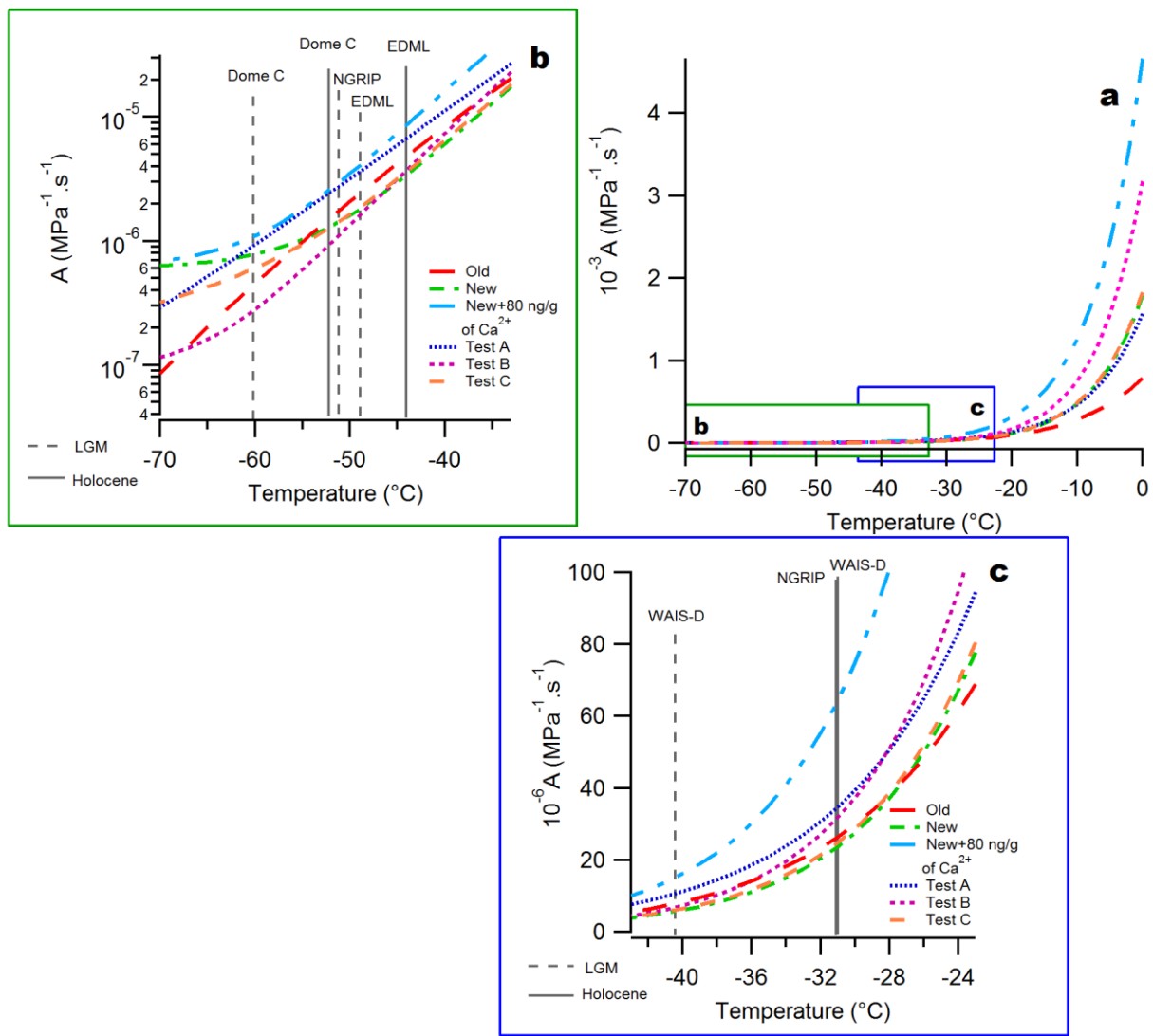


*Figure 8: Dependence of the creep parameter (Equation 7) as a function of temperature for 6 different*
*parameterizations. "Old" corresponds to the Goujon et al. (2003) version of the model; "New" corresponds to*
*the improved LGGE model with parameterization described in Table 1; "New + 80 ng/g of $Ca^{2+}$" corresponds*
*to the parameterization of Table 1 with the addition of the impurity effect following Equation (8) and a $[Ca^{2+}]$*
*value of 80 ng/g; Tests A, B and C are sensitivity tests run with the values presented on Table 3. Figure 8a*
*shows the creep parameter evolution for the whole temperature range, Figure 8b is a focus at very low*
*temperature and Figure 8c is a focus at intermediate temperature. The grey vertical lines indicates the*
*temperature for Early Holocene (EH, solid line) and LGM (dotted line) at the 4 study sites presented in Figures*
*6 and 7.*

| Test | Activation energy (J/mol) | Coefficient |
|---|---|---|
| | $Q_1 = 90000$ | $a_1 = 5.5*10^5$ |
| Test A | $Q_2 = 60000$ | $a_2 = 1.0$ |
| | $Q_3 = 30000$ | $a_3 = 4.5*10^{-8}$ |
| Test B | $Q_1 = 110000$ | $a_1 = 5.5*10^9$ |
| | $Q_2 = 75000$ | $a_2 = 1950.0$ |

| | $Q_3 = 1500$ | $a_3 = 9.0 \times 10^{-16}$ |
|---|---|---|
| | $Q_1 = 110000$ | $a_1 = 1.05 \times 10^9$ |
| Test C | $Q_2 = 75000$ | $a_2 = 1400$ |
| | $Q_3 = 15000$ | $a_3 = 8.7 \times 10^{-12}$ |
| | $Q1 = 110000$ | $a1 = 1.05 \times 10^9$ |
| Test D | $Q2 = 75000$ | $a2 = 980$ |
| | $Q3 = 1230$ | $a3 = 3.6 \times 10^{-15}$ |


*Table 3: Values used for the different sensitivity tests for three activation energies. These values have been chosen to illustrate the effects of varying activation energy for the different temperature ranges on the densification rate for the different ice core deep drilling sites (cf figure 8) and support the tuning presented on Table 1.*


In order to more quantitatively assess the robustness of the proposed parameterization in Table 1,
we confront in Figure 7 the measured and modelled $\delta^{15}N$ differences between the LGM and EH at
the 4 Greenland and Antarctic sites selected in Figure 7 above. For this comparison, we use not only
the parameterization of Table 1 but also sensitivity tests performed with different
parameterizations of the temperature dependency of activation energy and impurity effects (details
on Table 3).
When using the parameterization of Table 1 ("new model"), Figure 7 shows strong improvement of
the simulation of the $\delta^{15}N$ difference between EH and LGM at Vostok and Dome C. Indeed, the
modelled EH-LGM difference now has the correct sign at very cold sites of East Antarctica (Figure 7)
when compared with $\delta^{15}N$ measurements.
We present some sensitivity tests to illustrate the choice of our final parameterization (i.e. the new
model) through influences on the creep parameters and LGM vs EH $\delta^{15}N$ changes. As displayed in
Figure 8, test A has a higher creep parameter than the old model throughout the whole temperature
range. Compared to the output of the old model, the LGM vs EH $\delta^{15}N$ change simulated with test A
is slightly higher but the sign of the $\delta^{15}N$ change over the last deglaciation is still wrong at Dome C
and EDML. This test shows that it is not the mean value of the creep parameter that needs to be
changed, but the dependency to temperature. Test B has a higher creep parameter above -35°C,
but a lower creep parameter than the old model below -35°C, which starts flattening and hence
reaching values higher than the old model creep parameter below -65°C. The LGM vs EH $\delta^{15}N$ change
simulated with test B is still comparable with data at WAIS-Divide. However, the model – data
comparison deteriorates at NGRIP and EDML compared to the model-data comparison with the old
version of the model. Moreover, it does not solve the model – data mismatch at Dome C. This shows
that the change in the creep parameter at intermediate temperature is too steep. Strong differences

occur at high temperature (above -30°C) but it does not affect the modelled $\delta^{15}$N change between LGM and EH for our 4 sites. On the contrary, the slightly lower creep parameter at low temperature leads to a worse agreement between model and data for the Dome C deglaciation than when using the "new model". Test C has been designed so that the activation energy at low temperature corresponds to estimates of activation energy for ice surface diffusion (Jung et al., 2004; Nie et al., 2009), a mechanism that is expected to be important at low temperature (Ashby, 1974). Using such a parameterization leads to a fair agreement between the modelled and the measured $\delta^{15}$N change over the last deglaciation for the different sites. At Dome C, the correct sign for the $\delta^{15}$N evolution between LGM and the Holocene is predicted by the model. However, the modelled $\delta^{15}$N increase is still too small compared to the data and the $\delta^{15}$N calculated by the "new model". This is probably due to a too high creep parameter at low temperature.

Summarizing, the best agreement between data and model for Dome C is obtained for the parameters given on Table 1: the creep parameter of "new model" flattens below -50°C and is thus not very different for the LGM or the EH at Dome C. As a result, the modelled LID and hence $\delta^{15}$N are less sensitive to temperature, and the sign of the EH-LGM difference can be inverted, and brought closer to the observations. It should be noted that despite many sensitivity tests we could not find a parameterization able to reproduce the EH-LGM $\delta^{15}$N changes for all 4 sites. In the "new model" without impurity effect, it is not possible to reproduce the measured EDML $\delta^{15}$N change over the last deglaciation even when taking into account the uncertainty in the input parameters (temperature and accumulation rate, Supplementary Figure S9).

### 3.2.4   Impurity softening

The dust content in LGM ice is much larger than in Holocene ice (Figure 6), and impurity inclusions in ice have an impact on the grain structure, allowing it to deform more easily (Alley, 1987; Fujita et al., 2014). We incorporated dust softening using the parameterization of Freitag et al (2013) as detailed in Section 2.2.2. We compared two expressions for the impurity softening (tuned to be applied to the Herron and Langway model, or Pimienta and Barnola model), but found that the differences between the two parameterisations were minor (Figure 7). We use the Herron and Langway parameters in the following.

Figure 8 shows the effect of impurities on the creep parameter: densification is enhanced over the whole temperature range. At all sites, incorporating impurity softening reduces the firn thickness

during periods characterized by high impurity concentration in the ice (LGM). It thus leads to an
increase of the EH-LGM LID difference (Figure 7).

This effect clearly helps to bring in agreement modelled and measured $\delta^{15}N$ at Dome C, Vostok and
EDML (Figures 6, 7 and Supplementary Figure S8). The improvement through dust softening is
particularly important at EDML where the change of activation energy had only a modest effect. For
the 3 sites mentioned above, the model incorporating the parameterization of activation energy
depicted in Table 1 and the impurity effects is able to reproduce the $\delta^{15}N$ increase over the last
deglaciation. Note that short-lived peaks in impurities, likely triggered by volcanic events, have no
visible effect on bulk firn thickness (Figure 6). Contrary to the improved situation in cold Antarctic
sites, we observe that, at the warmer sites like NGRIP and WAIS-Divide, incorporating impurity
softening deteriorates the model data fit, which was already good in the older version of the model,
and also good with other firn densification models (Kindler et al, 2014; Buizert et al, 2015). It
produces almost no change in firn thickness between the LGM and the EH at NGRIP, which
contradicts $\delta^{15}N$ observations. The same mismatch is observed at WAIS-Divide using a different
model, as already noted by Buizert et al. (2015). We tested the sensitivity to the dust
parameterization by implementing the Freitag parameterization adapted to the Pimienta-Barnola
model instead of the parameters for the Herron and Langway model used with our improved model
(cf Section 2.2.2). The two different parameterizations of the impurity effect lead to very
comparable LGM to EH $\delta^{15}N$ changes over the last deglaciation on the 4 sites discussed here.
The model – data mismatch observed when incorporating the dust effect may be partially due to
the fact that we did not readjust $a_i$ and $Q_i$ after implementation of the impurity effect. To explore
this possibility, sensitivity test D has been designed with a re-parameterization of the $a_i$ and $Q_i$ values
after implementation of the impurity effect. To do so, we calculated the optimal creep parameter A
for each mean EH and LGM condition at each site, and adjusted sequentially $a_3$, $a_2$, $a_1$, $Q_3$, $Q_2$, and
$Q_1$ to minimize the model-data mismatch. Only $a_3$, $a_2$ and $Q_3$ needed adjustments, and their values
can be found in Table 3. We did not perform the adjustment on modern density profiles, because
these are only weakly sensitive to the dust parameterization, $Ca^{2+}$ concentrations being low.
Impurity concentration is very high at NGRIP during the glacial period. As a consequence, even if our
new parameterization of $a_i$ and $Q_i$ (new model) properly reproduces the Greenland $\delta^{15}N$ level at the
LGM, this glacial modelled Greenland $\delta^{15}N$ level is too low when including the impurity effect. The
re-parameterization of $a_i$ and $Q_i$, proposed as sensitivity test D, enables an improvement of the
agreement between model and data for glacial $\delta^{15}$N at WAIS-Divide, maintain the results at Dome-
C and EDML, but can still not produce reasonable results at NGRIP (Figure 7).

The mismatch observed for the $\delta^{15}$N simulations at WAIS-Divide and NGRIP when incorporating the
impurity effect suggests that the parameterization presented in Equations (8) and (9) is not
appropriate to be used on bulk [$Ca^{2+}$] concentration and/or for LGM simulation. Actually, the
proposed parameterization by Freitag et al. (2013) was tuned to density variability in present-day
firn, and may not be valid for LGM when [$Ca^{2+}$] concentrations were 10-100 times larger than
present-day. It is also possible that the dust effect saturates at high concentration, and is no longer
sensitive above a certain threshold. To further improve the model – data agreement with the dust
parameterization, a possibility is to add simple thresholds on a minimum and maximum effect of
calcium as proposed in supplementary material (Supplementary Text S2 and Figure S10).
Implementing threshold values on calcium reduces the largest inconsistencies between model
results and $\delta^{15}$N data, in particular at NGRIP (through the threshold at high calcium concentration)
and at WAIS (through the threshold at low calcium concentration).

It is also possible that the impurity influence, like temperature, acts differently depending on the
dominant mechanism for firn deformation, and that the impurity effect is more important at colder
temperature. The mechanisms by which impurities influence firn deformation are still poorly
understood. Dust particles do not always influence densification in the same way: dissolved particles
soften firn and ice while the softening or hardening effect of non-dissolved impurities is less clear
(Fujita et al., 2016; Alley et al., 1987). More work is thus needed before the correct "impurity effect"
component and the mechanisms by which it acts on densification are identified (e.g. Fujita et al.,
2014, 2016). Here, we have shown that a simple parameterization as a function of [$Ca^{2+}$]
concentration does not provide uniformly good results, and seems only suitable for sites on the
Antarctic Plateau.

To sum up, the new parameterization of the creep parameter has been designed to preserve good
agreement between the old model outputs and data at sites that were already well simulated
(WAIS-Divide, NGRIP, Talos Dome). In addition, this parameterization improves the simulation of
the deglaciation at cold Antarctic Sites (Dome C, Vostok). However, the EH-LGM $\delta^{15}$N change at
Dome C and EDML cannot be reproduced using only the temperature dependency of activation
energy. The inclusion of impurity effect following the Freitag parameterization improves the
situation for cold sites but leads to inconsistent $\delta^{15}$N evolutions over the deglaciation at WAIS-Divide
and NGRIP unless threshold effects are implemented.

## 4. Conclusion and perspectives

In this study, we have presented a revision of the LGGE firn densification model. We have
summarized the parameterization choices of this firn model that would explain a large part of the
disagreement between modelled and measured $\delta^{15}$N evolution over the last deglaciation for
extremely cold sites of East Antarctica. Based on analogy with ceramic sintering at hot temperature
and recent observations of the impurity effect on firn density, we have improved the LGGE
densification model by incorporating new parameterizations for the evolution of the creep
parameter with temperature and impurity contents within the firn densification module. We follow
previous studies evidencing different dominant firn sintering mechanisms for different temperature
ranges that support a temperature dependency of the creep activation energy. We showed that
these new parameterizations improve the agreement between model and data at low temperature
(below -30°C), and retain the good agreement at warmer temperature. In particular, the improved
LGGE firn density model is now able to reproduce the $\delta^{15}$N increase over deglaciations at cold sites
such as Dome C and Vostok.

The new parameterization implies a more rapid firn densification at lower temperature and high
impurity load than in classical firnification models. This result obtained with our associated
appropriate parameterization is in agreement with the study of Parrenin et al. (2012) showing that
the classical firn densification model overestimates LID during the last glacial period at EDC. With
our revised model, the simulated $\Delta$age is also significantly decreased for the glacial periods at low
accumulation and temperature sites of the East Antarctic plateau (Dome C, Vostok and Dome Fuji).
This has important consequences for building air vs ice timescales in Antarctica and hence for the
studies of the relationships between temporal evolutions of atmospheric composition vs. Antarctic
temperature. At EDC 21 ka (ice age), the modelled $\Delta$age decreases from 4840 years (old model) to
4270 years (new model) or 4200 years (new model including impurity effect). At Vostok 21 ka (ice
age), the modelled $\Delta$age decreases from 5630 years (old model) to 5030 years (new model) or 4900
years (new model including impurity effect). The latest results are in good agreement with the
recent determination of $\Delta$age within the AICC2012 timescale: 3920 years for EDC 21 ka (ice age) and
5100 years for Vostok 21 ka (ice age). This is not unexpected since the EDC LID in the construction
of the AICC2012 timescale is deduced from the EDC $\delta^{15}$N scenario, a hypothesis supported by the
available gas and ice stratigraphic markers over the last deglaciation (Parrenin et al., 2012).

Our finding is, however, associated with several limitations so that this new model does not propose
a definite re-evaluation of the formulation of the activation energy but proposes some ways to be
further tested and explored to improve firn densification models especially for applications in
paleoclimate reconstructions. Our approach remains empirical and we could not identify separately
the different mechanisms involved. The problem of a $\delta^{15}$N data-model mismatch in low temperature
and accumulation rate sites of East Antarctica is thus not definitively solved. Still, we showed that
revising the temperature and impurity dependence of firn densification rate can potentially strongly
reduce the $\delta^{15}$N data-model mismatch and proposed preliminary parameterizations easy to
implement in any firn densification model.
Finally, the new parameterization proposed here calls for further studies. First, laboratory or field
studies of firn densification at very cold controlled conditions are needed to check the
predominance of one mechanism over another at low temperature such as the predominance of
the boundary diffusion over grain boundary mechanism around -60°C; this is a real challenge
because of the slow speed of deformation. Second, we have suggested that the current
parameterization of impurity on firn softening should be revised, especially for very high impurity
load (Greenland) using for example thresholds on impurity concentrations. Third, the separate
effects of impurities and temperature on firn densification and hence $\delta^{15}$N evolution should be
tested on periods other than the last deglaciation. Sequences of events associated with non-
synchronous changes in surface temperature, accumulation rate and impurity content would be
particularly valuable for this objective. Finally, additional constraints on the firn modelling can also
be obtained through the use of cross-dating on new ice core with high resolution signals as already
used by Parrenin et al. (2012).

*Acknowledgements:* We thank Anders Svensson, Rob Arthern, Hans Christian Steen-Larsen and Xiao
Cunde for data sharing and Sarah Guilbaud for her work during her final internship study. Thanks to
Pierre Badel, Maurine Montagnat and Christophe Martin for insightful discussions about
densification mechanisms. Thanks to Myriam Guillevic for her work on the densification model and
helpful discussions. This work is supported by INSU/CNRS LEFE project NEVE-CLIMAT and the ERC
COMBINISO 306045.

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
