# Peer review of "Modelling the firn thickness evolution during the last deglaciation: constraints on sensitivity to temperature and impurities"

_Climate of the Past, 2016_

## Referee Comment (RC1) · Anonymous Referee #1 · 9 Nov 2016

Bréant et al. address an important outstanding problem in ice core research, namely the model-data mismatch of $\delta^{15}$N-N$_2$ as a proxy for firn thickness during the last deglaciation. They offer an interesting new solution to this problem, by proposing a temperature-dependent effective activation energy for firn sintering. In their framework, this can be understood as the effect of three separate firn densification mechanisms working in parallel, each with its own activation energy. Their modified firn densification model provides an improved fit to the deglacial $\delta^{15}$N evolution at cold interior sites, while still being able to fit relatively warm sites that were already modeled well by existing models.

I would ask the authors to consider the following points in a revised manuscript:

[Figure]

- The $\delta^{15}$N model-data mismatch has a long history in ice core research, and is described most clearly by Landais et al. 2006. Several solutions have been proposed for this problem. Without explicitly stating so, the present manuscript takes as the starting assumption that the temperature sensitivity of the densification model must be the problem, due to the absence of a modern analog. I'll refer to this as the "no-analog solution" to the LGM $\delta^{15}$N problem.

  I think it would be important to introduce the LGM $\delta^{15}$N problem better, and outline some of the other proposed solutions. For example, Landais et al. (2006) concluded that reconstruction of past accumulation rates was the most likely solution. Why was that explanation abandoned in favor of the no-analog explanation?

  It is unclear to me what the main objective is of the present paper. Is the purpose to simply test whether the LGM $\delta^{15}$N problem can be solved using a different activation energy scheme? Or is the purpose to present a new model that will replace the Goujon model in future research at LGGE? Both models fit present-day data equally well, so whether the new model is an improvement relies solely on whether you believe the no-analog solution to be the correct one.

  Finally, did they solve the problem? From the conclusion section it is not exactly clear whether or not the no-analog and dust mechanisms fully solve the LGM $\delta^{15}$N problem. It seems like the dust mechanism is insufficient by itself, given that it makes sites the fit to sites like GISP2, NGRIP and WAIS Divide worse. The no-analog solution seems to do a better job, yet it requires an unknown process with very low activation energy (see below). Moreover, EDML remains confusing to me. It's warm enough during the LGM to have modern analog sites, yet it does show the $\delta^{15}$N model-data mismatch in traditional firn models. I would appreciate some added discussion on whether the LGM $\delta^{15}$N problem has now been solved satisfactorily, and whether we can forget about other proposed solutions.

- To get the densification rate to increase meaningfully at low temperatures, the authors have to introduce a densification process with an extremely low activation

energy of Q3=1.5 kJ/mol (low enough to be essentially temperature-insensitive). They suggest this process to be surface diffusion. However, experimental studies suggest the activation energy for ice surface diffusion is on the order of 14 to 38 kJ/mol (e.g. Jung et al., doi: 10.1063/1.1770518, Nie et al., doi: 10.1103/Phys-RevLett.102.136101, and references therein). The value used by Bréant et al. seems an order of magnitude too small to be surface diffusion. Therefore, they are essentially invoking densification by an unknown process with very small Q.

The authors should acknowledge that the values they use for Q3 seems unrealistically low. In my view, this is an important piece of evidence that the "no-analog assumption" by itself may be insufficient to solve the LGM $\delta^{15}$N problem – the authors may not share this view.

At the other end, their high-Q process (suggested to be vapor diffusion) has a value that seems too high at Q1=110 kJ/mol. Vapor diffusion scales with the vapor pressure, and the enthalpy of sublimation in ice is only 51 kJ/mol.

- Ultimately the goal of firn modelling is to predict $\Delta$age, and therefore I was surprised that no $\Delta$age results are shown. How does the new activation energy scheme change the simulated $\Delta$age? In Greenland we have direct constraints on $\Delta$age via the thermal $\delta^{15}$N signals. How well does the model capture those?

Likewise, in evaluating the model performance on page 18, the authors test only how well the model predicts the LID (i.e. $\delta^{15}$N). The more important metric, in my mind, is how well the model predicts $\Delta$age. This can be evaluated via the integrated density from the surface to the LID (because $\Delta$age is essentially the mass of overlying snow divided by the accumulation rate).

- On lines 624-625 the authors conclude that uncertainties in the climate forcing cannot explain the LGM $\delta^{15}$N problem. However, to solve the $\delta^{15}$N problem one would need to make the LGM warmer, not colder! For some reason the temperature uncertainties in Fig. S9 are applied very asymmetrically, such that the LGM

is always very cold.

- It is not very clear how the parameters in Table 1 were selected. How was the model calibrated? Did the authors minimize some cost function? The authors do give three representative examples in Table 3, but no criteria for choosing the best model.

  In their preferred model (Table 1), process 1 (Q1 = 110 kJ/mol) doesn't do much. At all relevant temperatures, process 2 is at least an order of magnitude larger.

- The authors claim that the new model provides a better fit to modern data than the old model. The LID prediction improves by $1.2 \pm 6$ meter. That hardly seems like a statistically significant improvement. Using as Student's t-test it should be trivial to show whether the null hypothesis (both models perform equally well) can be rejected.

  As mentioned earlier, I would encourage the authors to not only compare the predicted LID to the fitted data, but also to compare the integrated density in the simulations to the integrated density of the fit. The latter metric is more representative of $\Delta$age.

- The closest analog to LGM conditions in East Antarctica is the Dome A site, with mean annual temperatures below $58°$ C. The old LGGE model provided a reasonable fit to density at this site (Cunde et al., 2008). Why is this site not included in the calibration data set?

- The MS does not give many technical details about the running of the models. What time and spatial step size are used? What is the lower model boundary? What geothermal heat flux is used? The latter is important in the stagnant firn columns of the LGM.

- The authors test the dust softening hypothesis of Johannes Freitag and Maria Hörhold. It is important to note that this model of Freitag et al (2013) was designed to simulate layering, rather than bulk density as it is used here. How were Ca data averaged in the model runs? How does the Ca data resolution compare to the model resolution?

Please discuss some of the caveats regarding dust. From talking to Johannes Freitag, I get the impression he believes layering to be more relevant than bulk density in this regard.

What do the authors recommend? Should future users incorporate dust or not?

- I do not understand the rationale behind the LID parameterization of Equation 10. The authors use a very complicated way to define the LID, namely the depth where the modelled $\delta^{15}$N in the open pores matches the $\delta^{15}$N in the mature ice. This approach involves simulating the bubble trapping process, which is very (!!) poorly understood. The depth range of bubble trapping is completely unknown at most sites, unless measurements of closed porosity are available (which is not the case at most sites). Also, trapping depends strongly on density layering, as early work at Law Dome and more recent work by Rachael Rhodes and others have shown.

The lock-in depth is very clearly visible in firn air sampling data, as an abrupt change in the concentration slope of many tracers. Why not use this commonly-used and simple metric, which can be directly derived from data? I fear that modelling something as complicated as bubble trapping could easily lead to errors. How does the fit of Figure S5 look when using the common definition of the lock-in depth?

- One of the important achievements of this work is to compile a large database of reliable firn density measurements. This would be an extremely valuable resource to the firn research community if it were publicly available in a format that is easy to use. I would like to kindly ask the authors to make this database publicly available as a supplement to the manuscript, as is also strongly encouraged

by the data policy of the journal (Climate of the Past).

The manuscript does not have a statement of data availability yet, which will need to be added as per the editorial guidelines of CP.

- throughout the paper the authors refer to "snow" as everything above the critical density, and "firn" as everything below. This is not common usage, and should be specified.

Some minor comments and typos:

L2: "constrains" should be "constraints"

L23: "to" should be "on"

L24: "existence". Maybe "dominance" would be better?

L35: "depict". What about "reconstruct"?

L61: The close relationship between A and T seems to be mostly an assumption. See e.g. Monnin (2004), Fudge (2016) or Van Ommen (2004).

L65: in HL the "change" in pore space is proportional to the increase in weight.

L89: Better write: "In the absence of thermal gradients", the $\delta^{15}$N trapped ..... The geothermal heat flux matters also

L101-102. Thermal fractionation does not only occur in Greenland, but anytime there is a thermal gradient.

L105: what is $\Omega$

L106: What does this statement mean? Can thermal fractionation not exceed 0.15 permil? Or is this the maximum observed value?

Section 2: It may be more useful to identify the different stages of densification via their density range, rather than their depth range.

L158-160: likely all three stages are blurred in reality, with the densification mechanisms overlapping, see e.g. Hörhold et al. (2011).

L213: remove "s" in "equals"

L236: I thought that $\gamma$ was just an ad-hoc scaling factor to make things fit. Does $\gamma$ have a real meaning, and does it correspond to some physical process?

L248: Please add a multiplication signs ($0.5 \times 10^9$)

L295: note that seasonally sensitive densification rates (as from Arthern et al.) cannot be compared to mean annual densification rates.

L328: Note that Arthern does not attribute his high activation energies to vapor diffusion. Vapor diffusion should have an activation energy of 51 kJ/mol, i.e. the enthalpy of sublimation. Again, please be careful not to conflate activation energies of models that do and do not resolve the seasonal temperature signal.

L357-358: This assumption is probably not valid for vapor diffusion? Section 2.2.3: please specify at what resolution you allow Ca to vary. Do you smooth/average the records in some way?

L376-377: This is an odd definition of the clos-off depth. Isn't the pressure in (closed) bubbles is always higher than that of the atmosphere? What about: the density at which the total pore volume, at the atmospheric pressure of the site, equals the air content of mature ice. (or similar)

L395: polar firn study "sites", . . .

L398 "trapping density" should be "lock-in density"

L403: where does the ln(1/Ac) functional form come from? Is this inspired by theory?

L405: . . .better agreement "between" the modelled LID "and" $\delta^{15}$N . . .

L405: is the $\delta^{15}$N measured on firn samples or ice samples?

L413-414: In fact, it makes it worse.

L458 and L490. The limits of the summation are listed the wrong way around; $i = 1$ should be printed below $N_{max}$

L490: why compare the model to the fit, and not just to the data as was done in Eq. 11? That way you compare apples to apples in Figure 5.

L493-494: is this improvement statistically significant? Please provide t-test significance or similar.

L505-506: I think this is somewhat of a uneven comparison, you should be comparing $\sigma_{model-data}$ to $\sigma_{fit-data}$. $\sigma_{model-fit}$ could in theory be smaller than $\sigma_{fit-data}$, as the data has some inherent scatter and layering.

L513-514: first stage is important in getting the correct $\Delta$age, though.

L541: Do you make a correction for the convective zone?

L544-546: Do you include the geothermal heat flux?

L552-553: How do you include the borehole calibration from Dahl-Jensen et al (1998)? This is not clear

L592-593: How well do the different models fit the exact timing of the d15N increase? This is set by $\Delta$age, so an important test for the models.

L639-642: Again, I think the link between Accumulation and temperature is not as strong as you suggest here, particularly near the margins where most accumulation is delivered by storms.

L666: "densification rates" (add "s")

L682: compatible with the data "except at Talos Dome". Table 3: please be more clear what all the numbers are. I assume these are Q1, Q2, Q3 etc, but this is not very clear.

L706-710: Please rewrite this sentence, it is really hard to follow.

L725: change "less good" to "worse"

L734-736: Could this be because the uncertainty in the input temperature does not include the possibility that the LGM was much warmer than the optimal scenario?

L756-757: Ca from volcanic events?

L817: Is your new modeled EDC $\Delta$age consistent with the work of Parrenin et al. (2013)?

---

## Referee Comment (RC2) · Anonymous Referee #2 · 23 Nov 2016

General:

The authors state that they have improved the LGGE firn densification model based on physical mechanisms. They argue with a better agreement between modelled and measured d15N data. But the physical arguments stand on shaky ground and the better agreement for some sites are opposed by significant lesser agreement of the model data at other sites. A general comment of caution regarding the physical approach: Though the concept of Arzt, based on monosized spheres, which deviates substantially from the physical reality, produces reasonable firn density profiles, they are not really better than those of empirical parametric models. The reason is the rigidity of the 'physical' model and it is not be surprising that an empirical approach with more free

parameters, as e.g. in the Pimienta-model, may even better catch the reality!

The authors now introduce rather arbitrarily two additional Arrhenius-type mechanisms. Thereby they can readily simulate a higher densification rate at very low temperatures. A corroboration of the new model by the better agreement with a small glacial delta age (or delta depth, i.e. shallower glacial firn depth) by Parrenin (2012) is unjustified because this agreement was exactly the purpose of tuning the model.

The approach regarding other transport mechanisms involved in sintering is not convincing. Why should surface diffusion explain the higher densification rates at low temperatures? First, surface diffusion itself does not lead to densification and second also indirect effects will most likely not favor densification. Surface diffusion increases neck diameters and thus decreases pressure at contact area, which decreases creep and in addition it increases curvature radii that decrease the generation of lattice vacancies, and thus decreasing lattice diffusion. Q3, the activation energy applied for surface diffusion seems unrealistically low. Higher values, between 30 and 50 kJ, have been reported.

Considering the influence of dust on densification is interesting but does not substantially contribute to solving the discrepancy between model and date, because the densification enhancement by dust leads for too many sites to a deterioration of the modelled densities. The mentioned possibility of saturation of the densification enhancement by dust at high concentration would only work for Greenland but not for WAIS divide.

My criticism shall not disesteem the huge work accomplished for improving the calibration of the model for modern firn densities that is also presented in this paper. This calibration with new improved firn density profiles certainly leads on average to slight optimization of the model parameters.

However, a better fit to glacial firn depths has only been achieved by a direct tuning of the creep factor at low temperatures. This should be clearly communicated as such. And as the authors mention in the supplement this can be achieved with any other of

the common densification models.

In my view, this lengthy paper clearly raises false expectations. The paper could be organized such that in a first part the improvement of the parameter calibration on modern density profiles are presented and in a second part one could investigate how the creep factor needs to be tuned in order to better simulate the glacial data for the different ice cores, with and without dust enhancement. The diverging results would readily show that so far no unified model can simulate the existing range of data.

Some specific comments:

L. 2: title "constrains" -> constraints

L. 23: " we introduce a dependency of the activation energy to temperature and impurities in the firn densification rate calculation". It is rather a temperature dependent creep factor. The authors 'apply' the impurity dependence, it was 'introduced' by Freitag et al.

L. 79: " questionable when used outside of its range of calibration". Not only then as in case of EDML.

L. 186: Is A0 a constant? Value?

L. 210-228: This 'bending into shape' demonstrates again that a more parametric approach can be closer to reality than a 'pure physical' one.

L. 245 (eq. 6) Is 0.1 bar the pressure at 2 m depth (L. 213)? This should be clarified in the text..

L. 305: " three different mechanisms highlighted above" There are more than 3 mechanisms mentioned above. Table 1, Fig. 2 may not be above.

L. 321, 824: "surface lattice diffusion" Is this term correct for 'surface diffusion'?

L. 358: "assuming that the impurity effect is the same for all mechanisms". This seems

very unlikely!

L. 362,3: "f1" subscript 1

L. 403: "A subscript c" is a strange notation for accumulation rate. Define abbreviation in text. This parameterization is surprising. Of course temperature and accumulation rate are strongly correlated, so we expect an corresponding correlation with accumulation rate. But for two sites with equal accumulation rate but different temperature we don't expect the same LID as densification is strongly temperature dependent. So this parameterization may be justified for most conditions but one must be careful applying it in general and during glacial conditions.

L. 427: Degree of polynomial fit?

L. 405,6: "This parameterization leads to a much better agreement of the modelled LID with d15N measured at the available firn sampling sites than when using the outputs of the old model" This has nothing do with the model. It is just a different parameterization. The better agreement would apply for the old model as well.

L. 456: "rough indicator of data quality" This seems a daring assumption, as natural variability might be in the same order of magnitude.

L. 471: explain "traction constraint"

L. 483 + 497: ?? " the original parameterization of Freitag et al. (2013) always remain in reasonable agreement with the data ….. the incorporation of the impurity effects following the Freitag et al. (2013) parameterization in our model most often deteriorates the model-data agreement".

L. 509: "This effect is due to a densification rate that is too high in the first stage, and this formulation is not affected by the new temperature sensitivity." If this is a general feature, why is it not accounted for correctly?

L. 613: "This observation questions the possible presence of a convective zone and/or

.." Is the presence questioned or the constancy of the convective zone?

L. 689: "Evolution" It is not 'evolution' but 'dependence'. Fig. 8b: Vertical axis title: not Log(A) is shown but A on a log-scale. Fig. 8 could be probably presented in one single graph with A on a log-scale facilitating the comparison between the different temperature ranges.

L. 709: "an inversion of the d15N difference" probably better: "a correct sign of the d15N difference".

L. 711: I strongly question the value of tests A to C. The chosen parameters are very arbitrary. I don't see what the authors want to show us, except that some parameters are better here and worse there, which is rather trivial.

L. 765: ".. instead of the Herron and Langway model.." -> instead of the parameters for the Herron and Langway model.

L. 781: "may act on deformation in opposite way" Why? Please explain.

L. 797: " an up-to-date version " This is an empty phrase.

L. 787: " the new parameterization of the creep parameter preserves good agreement between the old model outputs and data at sites that were already well simulated" Because the creep parameter is kept +- the same, so it is trivial.

L. 815: " This result is in agreement with the recent low delta age estimate by Parrenin et al. (2012) over the deglaciation at Dome C". No surprise, because it has been forced to agree.

L. 850: "Ore" -> Core

---

## Author Response (AR1)

Dear Hubertus,

We have strongly revised the manuscript by addressing all comments of the reviewers as mentioned in the online version and in the attached file.

We also have tried to take into account your comments in the best way.

As for paleoclimatic implications, we have mentioned some of them in the conclusion (yellow highlights). We would prefer not to add much more in order to avoid speculations using this approach which has still not being checked by any experimental set-ups at low temperature.

As for the dust influence, we have included a discussion as mentioned in our previous letter to you in February. A sensitivity test D has been included for showing the effect of "re-parameterizing" the activation energies after implementation of the impurity effects (Figures 7 and 8, Table 3 and associated discussion). In addition, we have detailed how a modification of the Freitag parameterization can improve the modelled $\delta^{15}N$ on the different polar sites over the last deglaciation compared to the current parameterization (Supplementary Text S2 and Figure S10).

Finally, we have noted that one reviewer finds the paper 'lengthy' and we basically agree. One suggestion would be to move in the supplementary materials the section associated with sensitivity tests A, B, C and D as well as the associated Figures 7 and 8 and Table 3. This modification has not be done yet. We are waiting for your advice on this proposition.

You will find in the file "answer to reviewers" the detailed answers to all comments as well as the new manuscript and supplementary materials with modifications highlighted in yellow.

With best regards,

Camille Bréant and the co-authors.

For the text below, the color are used as follows:
- Black: original comments from reviewers
- Red: answer from authors in the online version
- Green: additional precisions for the submission of the revised manuscript

Bréant et al. address an important outstanding problem in ice core research, namely the model-data mismatch of $\delta^{15}$N-N$_2$ as a proxy for firn thickness during the last deglaciation. They offer an interesting new solution to this problem, by proposing a temperature-dependent effective activation energy for firn sintering. In their framework, this can be understood as the effect of three separate firn densification mechanisms working in parallel, each with its own activation energy. Their modified firn densification model provides an improved fit to the deglacial $\delta^{15}$N evolution at cold interior sites, while still being able to fit relatively warm sites that were already modeled well by existing models.
I would ask the authors to consider the following points in a revised manuscript:

We thank the reviewer 1 for his detailed comments and took them into account as detailed below.

• The $\delta^{15}$N model-data mismatch has a long history in ice core research, and is described most clearly by Landais et al. 2006. Several solutions have been proposed for this problem. Without explicitly stating so, the present manuscript takes as the starting assumption that the temperature sensitivity of the densification model must be the problem, due to the absence of a modern analog. I'll refer to this as the "no-analog solution" to the LGM $\delta^{15}$N problem.
I think it would be important to introduce the LGM $\delta^{15}$N problem better, and outline some of the other proposed solutions. For example, Landais et al. (2006) concluded that reconstruction of past accumulation rates was the most likely solution. Why was that explanation abandoned in favor of the no-analog explanation?
It is unclear to me what the main objective is of the present paper. Is the purpose to simply test whether the LGM $\delta^{15}$N problem can be solved using a different activation energy scheme? Or is the purpose to present a new model that will replace the Goujon model in future research at LGGE? Both models fit present day data equally well, so whether the new model is an improvement relies solely on whether you believe the no-analog solution to be the correct one.
Finally, did they solve the problem? From the conclusion section it is not exactly clear whether or not the no-analog and dust mechanisms fully solve the LGM $\delta^{15}$N problem. It seems like the dust mechanism is insufficient by itself, given that it makes sites the fit to sites like GISP2, NGRIP and WAIS Divide worse. The no-analog solution seems to do a better job, yet it requires an unknown process with very low activation energy (see below). Moreover, EDML remains confusing to me. It's warm enough during the LGM to have modern analog sites, yet it does

show the δ¹⁵N model-data mismatch in traditional firn models. I would appreciate some added discussion on whether the LGM δ¹⁵N problem has now been solved satisfactorily, and whether we can forget about other proposed solutions.

The purpose of this paper is to evaluate the LGGE model of firn densification by comparison with data in modern and past climate conditions, with a focus on the major problem of the too small densification rate during the glacial in East Antarctica highlighted in Capron et al. (2013) and earlier studies. We will also state more clearly in the motivation for this study (introduction) that the modelled lock-in depth and associated Δage were too large at Dome C / Vostok during the last glacial maximum with the old model.

Line 132-136: "The differences in modelled and measured $\delta^{15}N$ for glacial period in cold sites of the East-Antarctic plateau have important consequences for the Δage estimate and hence the ice core chronology: using the firn densification models, the modelled Δage for glacial period at Vostok and Dome C is too large by several centuries (Loulergue et al., 2007; Parrenin et al., 2012). "

Indeed, Landais et al. (2006) have suggested that the forcing scenario can explain the model-data mismatch for sites like Law Dome and Kohnen (still, they needed a convective zone of 10 m during the LGM at Kohnen to reconcile δ¹⁵N and firn model run with the low estimate of accumulation rate) but it is not realistic to reconcile δ¹⁵N and model outputs in lower accumulation rate sites like Vostok and Dome C. Such model – data discrepancy was largely discussed in Capron et al. (2013) and we wanted to go one step further.

We did not perform a major revision of the model but tested the effect of relatively simple model changes (e.g. the dependence of activation energy to temperature and dust tests that can easily be adapted to other models). We agree that if the new model provides some ideas to solve the model – data mismatch at low temperature through modification of the activation energy, the problem is clearly not solved and this will be stated clearly and in a coherent manner in both conclusion and abstract. In the revised manuscript, we will thus clarify the fact that the association of three activation energies with three precise physical mechanisms is not proved.

Line 24-32 (Abstract): "We show that both the new temperature parameterization and the influence of impurities contribute to the increased agreement between modelled and measured $\delta^{15}N$ evolution during the last deglaciation at sites with low temperature and low accumulation rate, such as Dome C or Vostok. We find that a very low sensitivity of the densification rate to temperature has to be used in coldest conditions. The inclusion of impurities effects improves the agreement between modelled and measured $\delta^{15}N$ at cold East Antarctic sites during the last deglaciation, but deteriorates the agreement between modelled and measured $\delta^{15}N$ evolution in Greenland and Antarctic sites with high accumulation unless threshold effects are taken into account. "

Line 333-349: Following these arguments and despite the lack of experimental constraints to test this assumption, we propose a new parameterization of the activation energy in the LGGE firn densification model which increases the firn densification rate at low temperatures. We have thus enabled introduction of three adjusted activation energies as proposed in Table 1 and Figure 2. We have replaced the creep parameter in Equation (3) by:

$$A = A_0 \times \left( a_1 \times e^{\frac{-Q_1}{RT}} + a_2 \times e^{\frac{-Q_2}{RT}} + a_3 \times e^{\frac{-Q_3}{RT}} \right) \qquad (7)$$

We have chosen a minimal number of mechanisms (3) for simplicity in the following but the conclusions of our work would not be affected by a choice of more mechanisms.

[Figure]

- Close to melting temperature: mass transfer by diffusion (potential mechanism for high temperature)
  (1) mechanism 1 associated with activation energy $Q_1$

- Low temperature: lattice diffusion (classical mechanism)
  (2) mechanism 2 associated with activation energy $Q_2$

- Very low temperature : boundary diffusion from grain boundary (potential mechanism for low temperature)
  (3) mechanism 3 associated with activation energy $Q_3$

*Figure 2: Different sintering mechanisms of snow for different temperatures proposed by analogy with the hot ceramic sintering (inspired by Figure 1 in Ashby, 1974). Note that more sintering mechanisms can be found in the literature and the attributions of 3 different mechanisms for the firn densification model is only a working hypothesis here.*

Line 953-963 (conclusion): "First, laboratory or field studies of firn densification at very cold controlled conditions are needed to check the predominance of one mechanism over another at low temperature such as the predominance of the boundary diffusion over grain boundary mechanism around -60°C; this is a real challenge because of the slow speed of deformation. Second, we have suggested that the current parameterization of impurity on firn softening should be revised, especially for very high impurity load (Greenland) using for example thresholds on impurity concentrations. Third, the separate effects of impurities and temperature on firn densification and hence $\delta^{15}N$ evolution should be tested on periods other than the last deglaciation. Sequences of events associated with non-synchronous changes in surface temperature, accumulation rate and impurity content would be particularly valuable for this objective."

We will also introduce better the $\delta^{15}N$ problem referring to previous works (Landais et al., 2006; Capron et al., 2013; …) and summarizing the associated results and other options to solve the problem (e.g. convective zone, thermal effect).

Line 137-153 (Introduction): "Several hypotheses have already been evoked to explain the $\delta^{15}N$ model-data mismatch in Antarctica as detailed in Landais et al. (2006), Dreyfus et al. (2010) and Capron et al. (2013). First, the firnification models have been developed and tuned for reproducing present-day density profiles and it is questionable to apply them to glacial climate conditions in Antarctica for which no present-day analogues are available. Second, increasing impurity concentration has been suggested to fasten firn densification during glacial period (Freitag et al., 2013; Hörhold et al., 2012). Third, a ~20 m deep convective zone has been evidenced in the megadunes region in Antarctica (Severinghaus et al., 2006) hence suggesting that deep convective zones can develop in glacial periods in Antarctica and explain the mismatch between firn densification model and $\delta^{15}N$ data (Caillon et al., 2003). This hypothesis can explain the mismatch between modelled and measured $\delta^{15}N$ at EDML during glacial period by invoking a 10 m convective zone (Landais et al., 2006). However, it has been ruled out for explaining the strong mismatch between model and $\delta^{15}N$ data at EDC for the last glacial period (Parrenin et al., 2012). Fourth, firn densification is very sensitive to changes in temperature and accumulation rate so that uncertainties in the surface climate parameters can lead to biased value of the modelled LID and hence $\delta^{15}N$. Fifth, a significant thermal fractionation signal can affect the total $\delta^{15}N$ signal. However, this hypothesis has been ruled out by Dreyfus et al. (2010) based on $\delta^{15}N$ and $\delta^{40}Ar$ data on the last deglaciation at EDC."

In the conclusion and perspectives, we will also mention the possibility to improve the constraints on firn modeling through the use of cross-dating on new ice cores with high resolution signals as already used by Parrenin et al. (2012).

Line 974-976: "Finally, additional constraints on the firn modelling can also be obtained through the use of cross-dating on new ice core with high resolution signals as already used by Parrenin et al. (2012). »

• To get the densification rate to increase meaningfully at low temperatures, the authors have to introduce a densification process with an extremely low activation energy of Q3=1.5 kJ/mol (low enough to be essentially temperature-insensitive).
They suggest this process to be surface diffusion. However, experimental studies suggest the activation energy for ice surface diffusion is on the order of 14 to 38 kJ/mol (e.g. Jung et al., doi: 10.1063/1.1770518, Nie et al., doi: 10.1103/Phys-RevLett.102.136101, and references therein). The value used by Bréant et al. seems an order of magnitude too small to be surface diffusion. Therefore, they are essentially invoking densification by an unknown process with very small Q.
The authors should acknowledge that the values they use for Q3 seems unrealistically low. In my view, this is an important piece of evidence that the "no-analog assumption" by itself may be insufficient to solve the LGM δ15N problem – the authors may not share this view.
At the other end, their high-Q process (suggested to be vapor diffusion) has a value that seems too high at Q1=110 kJ/mol. Vapor diffusion scales with the vapor pressure, and the enthalpy of sublimation in ice is only 51 kJ/mol.

We thank the Referee for providing references for the activation energy of ice surface diffusion. We will include a test of a 15 kJ/mol in replacement of our previous "test C" in the revised manuscript. Using 15 kJ/mol leads to intermediate results between the "old" and "new" model results.

Test C is included in Figure 7, Figure 8 and Table 3.
Corresponding discussion in lines 825-831: "Test C has been designed so that the activation energy at low temperature corresponds to estimates of activation energy for ice surface diffusion (Jung et al., 2004; Nie et al., 2009), a mechanism that is expected to be important at low temperature (Ashby, 1974). Using such a parameterization leads to a fair agreement between the modelled and the measured $\delta^{15}$N change over the last deglaciation for the different sites. At Dome C, the correct sign for the $\delta^{15}$N evolution between LGM and the Holocene is predicted by the model. However, the modelled $\delta^{15}$N increase is still too small compared to the data and the $\delta^{15}$N calculated by the "new model"."

Actually, the only way to reconcile our model with $\delta^{15}$N data by invoking a change of activation energy with temperature is to have a negligible influence of temperature on the densification rate below -50°C. We agree that the values are surprisingly low for activation energy at low temperature in order to fit the $\delta^{15}$N data. There is a clear limitation of our approach which is empirical, and cannot isolate mechanisms, but it calls for further lab and field studies. The effect of temperature could also be misrepresented in our model (and in other densification models) by other ways than the value of activation energy.

The need for experimental check is stated in the conclusion.

Lines 964-968: "First, laboratory or field studies of firn densification at very cold controlled conditions are needed to check the predominance of one mechanism over another at low temperature such as the predominance of the boundary diffusion over grain boundary mechanism around -60°C; this is a real challenge because of the slow speed of deformation."

As written above, we will make clear that the association of our three activation energies with three precise physical mechanisms is not proved. Indeed, while several mechanisms have been highlighted for the densification of ice over several temperature ranges, there is no unambiguous attribution of a particular mechanism to a particular temperature. The determination of the activation energies for our model has an empirical basis as for previous studies. We will further emphasize this aspect in the revised version, especially when discussing Figure 2.

cf line 333-349: "Following these arguments and despite the lack of experimental constraints to test this assumption, we propose a new parameterization of the activation energy in the LGGE firn densification model which increases the firn densification rate at low temperatures. We have thus enabled introduction of three adjusted activation energies as proposed in Table 1 and Figure 2. We have replaced the creep parameter in Equation (3) by:

$$A = A_0 \times \left( a_1 \times e^{\frac{-Q_1}{RT}} + a_2 \times e^{\frac{-Q_2}{RT}} + a_3 \times e^{\frac{-Q_3}{RT}} \right) \qquad (7)$$

We have chosen a minimal number of mechanisms (3) for simplicity in the following but the conclusions of our work would not be affected by a choice of more mechanisms.

[Figure]

- Close to melting temperature: mass transfer by diffusion (potential mechanism for high temperature)
  (1) mechanism 1 associated with activation energy $Q_1$

- Low temperature: lattice diffusion (classical mechanism)
  (2) mechanism 2 associated with activation energy $Q_2$

- Very low temperature : boundary diffusion from grain boundary (potential mechanism for low temperature)
  (3) mechanism 3 associated with activation energy $Q_3$

*Figure 2: Different sintering mechanisms of snow for different temperatures proposed by analogy with the hot ceramic sintering (inspired by Figure 1 in Ashby, 1974). Note that more sintering mechanisms can be found in the literature and the attributions of 3 different mechanisms for the firn densification model is only a working hypothesis here.*"

Despite some empirical approach, it should be noted that although not necessarily linked to vapor diffusion, high values of activation energy have indeed been measured at high temperatures: values of $Q_1$ derived from ice creep tests (Jacka and Li 1994, Morgan 1991) can be up to 170 kJ/mol. Note also that the exact value of $Q_1$ does not influence the result of our study since we are working at low temperature where the effect of $Q_1$ is minor.

Line 323-325: an activation energy significantly higher than 60 kJ/mol could be favoured (up to 177 kJ/mol between -1 and -5°C [Jacka and Li, 1994]) in order to best fit density profiles with firn densification models (Arthern et al., 2010; Barnes et al., 1971; Jacka and Li, 1994, Morgan, 1991)."

Finally, note that the range of effective activation energies that we use ($Q_{eq}$ see supplementary figure S4) and allows to correctly simulate the observed densification rates is much smaller than the range of the $Q_1$, $Q_2$, $Q_3$ values. Above -25°C, $Q_{eq}$ ranges between 58 and 60 kJ/mol.

• Ultimately the goal of firn modelling is to predict Δage, and therefore I was surprised that no Δage results are shown. How does the new activation energy scheme change the simulated Δage? In Greenland we have direct constraints on Δage via the thermal δ15N signals. How well does the model capture those? Likewise, in evaluating the model performance on page 18, the authors test only how well the model predicts the LID (i.e. δ15N). The more important metric, in my mind, is how well the model predicts Δage. This can be evaluated via the integrated density from the surface to the LID (because Δage is essentially the mass of overlying snow divided by the accumulation rate).

The following table will be inserted in the supplement (together with supplementary table S1) for the comparison of Δage at the bottom of the firn for the different sites studied here. Even if the comparison based on Δage or on modeled density profiles does not lead to exactly the same conclusion for each site, the main features already observed for the comparison of the standard deviation between modeled and measured density profiles are also observed here. This is the case for the worsened agreement between model and data at Talos and Mizuho when using the new parameterization or the improved model vs data agreement at low temperature and low accumulation sites of the bottom of the table (B32, EDML, South Pole, Dôme C, Vostok, Dome A).

| Δage | Data | Old | New | New + dust |
|---|---|---|---|---|
| Dye 3 | 78 | **68.4** | 61.8 | - |
| DE08 | 35.5 | **36.8** | 31.3 | - |
| km105 | 104.4 | 111.7 | **106.4** | - |
| Site2 | 112.2 | **109.2** | 103.3 | - |
| Siple Dome | 329 | 287.9 | **296.9** | 284.9 |
| D-47 | 152 | **145.3** | 141.7 | - |
| Byrd | 238.9 | 225 | **226.9** | - |
| NEEM | 209.4 | 187.4 | **191.9** | 187.4 |
| Crete | 156.1 | **150** | 145.7 | - |
| km200 | 137.9 | 156.7 | **152.6** | - |
| WAIS divide | 225 | 206.5 | 206.5 | **233.3** |
| North GRIP | 248.4 | 226.2 | **236.5** | 224.5 |
| GRIP | 209.9 | **205.7** | **205.7** | 200 |
| B29 | 270.3 | 251 | **264.7** | 252.9 |
| Mizuho | 483.7 | **518.4** | 557 | - |
| Talos | 554.1 | **592.6** | 637.7 | 656.4 |
| B32 | 896.8 | 816.1 | **889.9** | 978.4 |
| EDML | 874.5 | 787.3 | **852.9** | 899.7 |
| SP | 1160.9 | 965.2 | 1002.8 | **1068.7** |
| DC | 2639.1 | 2473 | 2461 | **2557** |
| Vostok | 2814.3 | 2960.4 | **2810.4** | 2919.5 |
| Dome A | 2812.7 | 3024.8 | **2764** | - |

This table and associated caption is included in the supplement, Supplementary Table S2

For paleo application, we think that it is better to compare model outputs to $\delta^{15}N$ profiles because we do not have direct indication of the $\Delta$age (it necessary depends on the timescale). The only way to compare $\Delta$age model output with data is actually on the abrupt warming recorded very clearly both in the $\delta^{18}O$ in the ice phase and $\delta^{15}N$ in the gas phase on the NGRIP ice core because we have an accurate associated timescale (GICC05). This is the reason why we will present a table in the supplement showing the $\Delta$age outputs for the different model parameterizations and a comparison with data estimates based on $\delta^{18}O$ and $\delta^{15}N$ records of the abrupt warming in the ice and in the gas phases. The agreement between data and model is slightly better when using the new version but the addition of dust leads to a strong deterioration as observed on the $\delta^{15}N$ profiles.

| NGRIP | $\Delta$age (old version) | $\Delta$age (new version) | $\Delta$age (new version + dust) | DATA |
|---|---|---|---|---|
| Bolling /Allerod | 870 years | 920 years | 740 years | 1040±100 years |
| End of Younger Dryas | 760 years | 820 years | 640 years | 800±100 years |

NB: the uncertainty on the $\Delta$age from the data is mainly linked to the resolution of the $\delta^{15}N$ signal.

This table is included in the supplement, Supplementary Table S5

• On lines 624-625 the authors conclude that uncertainties in the climate forcing cannot explain the LGM δ15N problem. However, to solve the δ15N problem one would need to make the LGM warmer, not colder! For some reason the temperature uncertainties in Fig. S9 are applied very asymmetrically, such that the LGM is always very cold.

The uncertainties for the LGM temperature estimate on the Antarctic plateau are given in Jouzel et al. (2003) as cited in the manuscript in discussion. In this study, the authors have gathered all available constraints for the amplitude of temperature change between the LGM and the Holocene mainly at the Vostok and Dome C sites. The main conclusion is that temperature uncertainty for the amplitude of the last deglaciation is estimated to -10% to +30% in Antarctica. The reason for such asymmetry is mainly linked to outputs of atmospheric general circulation models equipped with water isotopes. These models suggest that the present day spatial slope between $\delta^{18}O$ and temperature most probably underestimate the amplitude of the temperature change between glacial and interglacial period. We have followed this estimate of asymmetric uncertainty on the amplitude of temperature change during deglaciation and this is the reason why the scenario with warmer LGM is not very different from the control scenario. We do not have references suggesting a much warmer LGM for Antarctica than the classical estimate from water isotopes. More recent studies focusing on the validity of the isotopes vs temperature relationship in Antarctica have also suggested that this relationship can be applied with confidence for glacial temperature reconstruction (Cauquoin et al., 2015) why one should be cautious for past interglacial temperature reconstruction (Sime et al., 2009). Finally, a recent estimate of the deglacial temperature increase based on $\delta^{15}N$ measurements at WAIS (Cuffey et al., PNAS, 2016) led to a 11.3°C temperature increase over the last deglaciation (1°C warming to be attributed to change in elevation), larger than the temperature increase reconstructed in East Antarctica from water isotopes by 2-4°C. This last estimate is against not in favour of a "warm" LGM.

Still, it should be noted that the uncertainty of 20% on LGM accumulation rate on central sites as given by DATICE in the AICC2012 construction is probably overestimated. Indeed, deglaciation occurs around 500 m depth at Dome C, hence with small uncertainty on the thinning function. Dating tie points can thus constraint quite tightly the accumulation rate.

Summarizing, we believe that the uncertainties given in our manuscript correspond to up to date uncertainties on temperature and accumulation for the last glacial maximum and do not underestimate the possible range of temperature and accumulation rate values for the LGM.

We will complete our manuscript with some explanations given above.

Line 643-669: "The reason for such asymmetry is mainly linked to outputs of atmospheric general circulation models equipped with water isotopes. These models suggest that the present day spatial slope between $\delta^{18}$O and temperature most probably underestimate the amplitude of the temperature change between glacial and interglacial period. We have followed this estimate of asymmetric uncertainty on the amplitude of temperature change during deglaciation in our study. Recent studies have also suggested that the relationships between water isotopes and temperature and between water isotopes and accumulation rate can be applied with confidence in Antarctica for glacial temperature reconstruction (Cauquoin et al., 2015) while one should be cautious for interglacial temperature reconstruction with warmer conditions than today (Sime et al., 2009). Finally, a recent estimate of the deglacial temperature increase based on $\delta^{15}$N measurements at WAIS (Cuffey et al., 2016) led to a 11.3°C temperature increase over the last deglaciation (1°C warming to be attributed to change in elevation). This is larger than the temperature increase reconstructed in East Antarctica from water isotopes by 2-4°C and again not in favour of a "warm" LGM.

In the construction of the AICC2012 chronology (Bazin et al., 2013; Veres et al., 2013), the first order estimate of accumulation rate from water isotopes for EDML, Talos Dome, Vostok and Dome C has been modified by incorporating dating constraints or stratigraphic tie points between ice cores (Bazin et al., 2013; Veres et al., 2013). The modification of the accumulation rate profiles over the last deglaciation for these 4 sites is less than 20% and the uncertainty of accumulation rate generated by the DATICE model used to build AICC 2012 from background errors (thinning history, accumulation rate, LID) and chronological constraints is 30% for the LGM (Bazin et al., 2013; Frieler et al., 2015; Veres et al., 2013). Still, it should be noted that the uncertainty of 20% on LGM accumulation rate on central sites as given in the AICC2012 construction is probably overestimated. Indeed, deglaciation occurs around 500 m depth at Dome C, hence with small uncertainty on the thinning function and on the accumulation rate. These values are consistent with previous estimates of accumulation rate uncertainties over the last deglaciation (± 10% for Dome C (Parrenin et al., 2007) and ± 30% in EDML (Loulergue et al., 2007))."

• It is not very clear how the parameters in Table 1 were selected. How was the model calibrated? Did the authors minimize some cost function? The authors do give three representative examples in Table 3, but no criteria for choosing the best model.
In their preferred model (Table 1), process 1 (Q1 = 110 kJ/mol) doesn't do much.
At all relevant temperatures, process 2 is at least an order of magnitude larger.

The values for the prefactors a1, a2 and a3 have been chosen to best reproduce the $\delta^{15}$N variations over deglaciation at Dome C or Vostok while keeping a good agreement (at least not deteriorating the model-data agreement obtained with the old LGGE model) for (1) the deglacial $\delta^{15}$N variations at higher accumulation rate sites and (2) the density profiles for present-day firn. Hundreds of

sensitivity tests have been performed using a strategy based on dichotomy to reduce the mismatch between modeled and measured $\delta^{15}N$ change over the last deglaciation at low accumulation rate sites. The constraint of keeping a correct agreement of model results with present day density profiles and for the last deglaciation at warm sites strongly reduces the possible choices of $a_i$ and $Q_i$. This is illustrated by the small range of Qeq values on supplementary Figure S4.

This will be better explained in the new manuscript.

Line 358-363: "Hundreds of sensitivity tests have been performed using a strategy based on dichotomy to reduce the mismatch between modeled and data. The constraint of keeping a correct agreement of model results with present day density profiles and for the last deglaciation at warm sites strongly reduces the possible choices of $a_i$ and $Q_i$ (Section 3). The best value obtained for $Q_3$ is lower than published values for surface or boundary diffusion but is necessary to reproduce the deglaciation at cold East Antarctic Sites. Sensitivity test C will illustrate the effect of using a higher value."

• The authors claim that the new model provides a better fit to modern data than the old model. The LID prediction improves by 1.2 +- 6 meter. That hardly seems like a statistically significant improvement. Using as Student's t-test it should be trivial to show whether the null hypothesis (both models perform equally well) can be rejected.
As mentioned earlier, I would encourage the authors to not only compare the predicted LID to the fitted data, but also to compare the integrated density in the simulations to the integrated density of the fit. The latter metric is more representative of Δage.

We agree that the new model is not significantly better than the old one for density fit as well as LID and Δage predictions. This will be written more clearly in the new manuscript. As explained above, our main aim is to preserve the good agreement (no deterioration) between modeled and measured firn density profile while improving the $\delta^{15}N$ model- data agreement over the last deglaciation on the East Antarctic plateau.

Line 548-553: "Still, looking at all different firn profiles, the general agreement between modeled and measured firn density profiles is preserved. The agreement between measured and modeled firn density is increased for some sites at (1) low accumulation rate and temperature in Antarctica (Dome A, Vostok and Dome C but not South Pole) and at (2) relatively high temperature and accumulation rate (Dye 3, Siple Dome, NEEM). In parallel, a larger disagreement between model and data is observed for some other sites particularly in coastal Antarctica (DE08, Km 200, WAIS Divide)."

Line 564-578: "The introduction of three different activation energies for different temperature ranges leads to changes of the modeled density profiles at high densities (above about 800 kg/m$^3$). A clear improvement is obtained for example at South Pole (Supplementary Figure S6), although the overall impact of using three activation energies remains small.

The incorporation of the impurity effect following the Freitag et al. (2013) parameterization in our model slightly deteriorates the model-data agreement because no specific re-adjustment of model parameters was performed. However the model prediction of the density profiles remains correct although the impurity effect parameterization was developed for a different purpose: simulating

density layering (Freitag et al., 2013). This encouraged us to test this simple parameterization in glacial climate conditions.

Overall, in terms of $\sigma_{model-fit}$, only an insignificant improvement (about 3%) is obtained by using the modified model (3 activation energies and implementation of impurity effect) rather than the former Goujon et al. (2003) mechanical scheme. However a systematic improvement is obtained at the six coldest sites."

Δage calculation for present-day firn will be provided in the supplement.

This result is displayed in supplementary Table S2

• The closest analog to LGM conditions in East Antarctica is the Dome A site, with mean annual temperatures below 58°C. The old LGGE model provided a reasonable fit to density at this site (Cunde et al., 2008). Why is this site not included in the calibration data set?

Dome A will be included in our revised manuscript as Dr Cunde kindly provided us the necessary data. Recent estimates of the accumulation rate at Dome A lead to a value of 2.3 cm w.eq. / yr (close to the values at Vostok and Dome C). With this accumulation rate, the model – data agreement is improved in the "new" simulations (see e.g. Δage values in the above table).

Dome A has been included in the different figures and tables in the main text and the supplement corresponding to section 3.1.

• The MS does not give many technical details about the running of the models. What time and spatial step size are used? What is the lower model boundary? What geothermal heat flux is used? The latter is important in the stagnant firn columns of the LGM.

The model spatial grids and lower boundary condition were not changed compared to the former model (Goujon et al. (2003), and the time step (site dependent) was chosen to lead to stable results. A section will be added in the Supplement to complete the description of the model running conditions.

The details are given now in supplementary Text S1

• The authors test the dust softening hypothesis of Johannes Freitag and Maria Hörhold. It is important to note that this model of Freitag et al (2013) was designed to simulate layering, rather than bulk density as it is used here. How were Ca data averaged in the model runs? How does the Ca data resolution compare to the model resolution?
Please discuss some of the caveats regarding dust. From talking to Johannes Freitag, I get the impression he believes layering to be more relevant than bulk density in this regard. What do the authors recommend? Should future users incorporate dust or not?

Our main intention is to show that the Freitag et al. (2013) parameterization leads to interesting results and that the dust effect deserves further attention in future studies. When transposed to our model, it provides a reasonable fit of the data at all modern sites. Then, we acknowledge that

the implementation of Ca effect in addition to the effect of temperature influence on the activation energy Q is not optimal since the parameterization of Q1, Q2, Q3 and associated a1, a2, a3 has been adjusted without Ca effect implementation. The implementation of Ca hence necessarily slightly deteriorates the model – data agreement.

As for the layering and its effect on the LID, it cannot be simply implemented in our firn densification model. This is the reason why the dust has been implemented through its influence on the bulk density which is indeed different from the simulations of Freitag et al. (2013). We will clarify the fact that we do not expect the Freitag et al. (2013) parameterization to be properly tuned for simulating the variations of firn thickness or Δage as it was not designed for this purpose.

Line 408-413: "The values of $a_i$ and $Q_i$ were not readjusted after the implementation of impurity effects to avoid adding tuning parameters. Still, because the large range of calcium concentrations encountered in past climate conditions has a strong impact on model results, this may be a solution to reduce the model-data mismatch. This is explored in Section 3 through a sensitivity test D. In the same section, we will also propose a modification of the Freitag parameterization using thresholds to reduce the model-data mismatch."

We expect that the dust influence on bulk density or individual layer density is quite similar. Indeed, from a physics point of view, the density of a given layer is not directly dependent of the density of the surrounding layers as the load pressure on a layer is only controlled by the weight of the layers above (which depends on the precipitation rate but not their density). Thus the impurity effects on the bulk density and on individual layers density should be similar. This probably explains why the comparison is rather good between the polynomial fit of the measured density profiles (hence excluding layering) and modeled firn density profiles with dust influences according to Freitag et al. (2013).

For modern simulations, a single average calcium concentration is used (the value in Table S1). In past climate simulations, the calcium concentration data used were directly taken from the source data file available for each ice core. The temporal variations in Calcium concentrations are then simply interpolated at each model time step. The details on the temporal resolution will be given in the supplement together with the parameters used to run the model. The caveats regarding dust (e.g. using calcium as a diagnostics of dust content and the intention of the Freitag et al. (2013) parameterization) will be better introduced.

Supplementary Text S1 and line 406-407: "We use raw data of the calcium concentration for all the sites when available even if question may arise on calcium concentration being the best diagnostic for dust content."

• I do not understand the rationale behind the LID parameterization of Equation 10. The authors use a very complicated way to define the LID, namely the depth where the modelled δ15N in the open pores matches the δ15N in the mature ice. This approach involves simulating the bubble trapping process, which is very (!!) poorly understood. The depth range of bubble trapping is completely unknown at most sites, unless measurements of closed porosity are available (which is not the case at most sites). Also, trapping depends strongly on density layering, as early work at Law Dome and more recent work by Rachael Rhodes and others have shown.

We agree with the Referee on the fact that gas trapping is insufficiently understood. However, the same uncertainties apply when using $\delta^{15}N$ measurements in ice (after this trapping) in past climate conditions to evaluate our model results. The intention of our definition referring to $\delta^{15}N$ in the mature ice is to get closer to what is used for past climate conditions. Systematic measurements of $\delta^{15}N$ in recent ice would be very helpful in the future to improve an LID definition relevant for ice-core interpretation. This will be mentioned in the revised manuscript.

The section 2.4 on the LID has been fully rewritten

The lock-in depth is very clearly visible in firn air sampling data, as an abrupt change in the concentration slope of many tracers. Why not use this commonly used and simple metric, which can be directly derived from data? I fear that modelling something as complicated as bubble trapping could easily lead to errors. How does the fit of Figure S5 look when using the common definition of the lock-in depth?

Different thresholds based on trace gas data in the open porosity of the firn were tested in Table 2 of Witrant et al. (2012). Unfortunately, different definitions (based on greenhouse gas concentration slope change, the $\delta^{15}N$ plateau, etc.) lead to fairly large differences especially at the most arid sites of the central Antarctic plateau which are a major focus of this study. In particular, as no $\delta^{15}N$ plateau is observed at these sites, the progressive bubble trapping should lead to less fractionated $\delta^{15}N$ values in mature ice than in deep firn at these sites. Only fairly small density variations are allowed by our LID parameterization. They will be further discussed in the revised manuscript. Our major conclusion is that the LID definition does not explain the mismatch between model results and $\delta^{15}N$ data during deglaciations.

The whole section 2.4 has been rewritten

• One of the important achievements of this work is to compile a large database of reliable firn density measurements. This would be an extremely valuable resource to the firn research community if it were publicly available in a format that is easy to use. I would like to kindly ask the authors to make this database publicly available as a supplement to the manuscript, as is also strongly encouraged by the data policy of the journal (Climate of the Past).
The manuscript does not have a statement of data availability yet, which will need to be added as per the editorial guidelines of CP.

We made our best to document the nature and sources of the density data that we used. We are not the owner of the data in most cases, this is why we will provide the polynomial fits of the data rather than the datasets.

We have gathered all polynomial fits in a document to be posted as the Supplementary material

• throughout the paper the authors refer to "snow" as everything above the critical density, and "firn" as everything below. This is not common usage, and should be specified.

We use the same words as Goujon et al. (2003). This will be explained in the new manuscript.

Line 185-186: "The first stage, named "snow densification" as in Goujon et al. (2003),"

Some minor comments and typos:

L2: "constrains" should be "constraints" ok

L23: "to" should be "on" ok

L24: "existence". Maybe "dominance" would be better? ok

L35: "depict". What about "reconstruct"? ok

L61: The close relationship between A and T seems to be mostly an assumption. See
e.g. Monnin (2004), Fudge (2016) or Van Ommen (2004).

This is indeed true but we focus better on site from the Antarctic plateau where the link should be true at least qualitatively. Moreover, we expect a certain correlation between change in accumulation rate and change in temperature when considering long term averages (several thousands of years for the LGM) while a certain decoupling between accumulation and temperature is expected at short scale because of the different possible snow accumulation processes. This will be explained in the new manuscript.

Line 61-70: "On glacial – interglacial timescales, increasing temperature is associated with increasing snow accumulation. Indeed, the thermodynamic effect dominates when dealing with long term averages (several thousands of years), even if accumulation and temperature are not always correlated on millennial and centennial timescale in polar regions, especially in coastal areas (e.g. Fudge et al., 2016; Altnau et al., 2014). As a consequence, when comparing LGM and Holocene averages, we observe for all available ice cores covering the last deglaciation increases in both accumulation and temperature. In the firn densification model, both effects partially compensate each other, with the temperature effect being dominant in the current densification models for the LID simulation over glacial – interglacial transitions in deep drilling sites of the East Antarctic plateau, hence leading to the modelled LID decrease."

L65: in HL the "change" in pore space is proportional to the increase in weight.

Yes, this is the meaning of our sentence, accumulation rate being directly linked with weight.

Line 75: "the weight of the overlying snow, hence the accumulation rate."

L89: Better write: "In the absence of thermal gradients", the δ15N trapped : : :.. The geothermal heat flux matters also ok

L101-102. Thermal fractionation does not only occur in Greenland, but anytime there is a thermal gradient.

This is true but in order to have a strong thermal fractionation, we need abrupt temperature changes at the surface. We have thus added "strong" before "thermal fractionation"

L105: what is $\Omega$ ?

This will be explained in the new manuscript (Grachev et Severinghaus, 2003)

Line 117: "where $\Omega$ is the thermal fractionation coefficient"

L106: What does this statement mean? Can thermal fractionation not exceed 0.15 permil? Or is this the maximum observed value?

This will be explained in the new manuscript

Line 119: "(the $\delta^{15}N_{therm}$ observed is in most cases lower than 0.15‰)."

Section 2: It may be more useful to identify the different stages of densification via their density range, rather than their depth range.

Yes, we are agree, this will be changed is the new manuscript.

Done.

L158-160: likely all three stages are blurred in reality, with the densification mechanisms overlapping, see e.g. Hörhold et al. (2011).

We agree with this comment, we will precise it in the new version.

Line 192-193: "In reality, the adjacent densification mechanisms likely coexist at intermediate densities."

L213: remove "s" in "equals" ok

L236: I thought that was just an ad-hoc scaling factor to make things fit. Does Y have a real meaning, and does it correspond to some physical process?

Equations (4) and (5) in our manuscript show the relationship between $\gamma$ used in Goujon et al., 2003 and the parameters in Alley, 1987 : $\gamma = (2/15)(\lambda/v)(R/r^2)$ where $\lambda$ is the bond thickness, v the bond viscosity, R the grain radius and r the bond radius. Alley (1987) then then calculates and discusses the activation energy for viscosity. On the other hand Goujon et al. (2003) simply used an adjusted value of $\gamma$, as explained in our manuscript text following Equation 5. We replace $\gamma$ in Equation 4 by $\gamma'\exp(-Q/RT)$ in Eq. 6 and evaluate that Q=-49.5 kJ/mol. As $\gamma'$ is still adjusted in the model, using $\gamma$ or $\gamma'\exp(-Q/RT)$ does not change the model results.

In the modified manuscript, we will remind the relationship between Y and the physical parameters in Alley (1987) at line 236.

Line 265-272: "Alley (1987) calculated a viscosity ($\nu$) related activation energy of 41 kJ/mol, consistent with recommended values for grain-boundary diffusion (42 kJ/mol) or measured from grain growth rate (Alley, 1987 and references therein). In Goujon et al. (2003), no explicit temperature effect is used but the parameter $\gamma$ varies by several orders of magnitude from site to

site. The parameter $\gamma$ is calculated to maintain a continuous densification rate between the first and second stages at a chosen critical density. We translate the variations of $\gamma = (2 \lambda R) / (15 \nu r^2)$ from site to site into $\gamma = \gamma'$ exp($-Q/RT$) , and calculate the activation energy Q using a classical logarithmic plot as a function of 1000/T (see e.g. Herron and Langway, 1980).”

L248: Please add a multiplication signs (0:5 _ 109) ok

L295: note that seasonally sensitive densification rates (as from Arthern et al.) cannot be compared to mean annual densification rates.

The large values up to 100-130kJ/mol are originally from Morgan et al, 1991, and Jacka and Li 1994 and their values are not related to seasonal thermal gradients, which are not taken into account in this study. Also note that the equivalent effective activation energy in out model (Qeq) remains lower than 60 kJ/mol (see supplementary Figure S4 and answer to second general comment).

L328: Note that Arthern does not attribute his high activation energies to vapor diffusion. Vapor diffusion should have an activation energy of 51 kJ/mol, i.e. the enthalpy of sublimation. Again, please be careful not to conflate activation energies of models that do and do not resolve the seasonal temperature signal.

We removed the attribution to a specific mechanism and simply note: "At warm temperature, empirical determinations of Q1 lead to values of the order of 100-130 kJ/mol (Arthern et al., 2010; Barnes et al., 1971; Zwally and Li, 2002)."

Line 322-325: “Actually, it has been observed that, at warm temperature, an activation energy significantly higher than 60 kJ/mol could be favoured (up to 177 kJ/mol between -1 and -5°C [Jacka and Li, 1994]) in order to best fit density profiles with firn densification models (Arthern et al., 2010; Barnes et al., 1971; Jacka and Li, 1994, Morgan, 1991).”

These high values are actually not linked to a seasonal temperature signal: Jacka and Li (1994) made isothermal laboratory measurements to determine them.

L357-358: This assumption is probably not valid for vapor diffusion?

This is indeed correct. We will specify it. It should still be noted that this does not have any significant influence for the relatively cold sites studied here.

Line 388-392: “We have implemented this parameterization in our model with the simple assumption that the impurity effect is the same for all mechanisms. It allows us to keep the number of tunable parameters to a minimum, even though this assumption is probably not correct for the vapor diffusion process. Note however that this will not affect the applications discussed below since vapor diffusion is only important for warm sites.”

Section 2.2.3:
please specify at what resolution you allow Ca to vary. Do you smooth/average the records in some way?

No, the records are not smoothed (cf answer on dust data above). This will be specified in the new manuscript.

cf Supplementary Text S1

L376-377: This is an odd definition of the clos-off depth. Isn't the pressure in (closed) bubbles is always higher than that of the atmosphere? What about: the density at which the total pore volume, at the atmospheric pressure of the site, equals the air content of mature ice. (or similar)

We use the same definition of the close-off density as in Goujon et al. (2003). It defines the boundary between the second (firn) and third (ice) mechanical formulations in the model. As no model change was performed, the phrasing will be simplified in the revised manuscript.

Line 416-418: "As in Goujon et al. (2003), the final densification stage begins at the close-off density derived from air content measurements in mature ice. Further porosity reduction results in an air pressure increase in the bubbles"

L395: polar firn study "sites", : : : ok
L398 "trapping density" should be "lock-in density" ok
L403: where does the ln(1/Ac) functional form come from? Is this inspired by theory?

No, this is just an ad-hoc adjustment. In the previous model version (Goujon et al., 2003), the LID definition was related to a value of the closed/total porosity ratio adjusted for each drilling site but time independent (see lines 387-389). This dependency on the geographic location with independence from climate is not completely self-consistent. This is why we tested another LID definition.
The description of our LID definition and tests will be modified following the comments from both Referees.

Section 2.4

L405: : : :better agreement "between" the modelled LID "and" δ15N : : : ok

L405: is the δ15N measured on firn samples or ice samples?

Line 4

Yes, they are measured in firn samples, we will replace « measured at the available firn sampling sites » by « measured in firn samples at available sites »

L413-414: In fact, it makes it worse. ok

L458 and L490. The limits of the summation are listed the wrong way around; i = 1 should be printed below Nmax ok

L490: why compare the model to the fit, and not just to the data as was done in Eq. 11? That way you compare apples to apples in Figure 5.

There is no simple way to compare model results and density data in a multi-site consistent way due to the strong site to site differences in the data (measurement techniques, sample size, number of

samples, depth dependent resolution, etc.). Model results are compared to the fit in order to better characterize:
- bulk density (as opposed to density variability) which we aim at predicting with the model
- site to site variations in the quality of the model prediction of the bulk density profile. The regular depth resolution used leads to an integrative diagnostics somewhat comparable to delta-age.
The manuscript section between lines 492 and 511 will be revised.

Line 513-523: "Note that we compare here the model to the fit of the data and not directly to data because of the strong site to site differences in the data (e.g. data resolution, sample size). Figure 5 and Supplementary Table S1 display the $\sigma_{model-fit}$ for the 22 different sites before and after modifications detailed in Section 2.

3.1.1. Data – model comparisons using the old model

Comparing our model results to density data is not trivial due to the diversity in measurement techniques and samplings discussed above, as well as the natural variability in density that we do not capture with a simplified model aiming at simulating very long time scales. A rough indication is given by comparing $\sigma_{model-fit}$ and $\sigma_{fit-data}$."

L493-494: is this improvement statistically significant? Please provide t-test significance or similar. No, it is not, this will be mentioned in the revised manuscript

Line 548-553: "Still, looking at all different firn profiles, the general agreement between modeled and measured firn density profiles is preserved. The agreement between measured and modeled firn density is increased for some sites at (1) low accumulation rate and temperature in Antarctica (Dome A, Vostok and Dome C but not South Pole) and at (2) relatively high temperature and accumulation rate (Dye 3, Siple Dome, NEEM). In parallel, a larger disagreement between model and data is observed for some other sites particularly in coastal Antarctica (DE08, Km 200, WAIS Divide)."

Line 564-578: "The introduction of three different activation energies for different temperature ranges leads to changes of the modeled density profiles at high densities (above about 800 kg/m$^3$). A clear improvement is obtained for example at South Pole (Supplementary Figure S6), although the overall impact of using three activation energies remains small.

The incorporation of the impurity effect following the Freitag et al. (2013) parameterization in our model slightly deteriorates the model-data agreement because no specific re-adjustment of model parameters was performed. However the model prediction of the density profiles remains correct although the impurity effect parameterization was developed for a different purpose: simulating density layering (Freitag et al., 2013). This encouraged us to test this simple parameterization in glacial climate conditions.

Overall, in terms of $\sigma_{model-fit}$, only an insignificant improvement (about 3%) is obtained by using the modified model (3 activation energies and implementation of impurity effect) rather than the former Goujon et al. (2003) mechanical scheme. However a systematic improvement is obtained at the six coldest sites."

L505-506: I think this is somewhat of a uneven comparison, you should be comparing _model⬚data to _fit⬚data. _model⬚fit could in theory be smaller than _fit⬚data, as the data has some inherent scatter and layering.

We agree on the fact that the sentences at lines 505-507 are too much of a shortcut. On the other hand, the polynomial fit to the data is close to the bulk density profile that we aim at modelling. The manuscript section between lines 492 and 511 will be revised.

cf change of text corresponding to answer to the problem line 490

L513-514: first stage is important in getting the correct Δage, though.
A Table with Δage results (see above) will be included in the Supplement and briefly commented in the manuscript.

The supplementary Table S2 gives the Δage values and comparisons

L541: Do you make a correction for the convective zone? yes, 2 m (L.580)

L544-546: Do you include the geothermal heat flux? Yes, the geothermal heat flux is included in the model and hence implemented for the calculation of $\delta^{15}N$ from the temperature profile.

L552-553: How do you include the borehole calibration from Dahl-Jensen et al (1998)?
This is not clear

The scenario for temperature evolution in NorthGRIP includes the borehole calibration from Dahl-Jensen et al., (1998) by imposing a 23°C temperature change between LGM and present-day. This corresponds to a temporal slope for the relationship between $\delta^{18}O$ and temperature of 0.3 permil.°C$^{-1}$ over the deglaciation. This slope is thus applied for reconstructing the temperature evolution for the last deglaciation at NorthGRIP in this study. Then, the temperature amplitude for B/A and Y/D is adjusted on the $\delta^{15}N$ data obtained by Kindler et al. (2014) on the NorthGRIP core.

A paragraph explaining the temperature scenario will be added in the supplement.

Some explanation are given in the caption of Supplementary Table S3
Also, in the main text, we added a few words on lines 633-636: "In Greenland (NGRIP, GISP2), the temperature is reconstructed using the $\delta^{18}O_{ice}$ profiles together with indication from borehole temperature measurements (Dahl-Jensen, 1998) and $\delta^{15}N$ data for NGRIP (Kindler et al., 2014) for the quantitative amplitude of abrupt temperature changes."

L592-593: How well do the different models fit the exact timing of the $\delta^{15}N$ increase?
This is set by Δage, so an important test for the models.

Indeed, the different models do not lead to the same calculated Δage as shown in the new table that we will insert in the supplementary materials. We will hence also comment this aspect on Figure 6.

Supplementary Table S4 provide the new table that we were referring to.
A text had also been added on lines 780-784: "Some differences are also observed for the timing of the ◻$^{15}N$ peaks for Bølling-Allerød and end of Younger Dryas at NGRIP when using the different model versions reflecting variations in the simulated Δage (cf Supplementary Table S5); the general agreement with the measured profile is preserved with even a slight improvement of the modelled ◻age with ◻$^{15}N$ constraints with the modified model."

L639-642: Again, I think the link between Accumulation and temperature is not as strong as you suggest here, particularly near the margins where most accumulation is delivered by storms.

We agree that the link between accumulation and temperature is not always so strong and we will nuance this sentence in the revised manuscript. Indeed, near the margin, the link between accumulation and temperature is not straightforward as discussed for example at Law Dome in Landais et al. (QSR, 2006). However, here, we mainly deal with sites that are on the East Antarctic plateau and there is no simple way how we could explain a different evolution for temperature and accumulation rate at Dome C on the long timescale (i.e. on the difference between LGM and present-day).

Line 61-70: "On glacial – interglacial timescales, increasing temperature is associated with increasing snow accumulation. Indeed, the thermodynamic effect dominates when dealing with long term averages (several thousands of years), even if accumulation and temperature are not always correlated on millennial and centennial timescale in polar regions, especially in coastal areas (e.g. Fudge et al., 2016; Altnau et al., 2014). As a consequence, when comparing LGM and Holocene averages, we observe for all available ice cores covering the last deglaciation increases in both accumulation and temperature. In the firn densification model, both effects partially compensate each other, with the temperature effect being dominant in the current densification models for the LID simulation over glacial – interglacial transitions in deep drilling sites of the East Antarctic plateau, hence leading to the modelled LID decrease."

L666: "densification rates" (add "s") ok

L682: compatible with the data "except at Talos Dome". Table 3: please be more clear what all the numbers are. I assume these are Q1, Q2, Q3 etc, but this is not very clear. ok

L706-710: Please rewrite this sentence, it is really hard to follow.

When using the parameterization of Table 1 ("new model"), Figure 7 shows strong improvement of the simulation of the $\delta15N$ difference between EH and LGM. Indeed, the modeled EH-LGM difference now has the correct sign at very cold sites of East Antarctica (Figure 7) when compared to $\delta15N$ measurements.

This sentence was inserted in the new text

L725: change "less good" to "worse" ok

L734-736: Could this be because the uncertainty in the input temperature does not include the possibility that the LGM was much warmer than the optimal scenario?

This is unfortunately not sufficient to explain the mismatch. Even if significantly higher LGM temperature (-6°C instead of -9°C) without any change in accumulation rate would decrease the $\delta^{15}N$ at LGM at EDML down to the level of the accu_minus scenario (Figure S9). This is unfortunately not enough to reconcile model and data on the amplitude of the change of $\delta^{15}N$ between the LGM and present-day. Moreover, note that such lack of correlation between temperature and accumulation rate changes can be observed in coastal sites but is again very unlikely in the East Antarctic plateau.

L756-757: Ca from volcanic events?

We do not know if the very fast peaks observed on the calcium concentrations correspond to volcanic eruptions, on the other hand we know that they do not affect the $\delta^{15}$N profiles.

L817: Is your new modeled EDC Δage consistent with the work of Parrenin et al.
(2013)?
Yes, our study is in line with the work of Parrenin et al. (2012 and 2013). This will be specified in the new manuscript and was actually one of the aims of the tuning of activation energy values to temperature.

Line 955-963 (conclusion): "Our finding is however associated with several limitations so that this new model does not propose a definite re-evaluation of the formulation of the activation energy but better proposes some ways to be further tested and explored to improve firn densification models especially for applications on paleoclimate reconstructions. Our approach remains empirical and we could not identify separately the different mechanisms involved. The problem of $\delta^{15}$N data-model mismatch in low temperature and accumulation rate sites of East Antarctica is thus not definitively solved. Still, we showed that revising the temperature and impurity dependence of firn densification rate can potentially strongly reduce the $\delta^{15}$N data-model mismatch and proposed preliminary parameterizations easy to implement in any firn densification model."

Anonymous Referee #2

We thank the reviewer for the comments that are indeed helpful to convey a comprehensive message. We have tried to take int account all comments as detailed below.

General:

The authors state that they have improved the LGGE firn densification model based on physical mechanisms. They argue with a better agreement between modelled and measured $\delta^{15}N$ data. But the physical arguments stand on shaky ground and the better agreement for some sites are opposed by significant lesser agreement of the model data at other sites. A general comment of caution regarding the physical approach: Though the concept of Arzt, based on monosized spheres, which deviates substantially from the physical reality, produces reasonable firn density profiles, they are not really better than those of empirical parametric models. The reason is the rigidity of the 'physical' model and it is not be surprising that an empirical approach with more free parameters, as e.g. in the Pimienta-model, may even better catch the reality! The authors now introduce rather arbitrarily two additional Arrhenius-type mechanisms. Thereby they can readily simulate a higher densification rate at very low temperatures. A corroboration of the new model by the better agreement with a small glacial delta age (or delta depth, i.e. shallower glacial firn depth) by Parrenin (2012) is unjustified because this agreement was exactly the purpose of tuning the model.

The purpose of this paper is to evaluate the LGGE model of firn densification by comparison with data in modern and past climate conditions, with a focus on the major problem of the too small densification rate at very low temperature in East Antarctica highlighted in Capron et al. (2013) and earlier studies. We did not perform a major revision of the model but tested the effect of relatively simple model changes (e.g. the activation energy and dust tests can easily be adapted to other models).

We agree that our parameterization has been chosen to better match the $\delta^{15}N$ data and hence the $\Delta$age estimate by Parrenin et al. (2012) at Dome C. This was already present in the previous manuscript and will be highlighted more clearly in the revised version. Our new parameterization should also not affect too much the agreement observed between model and data in higher temperature and accumulation rate conditions which is not trivial. We thus investigated to which extent the firnification model invoking different mechanism for firn densification is able to reproduce the $\delta^{15}N$ evolution over deglaciations in low accumulation rate sites of East Antarctica without degrading the good agreement observed with the old version of the model between measured and modeled firn density profiles and between modeled and measured $\delta^{15}N$ evolution over the last deglaciation at sites with higher accumulation rates.

The only way to reconcile our model with $\delta^{15}N$ data by invoking a change of activation energy with temperature is to have a negligible influence of temperature on the densification rate below -50°C. Several reasons for this effect can be invoked. We proposed here an interpretation with different activation energies for firn densification based on previous studies showing different densification mechanisms over different temperatures. Still, our proposed interpretation is not definitive for explaining firn densification over a large range of temperature. In the revised manuscript, we will make clear that the association of our three activation energies with three precise physical

mechanisms is not proved. Indeed, while several mechanisms have been highlighted for the densification of ice over several temperature ranges, there is no unambiguous attribution of a particular mechanism to a particular temperature. The determination of the activation energies for our model has an empirical basis as for previous studies. We will further emphasize this aspect in the revised version, especially when discussing Figure 2. We will also state that the effect of temperature could be misrepresented in our model by other ways than the value of activation energy opening the ways to other studies.

Line 333-349: "Following these arguments and despite the lack of experimental constraints to test this assumption, we propose a new parameterization of the activation energy in the LGGE firn densification model which increases the firn densification rate at low temperatures. We have thus enabled introduction of three adjusted activation energies as proposed in Table 1 and Figure 2. We have replaced the creep parameter in Equation (3) by:

$$A = A_0 \times \left( a_1 \times e^{\frac{-Q_1}{RT}} + a_2 \times e^{\frac{-Q_2}{RT}} + a_3 \times e^{\frac{-Q_3}{RT}} \right) \qquad (7)$$

We have chosen a minimal number of mechanisms (3) for simplicity in the following but the conclusions of our work would not be affected by a choice of more mechanisms.

[Figure]

- Close to melting temperature: mass transfer by diffusion (potential mechanism for high temperature) (1) mechanism 1 associated with activation energy $Q_1$

- Low temperature: lattice diffusion (classical mechanism) (2) mechanism 2 associated with activation energy $Q_2$

- Very low temperature : boundary diffusion from grain boundary (potential mechanism for low temperature) (3) mechanism 3 associated with activation energy $Q_3$

*Figure 2: Different sintering mechanisms of snow for different temperatures proposed by analogy with the hot ceramic sintering (inspired by Figure 1 in Ashby, 1974). Note that more sintering mechanisms can be found in the literature and the attributions of 3 different mechanisms for the firn densification model is only a working hypothesis here."*

In addition, several sentences have been removed all along Section 2.2

The approach regarding other transport mechanisms involved in sintering is not convincing. Why should surface diffusion explain the higher densification rates at low temperatures? First, surface diffusion itself does not lead to densification and second also indirect effects will most likely not favor densification. Surface diffusion increases neck diameters and thus decreases pressure at contact area, which decreases creep and in addition it increases curvature radii that decrease the generation of lattice vacancies, and thus decreasing lattice diffusion. Q3, the activation energy applied for surface diffusion seems unrealistically low. Higher values, between 30 and 50 kJ, have been reported.

We agree with the referee on the fact that surface diffusion does not directly lead to densification. We went back to Ashby 1974 (Acta Metallurgica, vol.22, pp.275-289) and actually the dominant mechanism at low temperature should rather be boundary diffusion from grain boundaries which is a mechanism enabling densification. We are not aware of activation energy values associated with

this mechanism in firn. As stated above, we will make clear that the association of our three activation energies with three precise physical mechanisms is not proved.

Lines 329-330 have been modified "lattice diffusion from the surface of the grains and/or boundary diffusion from grain boundaries should be favoured (Ashby, 1974)."

Figure 2 has also been modified for the very low temperature mechanism proposed.

As for the measured values of activation energy, Anonymous Referee#1 provided two references for the activation energy of surface diffusion in the range 14-38 kJ/mol and we will include in the revised manuscript a test showing that using 15kJ/mol leads to intermediate results between the "old" and "new" model results.

Test C is included in Figure 7, Figure 8 and Table 3.

In the main text, discussion on lines 825-832 has been inserted: "Test C has been designed so that the activation energy at low temperature corresponds to estimates of activation energy for ice surface diffusion (Jung et al., 2004; Nie et al., 2009), a mechanism that is expected to be important at low temperature (Ashby, 1974). Using such a parameterization leads to a fair agreement between the modelled and the measured $\delta^{15}$N change over the last deglaciation for the different sites. At Dome C, the correct sign for the $\delta^{15}$N evolution between LGM and the Holocene is predicted by the model. However, the modelled $\delta^{15}$N increase is still too small compared to the data and the $\delta^{15}$N calculated by the "new model". This is probably due to a too high creep parameter at low temperature."

Considering the influence of dust on densification is interesting but does not substantially contribute to solving the discrepancy between model and date, because the densification enhancement by dust leads for too many sites to a deterioration of the modelled densities.

The tuning of Q1, Q2, Q3 and associated a1, a2, a3 has been done without dust influence. As a conclusion and as noted by Referee 2 (comment on lines 483+497), the implementation of dust necessarily slightly deteriorates the model – data agreement. This will be clarified in the revised version when discussing the dust influence addition.

We will also clarify the fact that we do not expect the Freitag et al. (2013) parameterization to be properly tuned for simulating the variations of firn thickness as it was not designed for this purpose. One way to improve the constraints on the problem (how does dust influence the glacial firn depth ?) is to study other deglaciations, where the dust increase and the temperature increase are not synchronous.

Line 408-413: "The values of $a_i$ and $Q_i$ were not readjusted after the implementation of impurity effects to avoid adding tuning parameters. Still, because the large range of calcium concentrations encountered in past climate conditions has a strong impact on model results, this may be a solution to reduce the model-data mismatch. This is explored in Section 3 through a sensitivity test D. In the same section, we will also propose a modification of the Freitag parameterization using thresholds to reduce the model-data mismatch."

To further explore the dust influence for paleoclimate studies, we have implemented of Test D (Figure 7 and Table 3) + discussion line 874-887: "The model – data mismatch observed when

incorporating the dust effect may be partially due to the fact that we did not readjust $a_i$ and $Q_i$ after implementation of the impurity effect. To explore this possibility, sensitivity test D has been designed with a re-parameterization of the $a_i$ and $Q_i$ values after implementation of the impurity effect. To do so, we calculated the optimal creep parameter A for each mean EH and LGM condition at each site, and adjusted sequentially $a_3$, $a_2$, $a_1$, $Q_3$, $Q_2$, and $Q_1$ to minimize the model-data mismatch. Only $a_3$, $a_2$ and $Q_3$ needed adjustments, and their values can be found in Table 3. We did not perform the adjustment on modern density profiles, because these are only weakly sensitive to the dust parameterization, $Ca^{2+}$ concentrations being low.

Impurity concentration is very high at NGRIP during the glacial period. As a consequence, even if our new parameterization of $a_i$ and $Q_i$ (new model) properly reproduces the Greenland $\delta^{15}N$ level at LGM, this glacial modelled Greenland $\delta^{15}N$ level is too low when including the impurity effect. The re-parameterization of $a_i$ and $Q_i$ proposed as sensitivity test D enables an improvement of the agreement between model and data for glacial $\delta^{15}N$ at WAIS-Divide, maintain the results at Dome-C and EDML, but can still not produce reasonable results at NGRIP (Figure 7)."

We have also proposed a revised parameterization of the Freitag parameterization including additional threshold effects (Supplementary Figure S10 and Supplementary Text S2) and corresponding discussion on lines 895-897 +:
"To further improve the model – data agreement with the dust parameterization, a possibility is to add simple thresholds on a minimum and maximum effect of calcium as proposed in supplementary material (Supplementary Text S2 and Figure S10)."

The mentioned possibility of saturation of the densification enhancement by
dust at high concentration would only work for Greenland but not for WAIS divide.

Indeed, the « dust saturation hypothesis » cannot reconcile the Holocene firn thickness at NGRIP and WAIS-Divide, which are nearly identical with 10 times more Ca at NGRIP compared to WDC. Another possibility is that the impact of dust depends on the densification mechanism, and is much more important at cold temperature.

cf previous answer

My criticism shall not disesteem the huge work accomplished for improving the cali-
bration of the model for modern firn densities that is also presented in this paper. This
calibration with new improved firn density profiles certainly leads on average to slight
optimization of the model parameters.
However, a better fit to glacial firn depths has only been achieved by a direct tuning of
the creep factor at low temperatures. This should be clearly communicated as such.
And as the authors mention in the supplement this can be achieved with any other of
the common densification models.

As mentioned in the previous manuscript, the choice of the very low value for Q3 is indeed designated for increasing firn densification at low temperature. This tuning of the creep factor will be stated more clearly earlier in the manuscript (section 2.2)

Line 22-23: "we apply a dependency of the creep factor on temperature and impurities in the firn densification rate calculation."

Line 24-27: "We show that both the new temperature parameterization and the influence of impurities contribute to the increased agreement between modelled and measured $\delta^{15}N$ evolution during the last deglaciation at sites with low temperature and low accumulation rate, such as Dome C or Vostok"

Line 333-363: "Following these arguments and despite the lack of experimental constraints to test this assumption, we propose a new parameterization of the activation energy in the LGGE firn densification model which increases the firn densification rate at low temperatures. We have thus enabled introduction of three adjusted activation energies as proposed in Table 1 and Figure 2. We have replaced the creep parameter in Equation (3) by:

$$A = A_0 \times \left( a_1 \times e^{\frac{-Q_1}{RT}} + a_2 \times e^{\frac{-Q_2}{RT}} + a_3 \times e^{\frac{-Q_3}{RT}} \right) \tag{7}$$

We have chosen a minimal number of mechanisms (3) for simplicity in the following but the conclusions of our work would not be affected by a choice of more mechanisms.

[Figure]

- Close to melting temperature: mass transfer by diffusion (potential mechanism for high temperature)
  (1) mechanism 1 associated with activation energy $Q_1$

- Low temperature: lattice diffusion (classical mechanism)
  (2) mechanism 2 associated with activation energy $Q_2$

- Very low temperature : boundary diffusion from grain boundary (potential mechanism for low temperature)
  (3) mechanism 3 associated with activation energy $Q_3$

*Figure 2: Different sintering mechanisms of snow for different temperatures proposed by analogy with the hot ceramic sintering (inspired by Figure 1 in Ashby, 1974). Note that more sintering mechanisms can be found in the literature and the attributions of 3 different mechanisms for the firn densification model is only a working hypothesis here.*

When building the new parameterization of the activation energy (Equation 7), the determination of $Q_1$, $Q_2$ and $Q_3$ on the one side and $a_1$, $a_2$ and $a_3$ on the other side are not independent from each other. We first determine three temperature ranges corresponding to the dominant mechanisms, then we attribute values to the activation energies $Q_1$, $Q_2$ and $Q_3$. The coefficients $a_1$, $a_2$ and $a_3$ are finally adjusted to produce the expected evolution of the creep parameter with temperature, to best reproduce $\delta^{15}N$ evolution over deglaciations (Section 3.2) and respect the firn density profiles available (Section 3.1).

Hundreds of sensitivity tests have been performed using a strategy based on dichotomy to reduce the mismatch between modeled and data. The constraint of keeping a correct agreement of model results with present day density profiles and for the last deglaciation at warm sites strongly reduces the possible choices of $a_i$ and $Q_i$ (Section 3). The best value obtained for $Q_3$ is lower than published values for surface or boundary diffusion but is necessary to reproduce the deglaciation at cold East Antarctic Sites. Sensitivity test C will illustrate the effect of using a higher value."

Conclusion has also been largely revised in this direction

In my view, this lengthy paper clearly raises false expectations. The paper could be organized such that in a first part the improvement of the parameter calibration on modern density profiles are presented and in a second part one could investigate how the creep factor needs to be tuned in order to better simulate the glacial data for the different ice cores, with and without dust enhancement. The diverging results would readily show that so far no unified model can simulate the existing range of data.

>> We will implement a new (short) section 3.1 that present the agreement between modeled and measured firn density profiles using the revised version of the LGGE model with only one activation energy (i.e. corresponding to modifications depicted in section 2.1). To show the influence of this revisions compared to the old version, Figure 5 will display an additional vertical bar for each site.

Then, we will present our proposition that with a parameterization with three activation energies we are able to better fit $\delta^{15}$N profiles at low accumulation sites in Antarctica. The limitations of this assumption presented at the beginning of the answer to Ref 2 will be clearly stated. We will then show that the implementation of 3 activation energies does not significantly deteriorate the agreement between modeled and measured firn density profiles but that some divergences persist for paleoclimatic simulations especially when adding the dust influence which call for future studies for investigating the influence of both temperature and dust on firn densification and lock-in depth prediction.

The whole Section 3 has been reorganized following the suggestion of the reviewer.

Some specific comments:
L. 2: title "constrains" -> constraints ok

L. 23: " we introduce a dependency of the activation energy to temperature and impurities in the firn densification rate calculation". It is rather a temperature dependent creep factor. The authors 'apply' the impurity dependence, it was 'introduced' by Freitag et al.

Yes, this will be rewritten in the manuscript.

Line 22-24: "we apply a dependency of the creep factor on temperature and impurities in the firn densification rate calculation. The temperature influence intends to reflect the dominance of different mechanisms for firn compaction at different temperatures."

L. 79: " questionable when used outside of its range of calibration". Not only then as in case of EDML.

>> It is true that EDML is difficult to reconcile with the model. We have shown in Landais et al. (2006) that it can be reconciled with the model only if accounting for large uncertainties in past accumulation rate and a reasonable convective zone (10 m) (Landais et al., 2006). This will be specified in the revised manuscript.

Line 146-147: "This hypothesis can explain the mismatch between modelled and measured $\delta^{15}$N at EDML during glacial period by invoking a 10 m convective zone (Landais et al., 2006)."

L. 186: Is A0 a constant? Value?

Yes, $A_0 = 7.89 \times 10^{-15}$ Pa$^{-3}$.s$^{-1}$, this is specified in the new manuscript. It is the same value as in Equation A5 of Goujon et al. (2003)

L. 210-228: This 'bending into shape' demonstrates again that a more parametric approach can be closer to reality than a 'pure physical' one.

True, but aiming to use physical formulation is perhaps safer when we venture out of the calibrated range. Still, our approach is very much empirical at this stage, since the creep constant and other model parameters such as $D_0$ have been calibrated empirically. We will make clearer the fact that our approach is not a 'pure physical' one: we will add a sentence at line 86 clarifying the fact that the simplified physical mechanisms in our model include adjusted parameters.

The section 2.2 has be rewritten (cf also comments above)

L. 245 (eq. 6) Is 0.1 bar the pressure at 2 m depth (L. 213)? This should be clarified in the text..

0.1 bar is a rough approximation of the pressure at two to three meters depth, which depends on the density profile of the overlying snow.
The sentence at lines 211-213 will be replaced with:
Indeed, since the model is not able to represent the metamorphism in the first two meters, we impose a constant pressure of 0.1 bar (see Equation 6), which is an approximation of the pressure at 2-3 meters depth. It results in a nearly constant densification rate in the top 2-3 meters rather than a constant density in the top 2 meters.

Line 245-247: "we impose a constant pressure of 0.1 Bar (see Equation 6), which is an approximation of the pressure at 2-3 m depth. It results in a nearly constant densification rate in the top 2-3 m rather than a constant density in the top 2 meters. "

L. 305: " three different mechanisms highlighted above" There are more than 3 mechanisms mentioned above. Table 1, Fig. 2 may not be above.

Replaced by "three different mechanisms highlighted in Table 1 and Figure 2"

L. 321, 824: "surface lattice diffusion" Is this term correct for 'surface diffusion'?

This has been changed to "boundary diffusion from grain boundary" following Ashby (1974)

L. 358: "assuming that the impurity effect is the same for all mechanisms". This seems very unlikely!

Line 390-392: "even though this assumption is probably not correct for the vapor diffusion process. Note however that this will not affect the applications discussed below since vapor diffusion is only important for warm sites."

L. 362,3: "f1" subscript 1 ok

L. 403: "A subscript c" is a strange notation for accumulation rate. Define abbreviation

in text. This parameterization is surprising. Of course temperature and accumulation rate are strongly correlated, so we expect an corresponding correlation with accumulation rate. But for two sites with equal accumulation rate but different temperature we don't expect the same LID as densification is strongly temperature dependent. So this parameterization may be justified for most conditions but one must be careful applying it in general and during glacial conditions.

"A subscript c" A is more commonly used but here, we wish to avoid confusion with $A_i$ and $a_i$ used in Equation 7 and Section 2.2. We will use "Ac" instead of "$A_c$" in the revised manuscript.
In the previous model version (Goujon et al., 2003), the LID definition was related to a value of the closed/total porosity ratio adjusted for each drilling site but time independent (see lines 387-389). This dependency on the geographic location with independence from climate is not self-consistent. This is why we tested another LID definition. Note that we do not directly parameterize the lock-in depth but only allow moderate variations of the lock-in density. This will be clarified in the revised manuscript by using the notation "$\rho_{LI}$" instead of "$\rho_{LID}$" and concluding on the small impact of the definition change.

This is done in the new version

L. 427: Degree of polynomial fit?

It is site dependent primarily because the number of density measurements is highly site dependent. The polynomial fits used will be provided in the supplement of the revised manuscript.

L. 405,6: "This parameterization leads to a much better agreement of the modelled LID with $\delta^{15}N$ measured at the available firn sampling sites than when using the outputs of the old model" This has nothing do with the model. It is just a different parameterization. The better agreement would apply for the old model as well.

This will be corrected in the revised manuscript.

Line 454-456: "This parameterization leads to $\rho_{LI}$ variations in the range 780-840 kg/m$^3$ (Supplementary Figure S5) and a much better agreement between the modelled LID and $\delta^{15}N$ measured in firn samples at available sites than when using a fixed closed / total porosity ratio."

L. 456: "rough indicator of data quality" This seems a daring assumption, as natural variability might be in the same order of magnitude.

>> Completed as "rough indicator of data quality or estimate of natural variability"

L. 471: explain "traction constraint"

In the long term, mechanical strain conditions result in different crystal orientations in ice cores drilled near dome summits (where uniaxial compression along the vertical direction dominates) and along flow lines with a component of horizontal tension (see e.g. Montagnat et al., 2014, www.the-cryosphere.net/8/1129/2014/, and references therein). Moreover Vostok is located above a lake and the basal gliding results in a low shear stress, thus the crystal orientations likely results from a deformation in tension along the horizontal (Lipenkov et al., 1989). In the manuscript, we will replace "traction constraint" with "horizontal tension".

 "One possible reason is the very different flow regimes of the two sites, one being at a Dome summit, and the other on a flow line and subject to a horizontal tension (Lipenkov et al., 1989)."

L. 483 + 497: ?? " the original parameterization of Freitag et al. (2013) always remain in reasonable agreement with the data . . ... the incorporation of the impurity effects following the Freitag et al. (2013) parameterization in our model most often deteriorates the model-data agreement".

This is a good point, thanks. As some model tuning is performed prior to the incorporation of the impurity effect but not after, an overall slight but not significant degradation of the model results is obtained when adding the Freitag et al. (2013) parameterization. The inconsistency will be corrected in the revised manuscript.

Line 568-573: "The incorporation of the impurity effect following the Freitag et al. (2013) parameterization in our model slightly deteriorates the model-data agreement because no specific re-adjustment of model parameters was performed. However the model prediction of the density profiles remains correct although the impurity effect parameterization was developed for a different purpose: simulating density layering (Freitag et al., 2013). This encouraged us to test this simple parameterization in glacial climate conditions."

L. 509: "This effect is due to a densification rate that is too high in the first stage, and this formulation is not affected by the new temperature sensitivity." If this is a general feature, why is it not accounted for correctly?

Figure 3 in Alley (1987) indicates that the modeled densification rates tend to be somewhat too low at low densities and somewhat too high at higher densities. This induces an "S" shape of the density profile that is also visible in our model results. Correcting this bias would require an in-depth revision of the Alley (1987) mechanism. As we say at lines 513-514, the first stage of densification is not crucial for predicting the LID: the modifications of the Alley (1987) mechanism did not improve significantly the model results. In the revised manuscript, Figure 5 will include results with the "new model" and a single value of activation energy. This will further illustrate the small impact of the modifications in the first stage of densification.

Figure 5 has been modified

L. 613: "This observation questions the possible presence of a convective zone and/or .." Is the presence questioned or the constancy of the convective zone?

>> The **presence** of a significant convective zone is questioned for the LGM -> We will precise that it is clearly the presence of the convective zone at LGM that is questioned.

done

L. 689: "Evolution" It is not 'evolution' but 'dependence'. Fig. 8b: Vertical axis title: not Log(A) is shown but A on a log-scale. Fig. 8 could be probably presented in one single graph with A on a log-scale facilitating the comparison between the different temperature ranges.

The caption of Fig. 8 and axis of Fig.8b will be modified.
We think that it is important to keep the 3 figures. They are useful to see the different scenarios for the different temperatures and especially the addition of figure 8c (by comparison with only figure 8b) permits to observe the intersection points between the different scenarios.
We have tried different representations but did not find a better way to present the results and the associated discussion.

L. 709: "an inversion of the $\delta^{15}N$ difference" probably better: "a correct sign of the $\delta^{15}N$ difference". ok

L. 711: I strongly question the value of tests A to C. The chosen parameters are very arbitrary. I don't see what the authors want to show us, except that some parameters are better here and worse there, which is rather trivial.

>> We agree that test C was not very useful in the previous manuscript., We propose to replace the original test C by a sensitivity test run with Q3=15kJ/mol a value close to the lower bound of the observed range for activation energies (Jung et al., 2004, reference provided by Referee#1). Using 15 kJ/mol leads to intermediate results between the "old" and "new" model results. The only way to reconcile our model with $\delta^{15}N$ data by invoking a change of activation energy with temperature is to have a negligible influence of temperature on the densification rate below -50°C and hence to decrease Q3 largely below 15 kJ/mol (1.5 kJ/mol). This will be discussed in the revised manuscript.

[Figure]

Figure 8 has been changed as well as the associated discussion. See for example the implementation of test C (based on experimental values for the activation energy as provided by reviewer 1) on lines 825-835: "Test C has been designed so that the activation energy at low temperature corresponds to estimates of activation energy for ice surface diffusion (Jung et al., 2004; Nie et al., 2009), a mechanism that is expected to be important at low temperature (Ashby, 1974). Using such a parameterization leads to a fair agreement between the modelled and the measured $\delta^{15}$N change over the last deglaciation for the different sites. At Dome C, the correct sign for the $\delta^{15}$N evolution between LGM and the Holocene is predicted by the model. However, the modelled $\delta^{15}$N increase is still too small compared to the data and the $\delta^{15}$N calculated by the "new model". This is probably due to a too high creep parameter at low temperature. Summarizing, the best agreement between data and model for Dome C is obtained for the parameters given on Table 1: the creep parameter of "new model" flattens below -50°C and is thus not very different for the LGM or the EH at Dome C."

L. 765: ".. instead of the Herron and Langway model.." -> instead of the parameters for the Herron and Langway model. ok

L. 781: "may act on deformation in opposite way" Why? Please explain.

>> "In particular, the solubility of dust particles, and their position inside or at the grain boundaries may act on deformation in opposite way" – the sentence is indeed not clear

It has been modified as:
"Dust particles do not always influence densification on the same way: dissolved particles soften firn and ice while the softening or hardening effect of non-dissolved impurities is less clear (Fujita et al., 2016; Alley et al., 1987)"

L. 797: " an up-to-date version " This is an empty phrase.
The phrasing will be modified

Line 924: "In this study, we have presented a revision of the LGGE firn densification model."

L. 787: " the new parameterization of the creep parameter preserves good agreement between the old model outputs and data at sites that were already well simulated"
Because the creep parameter is kept +- the same, so it is trivial.
The phrasing will be modified

Line 913-915: "To sum up, the new parameterization of the creep parameter has been designed to preserve good agreement between the old model outputs and data at sites that were already well simulated (WAIS-Divide, NGRIP, Talos Dome)."

L. 815: " This result is in agreement with the recent low delta age estimate by Parrenin et al. (2012) over the deglaciation at Dome C". No surprise, because it has been forced to agree.

>> This is true and will be specified in the conclusion

"The latest results are in good agreement with the recent determination of $\Delta$age within the AICC2012 timescale: 3920 years for EDC 21 ka and 5100 years for Vostok 21 ka. This is not unexpected since the EDC LID in the construction of the AICC2012 timescale is deduced from the EDC $\delta^{15}N$ scenario".

L. 850: "Ore" -> Core ok

[revised manuscript text omitted]
), new version with impurity following Freitag parameterization (purple) and new version with impurity (blue) using a parameterization with threshold for high and low values of the calcium concentration) of the LGGE model for WAIS-Divide, NGRIP, EDML and Dome C. Blue boxes for each sites indicate the periods over which the $\delta^{15}N$ average for the LGM and EH have been estimated for the calculation of the amplitude of the $\delta^{15}N$ change over the deglaciation*

| Sites | Location (Latitude ; Longitude) | Temperature (°C) | Accumulation (cm.w.eq.yr) | [Ca$^{2+}$] (ng/g) | Surface density (kg/m$^3$) | $\sigma_{fit\text{-}data}$ (kg/m$^3$) | $\sigma_{old\ model\text{-}fit}$ (kg/m$^3$) | $\sigma_{new\ model\text{-}fit}$ (kg/m$^3$) | $\sigma_{new\ model\ with\ dust\text{-}fit}$ (kg/m$^3$) |
|---|---|---|---|---|---|---|---|---|---|
| Dye 3 [1] | 65°11'N ; 43°50'W | -18.0 | 50.0 | X | 357 | 13.8 | 32.7 | **30.8** | X |
| DE08 [2] | 66°43'19"S ; 113°11'58"E | -19.0 | 120.0 | X | 384.2 | 10.3 | **20.9** | 29.8 | X |
| Km105 [4] | 67°58'S ; 93°70'E | -24.5 | 34.1 | X | 460.5 | 14.0 | **10.7** | 21.1 | X |
| Site 2 [5] | 76°59'N ; 56°04'W | -25.0 | 36.0 | X | 350.1 | 5.8 | **18.3** | 19.2 | X |
| Siple Dome [3] | 81°39'3"S ; 148°47'66"W | -25.4 | 10.0 | 8.0 | 382.7 | 8.9 | 29.4 | **13.2** | 19.5 |
| D-47 [6] | 67°23'S ; 138°43'E | -25.4 | 25.0 | X | 377.4 | 12.5 | **15.9** | 21.8 | X |
| Byrd [7] | 80°S ; 120°W | -28.0 | 15.6 | X | 347.1 | 3.8 | **15.9** | 18.6 | X |
| NEEM [8] | 77.45°N ; 51.06°W | -28.9 | 20.0 | 7.4 | 307.2 | 8.0 | 27.2 | **13.5** | 15.8 |
| Crête (site A) [9] | 70°38'5.68"N ; 324°10'48"E | -29.0 | 28.2 | X | 321.7 | 7.0 | 18.7 | **14.0** | X |
| Km200 [10] | 68°15'S ; 94°05'E | -30.5 | 28.7 | X | 454.4 | 10.5 | **21.8** | 37.7 | X |
| WAIS divide [11] | 79°28'S ; 112°05'W | -31.0 | 20.2 | 1.58 | 393.7 | 8.7 | **18.2** | 21.3 | 35.1 |
| Ngrip [12] | 75°10'N ; 42°32'W | -31.2 | 17.5 | 10.0 | 299.9 | 6.9 | 22.2 | **9.9** | 13.4 |
| Grip [13] | 72°34'N ; 37°37'W | -31.7 | 21.0 | 7.8 | 367.0 | 6.7 | 18.9 | 10.3 | **8.8** |
| B29 [14] | 76°00'N ; 43°29'E | -31.6 | 15.3 | 9.2 | 325 | 10.1 | 22.7 | **7.2** | 10.2 |
| Mizuho [15] | 70°41'53"S ; 44°19'54"E | -34.5 | 7.0 | X | 421 | 9.8 | **50.0** | 66.5 | X |
| Talos Dome [16] | 72°49'S ; 159°11'E | -41.0 | 8.0 | 4.0 | 315.3 | 12.1 | **29.6** | 46.0 | 51.9 |

| Site | Location | | | | | | | | |
|---|---|---|---|---|---|---|---|---|---|
| B32 [17] | 75°00'S ; 0°00'E | -44.1 | 6.1 | 1.7 | 334.5 | 6.2 | 15.6 | **13.3** | 28.7 |
| EDML [18] | 75°S ; 0'04°E | -44.1 | 6.4 | 3.0 | 305.9 | 5.3 | 19.4 | **16.9** | 23.6 |
| South Pole [19] | 90°S | -49.0 | 6.37 | 2.0 | 394.4 | 6.4 | 35.5 | 26.7 | **15.2** |
| Dôme C [20] | 75°06'S ; 123°21'E | -55.0 | 2.5 | 1.8 | 309.2 | 6.2 | 15.3 | **11.0** | 11.0 |
| Vostok [21] | 78°28'S ; 106°48'E | -57.0 | 2.2 | 1.6 | 326.5 | 8.0 | 28.0 | 23.5 | **23.0** |
| Dome A [22] | 80°22'01.6"S 77°22'22.3E | -58.3 | 2.3 | X | 374 | 9.8 | 32 | **13.4** | X |

References below were used for the following data: [a] location, [b] density, [c] temperature, [d] accumulation, [e] calcium concentration

Best efforts were made to find information about the methodologies used for density measurements. The following Greek letters are used to indicate the method used, the use of several letters for the same site implies that several data series were used.

α: volume and weight measurements on whole cores or bags. The precision of such measurements is dependent on the regularity of the core shape.

β: volume and weight measurements in firn, and high precision hydrostatic weighing measurements in ice

γ: volume and weight measurements on machined samples (regular volume, samples are often small)

δ: gamma ray beam attenuation through the ice core (very high resolution)

ε: camera assisted volume measurements, and weight measurements (high resolution)

[1] [a, c, d] Robin, 1983; [b] http://gcmd.nasa.gov/r/d/LSSU_and PSU_Firn_data Spencer et al., 2001

[2] [a, b, c, d] Etheridge and Wookey, 1989; [b] Arnaud et al., 1998, 2000, γ density measurement method

[3] [a, b] https: //nsidc.org/data/waiscores/corec.html, [c] Butler et al., 1999; Jones et al., 2014; Kreutz et al., 1999, 2000, α density measurement method

[4] [a, b, c, d] Salamatin et al., 2009, γ density measurement method

[5] [a, c,] Robin, 1983; [b] http://gcmd.nasa.gov/r/d/LSSU_PSU_Firn_data Spencer et al., 2001 originally from Langway, 1967 with β density measurement method

[6] [a, b, c, d] Arnaud et al., 1998 with β density measurement method

[7] [a, b, c, d] http://gcmd.nasa.gov/r/d/LSSU_PSU_Firn_data Spencer et al., 2001, originally from Gow, 1968 with β density measurement method

[8] [a, b, c] Buizert et al., 2012; [b] Steen-Larsen et al., 2011 ; [e] Gfeller et al., 2014, α density measurement method

[9] [a, b, c, d] http://gcmd.nasa.gov/r/d/LSSU_PSU_Firn_data and Spencer et al., 2001 ; [b] originally from Clausen et al., 1988

[10] [a, b, c, d] Salamatin et al., 2009; [b] Arnaud et al., 1998, γ density measurement method

[11] [a, b, c, d] Fitzpatrick et al., 2014 ; [e] Cole-Dai et al., 2013

[12] [a, c, d] Ngrip community members, 2004, [b] H.C. Steen-Larsen Pers. Comm., [e] Svensson pers. Comm., 2016, α density measurement method

[13] [a, b, c, d] http://gcmd.nasa.gov/r/d/LSSU_PSU_Firn_data and Schwander et al., 1997; [e] Iizuka et al., 2008, α and γ density measurement methods

[14] Freitag et al., 2013; Hörhold et al., 2011, δ density measurement method

[15] a, c, d (Nishio et al., 1979) ; b (Narita and Maeno, 1978), γ density measurement method

[16] a, c www.taldice.org/project/site ; b www.taldice.org/data (data from F. Parrenin) ; d Stenni et al., 2002 e Schüpbach et al., 2013

[17] Freitag et al., 2013; Hörhold et al., 2011, δ density measurement method

[18] a, b Kipfstuhl et al., 2009 ; c Freitag et al., 2013; d Oerter et al., 2004; e Fischer et al., 2007, α density measurement method

[19] a, b, c http://gcmd.nasa.gov/r/d/LSSU_PSU_Firn_data and Spencer et al., 2001 ; d Mosley-Thompson et al., 1995; e Ferris et al., 2011

[20] a, d Gautier et al., 2016, b R. Mulvaney pers. com. and Leduc-Leballeur et al., 2015; c Arnaud et al., 2000 ; e Lambert et al., 2012, α and ε density measurement methods

[21] a, c, d Arnaud et al., 2000 ; b http://gcmd.nasa.gov/r/d/LSSU_PSU_Firn_data and Spencer et al., 2001 and J.-M. Barnola, unpublished (using γ density measurement method); e De Angelis et al., 1997

[22]a, b, c Cunde et al., 2008 ; Cunde pers. com. 2016 ; d: Hou et al., 2007 and Ding et al., 2016, α density measurement method

*Supplementary Table S1: Standard deviation between modeled and measured density profiles for 22 polar sites, for the old LGGE model and the new LGGE model (with three different activation energies in the firn densification module noted "new model" and with three different activation energies depending on Ca$^{2+}$ concentration in the firn densification module noted "new model with dust"). The values in bold indicate the lowest standard deviation between modeled and fitted density profiles for each site.*

| Δage | Data | Old | New | New + dust |
|---|---|---|---|---|
| Dye 3 | 78 | **68.4** | 61.8 | - |
| DE08 | 35.5 | **36.8** | 31.3 | - |
| km105 | 104.4 | 111.7 | **106.4** | - |
| Site2 | 112.2 | **109.2** | 103.3 | - |
| Siple Dome | 329 | 287.9 | **296.9** | 284.9 |
| D-47 | 152 | **145.3** | 141.7 | - |
| Byrd | 238.9 | 225 | **226.9** | - |
| NEEM | 209.4 | 187.4 | **191.9** | 187.4 |
| Crete | 156.1 | **150** | 145.7 | - |
| km200 | 137.9 | 156.7 | **152.6** | - |
| WAIS divide | 225 | 206.5 | 206.5 | **233.3** |
| North GRIP | 248.4 | 226.2 | **236.5** | 224.5 |
| GRIP | 209.9 | **205.7** | **205.7** | 200 |
| B29 | 270.3 | 251 | **264.7** | 252.9 |
| Mizuho | 483.7 | **518.4** | 557 | - |
| Talos | 554.1 | **592.6** | 637.7 | 656.4 |
| B32 | 896.8 | 816.1 | **889.9** | 978.4 |
| EDML | 874.5 | 787.3 | **852.9** | 899.7 |
| SP | 1160.9 | 965.2 | 1002.8 | **1068.7** |
| DC | 2639.1 | 2473 | 2461 | **2557** |
| Vostok | 2814.3 | 2960.4 | **2810.4** | 2919.5 |
| Dome A | 2812.7 | 3024.8 | **2764** | - |

Supplementary Table S2: Comparison of Δage at the bottom of the firn for the different sites studied here. The main features already observed for the comparison of the standard deviation between modeled and measured density profiles (Table S1) are also observed here. This is the case for the worsened agreement between model and data at Talos and Mizuho when using the new parameterization or the improved model vs data agreement at low temperature and low accumulation sites of the bottom of the table (B32, EDML, South Pole, Dome C, Vostok, Dome A).

| Sites | Temperature scenario | Accumulation rate scenario | Calcium scenario |
|---|---|---|---|
| NGRIP | Dahl-Jensen et al., 1998 ; Kindler et al., 2014 | Bazin et al., 2013 | Svensson (comm. pers.) and Seierstad et al., 2014 |
| Dome C | Stenni et al., 2010 | Bazin et al., 2013 | Fischer et al., 2007 and Lambert et al., 2012 |
| EDML | Stenni et al., 2010 | Bazin et al., 2013 | Fischer et al., 2007 |
| WAIS-Divide | Buizert et al., 2015 and WAIS Divide Project Members, 2013 | Fudge et al., 2016 | Buizert et al., 2015 |
| Berkner Island | Capron et al., 2013 | Capron et al., 2013 | Capron et al., 2013 |
| GISP2 | Cuffey and Clow, 1997 | Cuffey and Clow 1997 | X |
| Talos Dome | Buiron et al., 2011 | Bazin et al., 2013 | Schüpbach et al., 2013 |
| Vostok | Cuffey and Vimeux, 2001 | Bazin et al., 2013 | Legrand et al., 1988 |

*Supplementary Table S3:* *References of the temperature, accumulation rate and calcium scenarios over the last deglaciation for NGRIP, Dome C, EDML, WAIS-Divide, Berkner Island, GISP2, Talos Dome and Vostok. The temperature (T) scenario for NGRIP has been built so that its temporal evolution reflects the $\delta^{18}O$ temporal evolution (T=a$\delta^{18}$O+b) with an adjustment of a and b to reflect the ~ 20°C temperature change between LGM and Holocene (Dahl-Jensen et al., 1998) as well as abrupt change of temperature of ~12.5°C and ~10.5°C at the onset of the Bølling-Allerød and end of Younger Dryas as given by Kindler et al. (2014)*

| EH – LGM (‰) | Dome C | EDML | NGRIP | WAIS-D |
|---|---|---|---|---|
| Measured | 0.1051 | 0.0538 | -0.0986 | -0.0780 |
| Old | -0.0404 | -0.0446 | -0.1080 | -0.0639 |
| Old + dust | 0.0904 | 0.0545 | -0.0088 | -0.0078 |
| New parameterization (Table 1) | 0.0519 | -0.0313 | -0.112 | -0.0909 |
| New parameterization + dust | 0.0930 | 0.0651 | -0.00937 | -0.0129 |
| Test A | -0.0183 | -0.0331 | -0.0842 | -0.0648 |
| Test A + dust | 0.0714 | 0.0474 | -0.00282 | -0.00658 |
| Test B | -0.0336 | -0.0644 | -0.147 | -0.10 |
| Test B + dust | 0.106 | 0.0604 | -0.0272 | -0.0225 |
| New parameterization + dust following Freitag parameterization for the Pimienta – Barnola model | 0.0915 | 0.0646 | -0.00717 | -0.0092 |

*Supplementary Table S4: Results of the difference between the average of the Early Holocene (EH) and the average of the Last Glacial Maximum (LGM) for the sensitivity tests displayed on Figure 7 for the 4 sites described in the main text.*

| NGRIP | Δage (old version) | Δage (new version) | Δage (new version + dust) | DATA |
|---|---|---|---|---|
| Bølling /Allerød | 870 years | 920 years | 740 years | 1040±100 years |
| End of Younger Dryas | 760 years | 820 years | 640 years | 800±100 years |

*Supplementary Table S5: Comparison of measured and modelled Δage with different parameterizations of the model at two different points of the last deglaciation at NGRIP. The only way to compare Δage model output with data is actually on the abrupt warming recorded very clearly both in the $\delta^{18}O$ in the ice phase and $\delta^{15}N$ in the gas phase on the NGRIP ice core because we have an accurate associated timescale (GICC05). The agreement between data and model is slightly better when using the new version but the addition of dust leads to a strong deterioration as observed on the $\delta^{15}N$ profiles.*

*NB: the uncertainty on the Δage from the data is mainly linked to the resolution of the $\delta^{15}N$ signal.*

**Supplementary Text S1**: Model input files and running conditions

For paleoclimate applications, we have one input file for each site. The file has 5 columns: depth, age, difference of temperature with respect to present-day surface temperature, accumulation rate and calcium concentration. Except for WAIS-D, the spatial resolution is constant from the top to the bottom of each ice core and varies between sites between 1 and 2 m. Such resolution is largely sufficient to depict the temporal variability of temperature and accumulation over the last deglaciation (the lowest temporal resolution is 200 years at Vostok for the LGM; at NGRIP, the temporal resolution corresponding to 1 m spatial resolution in the input file is about 50 years at LGM). For WAIS-D, the input files were designed so that the temporal resolution is constant to 10 years all along the ice core.

The model running conditions are very similar to those described in Goujon et al. (2003). For density calculation, the depth step is 0.25 m down to 150 m depth, 2.5 m depth down to 550 m depth and 25 m depth below. We only simplified the boundary between the 2.5 m and 25 m grids to guarantee that the change occurs at a grid node (Goujon et al. (2003) used a criterion based on the difference between pure ice density and the modelled density). Relative coordinates z/H (H being the ice thickness) are used for solving the ice sheet vertical velocity equation as in Goujon et al. (2003).

In this study, we modified the time step: Goujon et al. (2003) used a one year time step for all sites, whereas we adjusted the time step in link with the maximum accumulation rate in the input file. Indeed the accumulation rate defines the firn sinking speed. Stable results are obtained with a time step of 0.3/max(Ac) where Ac is the accumulation rate in m w.eq.yr$^{-1}$. Thus our time step is shorter than one year only at very high accumulation rate sites (Ac > 0.3m w.eq./yr). Numerical tests showed that with the above depth and time grid, Eulerian and Lagrangian calculations of the density profiles lead to consistent results. However, for gas age calculations, a lagrangian tracking of firn layers is necessary to simulate the detailed shape of $\delta^{15}$N anomalies during Dansgaard-Oeschger events in Greenland.

As in Goujon et al. (2003), a limitation of our model is that the ice sheet thickness and lower boundary condition do not vary with time. Two parameters are imposed at the boundary between the ice and bedrock: the temperature and vertical speed. Basal temperature is inferred from borehole measurements. If it is colder than the melting temperature estimated as a function of hydrostatic pressure (equation 8 in Goujon et al., 2008), the basal vertical speed is nul. Otherwise, the melting temperature is used and the basal vertical speed (melting rate) needs to be specified in input.

*Supplementary Text S2:*

The major discrepancies between our model results with dust and $\delta^{15}$N data with dust occur outside the range of calcium concentrations at modern sites. We hence illustrate here how the addition of simple thresholds on a minimum and maximum effect of calcium can strongly reduce the main discrepancies between our model results with dust and $\delta^{15}$N data. We have thus adjusted the dust parameterization proposed by Freitag with two threshold ($Ca_{min}$ and $Ca_{max}$) such as:

- If $Ca < Ca_{min}$ : use the Freitag parameterization with $Ca = Ca_{min}$
- If $Ca > Ca_{max}$: use the Freitag parameterization with $Ca = Ca_{max}$
- If $Ca_{min} < Ca < Ca_{max}$, use the original Freitag parameterization

In this study, we have chosen values of 2ng/g and 50ng/g for respectively $Ca_{min}$ and $Ca_{max}$. Implementing threshold values on calcium (blue lines on the Figure S10) reduces the largest inconsistencies between model results and $\delta^{15}$N data, in particular at NGRIP (through the threshold at high calcium concentration) and at WAIS (through the threshold at low calcium concentration).

---

## Author Response (AR2)

Dear Hubertus,

We thank you and the reviewer for the careful reading of our revised manuscript.
We detail below how we have addressed the new comments in this update version of the manuscript. We hope that this revised version can be accepted for publication in climate of the past,

Camille Bréant and the co-authors.

**Main comments from the editor (blue)**:

- to warn the reader, please make clear in the abstract and manuscript early on that the changes done in the model do not provide a solution to the firnifaction issue at all sites.

We have precised that the improvement was only visible at "coldest sites of East Antarctica (Vostok, Dome C)".
We have also included a sentence at the end of the abstract:
"We thus do not provide a definite solution to the firnification at very cold Antarctic sites but propose potential pathways for future studies."

- explain more carefully how you quantified the model parameters in Table 1 and 3

Some sentences have been added to address this issue:
- P. 12: "Hundreds of sensitivity tests have been performed imposing 3 activation energies at 3 different typical temperatures, $T_i$. The initial values for $Q_i$ are chosen as explained above (high value for $Q_1$ [Jacka and Li, 1994], classical value between 60 and 70 kJ/mol for $Q_2$ and low value for $Q_3$ to increase the densification rate at low temperature). The initial values for $a_i$ are derived through $a_i * exp(-Q_i/RT_i) = a_0 * exp(-60000/RT_i)$ and variations around the initial values of $Q_i$ and $a_i$ are randomly generated. Only the values leading to realistic densification speed are kept and we found the optimal tuning through reduction of the mismatch between model and data especially for the deglacial amplitude of $\delta^{15}N$ in Dome C and Vostok."
- Caption of Table 3: "*These values have been chosen to illustrate the effects of varying activation energy for the different temperature ranges on the densification rate for the different ice core deep drilling sites (cf figure 8) and support the tuning presented on Table 1.*"

- rephrase the discussion of the physical processes as outlined by the external reviewer.

- We have fully rewritten the caption of Figure 2 based on the paper by Ashby, 1974:
"*Different sintering mechanisms of snow for different temperatures proposed by analogy with the hot ceramic sintering (inspired by Figure 1 in Ashby, 1974). Note that more sintering mechanisms can be found in the literature: in its initial figure, Ashby (1974) mentioned 6 different mechanisms but only 2 permit densification (lattice diffusion and boundary diffusion from grain boundary). The attributions of 3 different mechanisms for the firn densification model based on the powder aggregate study from Ashby (1974) is only a working hypothesis here.*"

- please include the corrections and clarify the points raised in my annotated pdf attached to my editor comment

- All comments have been taken into account except the one on l. 216 ("but Arrhenius expressions cannot represent deformation effects linked to ice melting" glacial ice has much smaller grain sizes due to high dust content. Does this already apply for the firn. If yes, is the contact area (hence the deforming pressure) in glacial firn the same as in interglacial firn" Indeed, this is a very interesting but complex question for which we could not find a simple answer. The absolute contact area per grain is directly dependent on grain size, but the relative contact area (a) per squared mean grain size ($l^2$) used to relate the load and effective pressures in Goujon et al. (2003) is less easy to predict. Moreover, based on image analysis, Arnaud et al. (Ann. Glaciol., 1998) conclude that the observed contact areas are much larger than predicted by the firn model especially in the early stage of densification where such contacts prevent grain boundary sliding. They thus emphasize the role of clusters of snow crystals formed at the beginning of the densification process. Micro-mechanical studies using Discrete Elements Method have investigated the evolution of the contact area in monodispersed (e.g. Martin et al., Journal of the Mechanics and Physics of Solids, 2003) and polydispersed (Wiacek et al., International Journal of Solids and Structures, 2014) granular materials. Martin et al., 2003 emphasize the role of particle rearrangement and friction between particles (which may be affected by dust pinning) but conclude that for isostatic compaction, the main effect is to widen the distribution of contact areas, leading to a less uniform effective pressure than expected from the homogeneous strain field solution. Wiacek et al., 2014 investigated the effect of particle size distribution and observed little sensitivity of the macromechanical response to particle size heterogeneity.
Earlier ice core studies such as Montagnat and Duval, EPSL, 2000 describe different dominant mechanisms for ice structure evolution in specific depth ranges: normal grain growth driven by the decrease in free energy that accompanies a reduction in grain boundary area in the upper several hundreds of meters of the ice sheet and recrystallization processes below.
Durand et al., JGR Earth Surface, 2006 further discuss the roles of soluble and insoluble impurities on grain growth. They suggest that high soluble impurity content does not necessarily imply a slowdown of grain growth kinetics, whereas the pinning of grain boundaries by dust explains all the observed modifications of the microstructure in EPICA Dome Concordia ice core.
Kipfstuhl et al. 2009 observed dynamic recrystallization features in the EDML firn, which may be affected by dust pinning.
Another possible process, very close to the surface, is a dust-related decrease the snow albedo (especially related to black carbon) which may favor metamorphism and thus grain growth (e.g. Casey et al., JGR Atmos., 2017).

As we could not reach a clear conclusion, we suggest to avoid complexifying the manuscript by including this discussion.

Two other comments are associated with significant changes:
>> A sentence has been added " The improvement through dust softening is particularly important at EDML where the change of activation energy had only a modest effect."

L627: An age distribution with a width of 20% of delta-age is much too wide. The Kohler et al. 2011 paper is very clearly flawed. It is very instructive to read the exchange between the author and reviewer for that paper (http://www.clim-past.net/7/473/2011/cp-7-473-2011-discussion.html) – where the reviewer is correct (in my view). The wide filter they use results in a large overshoot in CO2 of up to 35 ppm, which we now know is incorrect because of the high-res WAIS Divide record, which proves that the 35 ppm overshoot is only a fantasy and not real.

>>> We have used the published value and were not aware of the controversial aspect of this value. We understand your point. Still, using a lower width does not change the amplitude of the difference in modelled d15N between the LGM and the EH. This is stated in the new version.

Figure 7: It seems that dust is much more efficient at solving the glacial d15N mismatch at EDML and EDC than the activation energy alone. This could be stressed more clearly.

Finally, figure 7 has also been changed following the comments of editor to avoid the "messy" aspect and the caption rewritten.

**Comments from the reviewer (blue)**

* Both reviewers argued that there is not really a good physical underpinning of the model. But Figure 2 still suggests a physical mechanism for each value of Q, even though Q3 is an order of magnitude too small to for any known process, and Q1 (suggested to be vapor diffusion) has a value that seems too high at Q1=110 kJ/mol. Vapor diffusion scales with the vapor pressure, and the enthalpy of sublimation in ice is only 51 kJ/mol.
Having an empirical model is fine of course, and it should be made more explicit that the final model Q factors do not correspond to known individual processes (unlike what is suggested by Fig. 2).

>> The caption of figure 2 has been rewritten based on Figure 1 from Ahsby (1974) and we emphasize on the limitations of using analogy with powder ceramics. Then, we have also removed the association between mechanism and activation energy 1, 2 and 3 so that we cannot associate a particular Q value to a given process.

* In their preferred model (Table 1), process 1 (Q1 = 110 kJ/mol) doesn't do much. At all relevant temperatures, process 2 is at least an order of magnitude larger.

>> The Early Holocene temperature for WAIS and NGRIP is about -31°C. At that temperature and taking ai and Qi values from table 1, processes 1 and 2 have the same order of magnitude (only a factor of 2 between the two processes). Moreover, we have included in our study sites with quite high temperatures (Dye 3, mean temperature of -18°C) so that process 1 is important in this study.

* The authors did not do a clear significance test of the improvement of the model fit to present day observations (d15N, Delta-age). They do now state that the improvement is

"insignificant". This is not how it should be presented. Either the improvement is statistically significant (in which case you can call it an improvement), or the improved model fit is statistically not significant, in which case you must conclude that both models perform equally well. An insignificant improvement (L576) is not an improvement, and cannot be called such. The authors should instead state both models perform equally well.

>>> We now state that both models perform equally well.

* The authors include tests of Delta age, which is a big improvement. However, it is unclear where the delta-age "data" values (Table S2) come from. It is not easy to know Delta-age, unless one has a very accurate ice age scale (from layer counting or volcanic references), coupled with a very accurate gas age scale (e.g. from firn air sampling, or high-res CH4 data). Where do these numbers come from? In some cases they are different from published values.

>> The "data" values for the $\Delta$age have been deduced from the identification of the peak of the $d^{15}N$ and mid-slopes of $d^{18}O_{ice}$ increases recording the most abrupt warming phases of the last deglaciation associated with the Bølling-Allerød and the end of the Younger Dryas. The $d^{15}N$ record is given in Kindler et al. (2014) and the $d^{18}O_{ice}$ record is given in NGRIP comm members (2004). The respective depths for $d^{15}N$ peaks are 1629.4 and 1515.3 m. The respective depths for the $\delta^{18}O_{ice}$ mid-slopes are 1604.2 and 1491.5 m. We obtained the $\Delta$age indicated in the Table S2 when translating these depth differences in age through the use of the GICC05 age scale. This is now stated in the caption.

The other minor comments have been taken into account.

[revised manuscript text omitted]